# Source apportionment of highly time resolved elements during a firework episode from a rural freeway site in Switzerland

Pragati Rai[1], Markus Furger[1], Jay G. Slowik[1], Francesco Canonaco[1], Roman Fröhlich[1], Christoph Hüglin[2], María Cruz Minguillón[3], Krag Petterson[4], Urs Baltensperger[1] and André S.H. Prévôt[1]

[1]Laboratory of Atmospheric Chemistry, Paul Scherrer Institute, Villigen PSI, 5232, Switzerland
[2]Laboratory for Air Pollution / Environmental Technology, Empa, 8600 Dübendorf, Switzerland
[3]Institute of Environmental Assessment and Water Research (IDAEA), CSIC, 08034 Barcelona, Spain
[4]Cooper Environmental Services (CES), 9403 SW Nimbus Avenue, Beaverton, OR 97008, USA

*Correspondence to*: André S. H. Prévôt (andre.prevot@psi.ch) and Markus Furger (markus.furger@psi.ch)

**Abstract.** The measurement of elements in $PM_{10}$ was performed with 1 h time resolution at a rural freeway site during summer 2015 in Switzerland using the Xact 625 multi-metals monitor. On average the Xact elements (without accounting for oxygen and other associated elements) make up about 20 % of the total $PM_{10}$ mass (14.6 µg m$^{-3}$). We conducted source apportionment by positive matrix factorization (PMF) of the elemental mass measurable by the Xact (i.e., major elements heavier than Al), defined here as $PM_{10el}$. Eight different sources were identified in $PM_{10el}$ (elemental $PM_{10}$) mass driven by the sum of 14 elements (notable elements in parentheses): fireworks-I (K, S, Ba, Cl), fireworks-II (K), sea salt (Cl), secondary sulfate (S), background dust (Si, Ti), road dust (Ca), non-exhaust traffic-related (Fe) and industrial (Zn, Pb). The major components were secondary sulfate and non-exhaust traffic-related followed by background dust and road dust factors, explaining 21 %, 20 %, 18 % and 16 % of the analysed $PM_{10}$ elemental mass, respectively, with the factor mass not corrected for oxygen content. Further, there were minor contributions (on the order of a few percent) of sea salt and industrial sources. The regionally influenced secondary sulfate factor showed negligible resuspension, and concentrations were similar throughout the day. The significant loads of the non-exhaust traffic-related and road dust factors with strong diurnal variations highlight the continuing importance of vehicle-related air pollutants at this site. Enhanced control of PMF implemented via the Source Finder software (SoFi Pro version 6.2, PSI, Switzerland) allowed for a successful apportionment of transient sources such as the two firework factors and sea salt, which remained mixed when analysed by unconstrained PMF.

## 1 Introduction

Ambient particulate matter (PM) plays a major role in affecting human health and air quality. Trace elements represent a minor fraction of the atmospheric aerosol on a mass basis, but they can act as specific markers for several emission sources. The short- or long-term exposure of ambient particulate matter (PM) has significant negative effects on human health (Dao et al., 2012; Ancelet et al., 2012; Zhao and Hopke, 2004; Pope and Dockery, 2006; Dockery et al., 1993; Zhou et al., 2018). Cakmak et al. (2014) found significant association of acute changes in cardiovascular and respiratory physiology with $PM_{2.5}$ metals in

Ontario, Canada. In Stockholm, Meister et al. (2012) estimated a 1.7 % increase in the daily mortality per 10 µg m$^{-3}$ increase in the coarse fraction of PM. Metallic components of PM, especially the fine fraction of elements such as Fe, Ni, Cu, V, Pb and Zn, appear to be a significant cause of both pulmonary and cardiovascular diseases (Kelly and Fussell, 2012). Airborne particles and associated (trace) elements originate from various emission sources, such as motor vehicles, power plants,

construction activities, in a broad size range. Among them, traffic-related emissions are of particular interest (Brauer et al., 2002). Traffic-derived PM has a high risk of respiratory illness, asthma and cardiovascular diseases, resulting in an increased rate in mortality (Kelly and Fussell, 2011). Traffic-related PM is emitted mainly as exhaust emissions (tailpipe exhaust from gasoline and diesel engines) and non-exhaust emissions (resuspension of road dust and brake and tire wear emissions) (Lawrence et al., 2013; Lin et al., 2015; Thorpe and Harrison, 2008; Zhou et al., 2018; Grigoratos and Martini, 2015; Amato

et al., 2014b; Bukowiecki et al., 2010). Exhaust emissions are predominantly in the fine fraction of PM, whereas non-exhaust emissions contribute mostly to the coarse fraction (Amato et al., 2011; Thorpe and Harrison, 2008). Exhaust emission-related elements are comprised of Pb, Zn, Ni and V (Lin et al., 2015; Minguillon et al., 2012), while the non-exhaust emissions are dominated by Fe, Cu, Ba, Ca, Sb, Sn, Cr and Zn from brake lining and tire wear. The presence of Fe in brake lining can reach up to 60 % by weight (Chan and Stachowiak, 2004; Schauer et al., 2006). Brake pads are usually filled with $BaSO_4$, while Sb,

Sn and Mo sulfides are often added as lubricants, and Cu, Cr and Zn are major additives to lubricating oils and normally used to improve friction (Thorpe and Harrison, 2008; Amato et al., 2014a). Sb has been identified as a major tracer of brake wear, due to significant (1–5 %) percentage of Sb in brake linings in the form of stibnite ($Sb_2S_3$) (Grigoratos and Martini, 2015; Bukowiecki et al., 2009). It has been reported that asphalt pavement-induced particles were characterized mainly by high concentrations of Cu, Cr, Ni, As and Pb (Yu, et al., 2013) as well as Ca, Si, Mg, Al, Fe, P, S, Cl, K, V, Mn, Na (Fullova et al.,

2017). Therefore, it is important to monitor traffic emissions for health risk assessment, the study of which relies heavily on the source apportionment (SA) of PM using chemically speciated data (Zhou et al., 2018).

Source quantification and characterization is an important step in understanding the relationship between source emissions, ambient concentrations, and health and environmental effects. SA by receptor models has been widely used in recent years to identify and apportion the contributions of various sources to the airborne PM concentrations. Positive Matrix Factorization

(PMF) is one of the most widely used receptor models for SA of trace elements (Rahman et al., 2011; Ancelet et al., 2012; Cesari et al., 2014; Ducret-Stich et al., 2013; Kim et al., 2003; Rai et al., 2016; Zhang et al., 2013; Harrison et al., 2011; Hedberg et al., 2005). However, a limited number of studies are available for trace elements emission sources with high time resolution (hourly or sub-hourly) (Visser et al., 2015; Crilley et al., 2016; Bukowiecki et al., 2010; Richard et al., 2011; Dall'Osto et al., 2013; Manousakas et al., 2015; Jeong et al., 2019; Wang et al., 2018, among others). Hourly elements data

can be used to explore the diurnal patterns of emissions from traffic, biomass burning and industrial sources, thereby yielding more accurate and exposure-relevant SA results. Currently, there are very few offline instruments available for field sampling of elements with high time resolution, such as the rotating drum impactor (RDI) (Bukowiecki et al., 2008), the streaker sampler (PIXE International Corporation) (Lucarelli et al., 2011) and the semi-continuous elements in aerosol sampler (SEAS) (Kidwell and Ondov, 2001). The large quantity of samples generated by these samplers requires a labour intensive and time

consuming offline analysis. These offline analyses require high precision and low detection limit techniques such as synchrotron radiation induced X-ray fluorescence spectrometry (SR-XRF) of aerosol samples collected with a RDI, particle induced X-ray emission (PIXE) with the streaker sampler, and graphite furnace atomic absorption spectrometry (GFAAS) with the SEAS. In practice, the offline samplers lead to undesirable trade-offs between time resolution and data coverage even for short duration field campaigns, whereas highly time-resolved long-term measurements are impractical. A recently introduced online high time resolution instrument can collect samples and perform analysis for elements simultaneously in a near real time scenario for long-term measurements without waiting for laboratory analysis. The XRF-based Xact 620, 625 and the newer 625i ambient metals monitors (Cooper Environmental Services, Beaverton, Oregon, USA) have been developed in recent years and have been used in several field studies (Fang et al., 2015; Cooper et al., 2010; Furger et al., 2017; Park et al., 2014; Phillips-Smith et al., 2017; Tremper et al., 2018; Chang et al., 2018; Liu et al., 2019). However, only 10 studies included SA on Xact data (Park et al., 2014; Fang et al., 2015; Phillips-Smith et al., 2017; Chang et al., 2018; Liu et al., 2019; Ji et al., 2018; Sofowote et al., 2018; Jeong et al., 2019; Belis et al., 2019; Cui et al., 2019).

The main focus of this work is the exploration of the use of the Xact for source apportionment in Europe where the concentrations are considerably lower than in polluted areas in Asia. In the present study, we conducted SA using PMF to characterize the source contributions of highly time-resolved metals during a three-week campaign at a traffic-influenced site in Härkingen, Switzerland. PMF was implemented through the multilinear engine-2 (ME-2) solver and controlled via SoFi, which allows for a comprehensive and systematic exploration of the solution space (Bozzetti et al., 2016; Canonaco et al., 2013). The rotational control available in ME-2 provides a means for treating extreme events such as fireworks within a PMF analysis. Such events are often excluded from the PMF input matrix to avoid modelling errors due to the pulling of a solution by outliers (Ducret-Stich et al., 2013; Norris et al., 2014). Although a few studies have already been carried out in the past at this location (Lanz et al., 2010; Hueglin et al., 2006; Furger et al., 2017), none of them have reported SA on elements.

## 2 Experimental setup and data analysis

### 2.1 Sampling location

$PM_{10}$ sampling was performed from 23 July until 13 August 2015 in Härkingen, Switzerland, a permanent station of the Swiss Air Pollution Monitoring Network (NABEL) (Fig. 1). Extreme firework episodes were captured during the Swiss National Day Celebration (1 August). The site is situated next to the A1 freeway, which is the main traffic route between eastern (Zurich) and western (Bern) Switzerland. The measurement site is bordered by agricultural areas to the west and north while there are villages in the south and east direction. There is a metal processing company to the south-east across the freeway which manufactures wheels for passenger cars and commercial vehicles, and some small scale industrial buildings to the north-west. The emissions reaching the measurement site depend on wind direction. The site is strongly influenced by local road traffic emissions when winds come from the southern sector, while the northern wind sector represents the air constituents from a

rural area. A detailed description of the sampling site can be found in previous studies (Furger et al., 2017; Hueglin et al., 2006)

## 2.2 Sampling and analysis

Sampling and analysis was conducted with an Xact 625 ambient metals monitor equipped with a $PM_{10}$ inlet. The instrument was set up to quantify 24 elements (Si, S, Cl, K, Ca, Ti, V, Cr, Mn, Fe, Co, Ni, Cu, Zn, As, Se, Cd, Sn, Sb, Ba, Pt, Hg, Pb and Bi) with 1 h time resolution. In addition, 24-h $PM_{10}$ samples were collected by a HiVol sampler (Digitel DA-80H, DIGITEL Hegnau, Switzerland) with quartz fiber filters (Pallflex XP56 Tissuquartz 2500QAT-UP, Pall, AG, Switzerland). Ten of these 24-h $PM_{10}$ samples were analysed by inductively coupled plasma optical emission spectrometry (ICP-OES) for the concentrations of Na, Mg, Al, P, S, K, Ca, Ti and Fe. Moreover, the station was equipped with other instruments such as a tapered element oscillating microbalance filter dynamics measurement system (TEOM FDMS 8500, Thermo Fischer Scientific, MA, USA) for 10 min $PM_{10}$ mass concentration measurements, a multi angle absorption photometer (MAAP, Thermo 5012, Thermo Fischer Scientific, MA, USA) for black carbon (BC) measurements in $PM_{2.5}$ and standard meteorological sensors (temperature, wind speed and direction, precipitation). The trace gases were measured with conventional instruments by the Swiss National Air Pollution Monitoring Network, NABEL (Empa, 2011). $NO_x$ was measured by chemiluminescence spectroscopy (Horiba APNA 370). The time resolution of all these instruments was 10 min. The station also provided hourly traffic counts for the freeway in the form of total number of vehicles, number of heavy-duty vehicles (HDV), and the number of light-duty vehicles (LDV). Specifically for the campaign, a quadrupole aerosol chemical speciation monitor ( Q–ACSM 140-145, Aerodyne Inc, Billerica, MA, USA) with vacuum aerodynamic diameters smaller than 1 µm (non-refractory-$PM_1$) was deployed (Ng et al., 2011; Crenn et al., 2015) and the data were used for comparison of the factors during the SA analysis. The time resolution of the ACSM was set to 30 min.

## 2.3 PMF using ME-2

Positive matrix factorization is one of the most common receptor models based on a weighted least squares fit (Paatero and Tapper, 1994). It is used to describe the variability of a multivariate dataset as the linear combination of a set of constant factor profiles and their corresponding time series as shown in Eq. (1) in cell notation:

$$x_{ij} = \sum_{k=1}^{p} g_{ik} f_{kj} + e_{ij} , \tag{1}$$

where **X**, **G**, **F** and **E** represent the data matrix, factor time series, factor profiles and residual matrices, respectively, while $i$, $j$ and $k$ indices denote time, element, and factor number. The index $p$ represents the total number of factors in the PMF solution. The PMF model iteratively solves Eq. (1) by minimizing the object function ($Q$), defined as:

$$Q = \sum_i \sum_j \left( \frac{e_{ij}}{s_{ij}} \right)^2 , \tag{2}$$

Here, $s_{ij}$ corresponds to the measurement uncertainty (error matrix) for the input point $ij$.

The PMF algorithm was solved using ME-2 (Paatero, 1999), which enables an efficient exploration of the solution space by introducing *a priori* information to **G** and / or **F** into the PMF model. Using the constraining technique of the *a*-value, one or more factor profile/factor time series can be confined by the scalar $a$ ($0 \leq a \leq 1$), which can be applied to the entire profile/time series or to individual variables/data points of the profile/time series. The scalar $a$ defines how much the resolved factors are allowed to deviate from the input profile/time series, according to:

$$f'_{kj} = f_{kj} \pm a \times f_{kj} , \tag{3}$$

$$g'_{ik} = g_{ik} \pm a \times g_{ik} , \tag{4}$$

where the subscript $j$ varies between 0 and the number of variables and $i$ varies between 0 and the number of measured data points in time. $f_{kj}$ and $g_{ik}$ are the starting value used as *a priori* knowledge from the base case solution in this SA study, and $f'_{kj}$ and $g'_{ik}$ are the resulting values in the solution. Normalization in Eq. (3) and (4) can lead to the resulting values slightly outside the specified *a*-value boundaries. This method reduces the available solution space and directs the solution towards an optimized and environmentally meaningful solution. The SoFi Pro (Canonaco et al., 2013) which is coded in the Igor Pro software environment (Wavemetrics, Inc., Portland, OR, USA) was used for the PMF configuration and analysis.

## 2.4 Conditional bivariate polar function (CBPF) plots

CBPF is a data analysis tool to identify the direction of source contributions and was applied to the PMF source factors. Polar plots are used to present the CBPF analyses, where the number of events with a concentration greater than the i[th] percentile (0 to 100) is plotted as a function of both wind speed and direction, as shown in Eq. (5):

$$CBPF = \frac{m_{\theta,r}}{n_{\theta,r}} , \tag{5}$$

where $m_{\theta,r}$ is the number of samples in wind sector $\theta$ and wind speed sector $r$ with a concentration greater than the i[th] percentile and $n_{\theta,r}$ is the total number of samples with the same wind direction and speed (Carslaw and Beevers, 2013). The resultant CBPF polar plots present the probability that high concentrations of a pollutant correspond to a particular wind direction and speed and can give insight into the contributions from local and regional sources.

## 3 SA method and solutions selection

### 3.1 PMF input preparation

In our study, the PMF input consists of a data matrix and an error matrix of hourly element measurements, where the rows represent the time series (456 points with 1 h steps) and the columns contain the elements (14 variables). The input preparation

of PMF was done by excluding some specific elements for better source apportionment results. A common approach for the choice of species to include in the PMF input depends on: 1) the percentage of data below detection limit (Polissar et al., 1998), and 2) Xact data comparison with offline 24-h $PM_{10}$ filters (Pearson coefficient of determination $r^2$). The minimum detection limits (MDL), $r^2$ and data points below MDL of Xact elements are listed in Furger et al. (2017). The MDLs were given by the manufacturer and were calculated by using the sensitivity of the element and counts in the region of interest of a blank section of the tape from where $1\sigma$ interference-free detection limits are reported. Elements (% of data points below MDL, $r^2$ value) which had more than 50 % of data points below MDL and low $r^2$ ($< 0.5$) between Xact and offline data were not included in the PMF input, such as: V (98 %, 0.57), Co (100 %, 0.05), Ni (32 %, 0.22), As (96 %, 0.5), Se (62 %, 0.3), Cd (87 %, 0.18), Sn (15 %, 0.27), Sb (6%, 0.42), Hg (13 %, 0.12) and Pt (98 %, not measured on the filters). The element Bi (93% data points below MDL) was an exception to include in the PMF input due to an excellent correlation between Xact and offline data ($r^2 = 0.98$) during fireworks peaks. The detailed description of the Xact data quality is given in the previous study (Furger et al., 2017). Missing data points in time (e.g,. a power failure during sampling or a filter tape change) were removed from the data and error matrices. In the present work, if the element concentration was less than or equal to the MDL provided, the error matrix element $s_{ij}$ was calculated using the Eq. (6), and if the concentration was greater than the MDL provided, the error matrix element $s_{ij}$ was calculated using Eq. (7) (Reff et al., 2007; Tian et al., 2016; Polissar et al., 1998):

$$s_{ij} = \frac{5}{6} \times MDL_j , \tag{6}$$

$$s_{ij} = \sqrt{(p_j \times x_{ij})^2 + (MDL_j)^2} , \tag{7}$$

where $\mathbf{X}$ indicates the data matrix, while subscripts i and j are indices for time and elements. In this study, an estimated analytical uncertainty ($p_j$) of 10 % was used to derive the error matrix data set (Kim et al., 2005; Kim and Hopke, 2007; Tian et al., 2016; Ji et al., 2018), which did not change the PMF solution. The metal-specific analytical uncertainty was also considered from the previous studies (Jeong et al., 2016; Phillips-Smith et al., 2017), where it was calculated on the basis of high/medium-concentration metal standards laboratory experiments and the additional 5-10% flow rate accuracy, which yielded similar PMF solutions compared to an overall 10 % analytical uncertainty.

**3.2 PMF setup**

An important step in the PMF analysis is the selection of the number of factors by the user, as mathematical diagnostics alone are insufficient for choosing the correct number of factors (Ulbrich et al., 2009; Canonaco et al., 2013). The selection of factors is often based on an analysis of total $Q$ or $Q/Q_{exp}$, scaled residuals ($e_{ij}/s_{ij}$), comparison of time series of the factor with external tracers, as well as diurnal patterns, and the evaluation of the residual time series as a function of the number of resolved factors.

In a first step, we examined a range of solutions with three to 10 factors at 10 seeds (number of PMF repeats) from unconstrained runs. The unconstrained PMF solution resulted in mixed factors, such as sea salt mixed with fireworks, in all factors solutions. We show an example of a mixed nine-factor solution in Fig. S1. This is likely because of the very high concentration and variation in composition of fireworks emissions during the fireworks period. Because the signal-to-noise ratio is very high, imperfections in the model description exert a strong influence on $Q$, and the model therefore tries to compensate by assigning fireworks mass to other factors. This was particularly evident for the sea salt and secondary sulfate factors, where constraints on factor profiles and/or time series were necessary to obtain clean separation. Here we discuss the method for achieving this separation.

The input data set was divided into two parts: fireworks days (FD; 31 July–4 August) and non-fireworks days (NFD; all days except 31 July–4 August). To obtain a specific fireworks profile, we further selected only fireworks hours (FH; 31 July 21:00– 1 August 07:00 LT (local time = Coordinated Universal Time + 2 h) as input data. The PMF analysis was performed on the NFD, FD, FH and the complete datasets separately for three to 10 factors, with each of these solutions investigated with different seeds (each seed represents a different random initialization).

The unconstrained NFD PMF analysis resolved seven factors at all seeds such as sea salt, secondary sulfate, non-exhaust traffic-related, road dust, background dust, industrial and a K-rich factor. The sea salt factor profile shows excellent correlation ($r^2 = 0.99$) between the Cl and the identified sea salt factor time series. Solutions with less than seven factors showed significant scaled residuals for elements and time series, while solutions with more than seven factors revealed a split of the non-exhaust traffic-related, industrial and background dust factors. The NFD analysis therefore provides a sea salt profile that can be used as a constraint in the complete dataset.

The unconstrained PMF analysis of the complete dataset identified a secondary sulfate factor (most of the S is apportioned in this factor, with 91 % of the factor mass) in the nine-factor solution (Fig. S1). The identified secondary sulfate factor time series correlated very well with ACSM sulfate ($r^2 = 0.91$) (Fig. S1) at 24 seeds out of 100 seeds while $r^2$ was $\leq 0.88$ for the remaining 76 seeds. Although these $r^2$ are quite similar at all 100 seeds, the solution characteristics are notably different at 24 seeds ($r^2 = 0.91$) and 76 seeds ($r^2 \leq 0.88$). For the nine-factor solution shown in Fig. S1, the secondary sulfate factor was slightly enhanced, which is in agreement with the ACSM sulfate, during the fireworks peaks (see details in Sect. 4.2 "secondary sulfate"), while the sea salt factor was strongly enhanced during the fireworks peaks. In contrast, for the solutions at 76 seeds, visible contamination (i.e., concentration spikes) during the fireworks plumes were observed in the secondary sulfate factor, suggesting mathematical mixing. S is one of the major components of fireworks emissions, the composition of which is highly variable. Because of their high sensitivity (and thus high signal-to-uncertainty ratio), imperfections in the model description of the fireworks composition yield high-signal residuals which strongly influence $Q$. The model responds by apportioning fireworks residuals to the other factors during the fireworks days. A similar issue also occurred for the sea salt factor due to the significant amount of Cl in the fireworks factor profile. Therefore, such events are often excluded from traditional PMF analyses (i.e., time periods removed from the input matrix), to avoid modelling errors due to the pulling of a solution by outliers. Here we take a different approach, exploiting the rotational control available in ME-2 to isolate environmentally

reasonable, unmixed solutions. The $Q/Q_{exp}$ values for all 100 seeds were in the range of 0.41- 0.45 (Fig. S2). The scaled residual (over the time series) of S in this solution (Fig. S1) was within the range of $\pm$ 3 with very small values, as shown in Fig. S3. The strong influence of fireworks in this solution yielded small values of the scaled residual for S which are consistent also with lower-than-expected $Q/Q_{exp}$ (0.4) values, suggesting an overestimation of uncertainties by the generic error model used herein (see Eqs. 6 and 7).

We then performed the constrained PMF analysis on the FD and FH datasets. Here we constrained the secondary sulfate factor profile (*a*-value 0.1) and the time series (*a*-value 0.01) using the results of the nine-factor unconstrained PMF analysis of the complete dataset. We tested several approaches for the secondary sulfate constraint: a) constraining factor profile only; b) constraining the factor time series during fireworks days only; c) constraining both the factor profile and the entire factor time series; d) constraining the factor profile and the factor time series during fireworks days only. Of the above methods, only c and d yielded secondary sulfate factors without visible mixing from the fireworks period. The approach d was used for PMF analysis as it provides maximum freedom to the algorithm.

In the FD and FH PMF analyses, the sea salt factor profile (*a*-value 0.1) was also constrained from the NFD unconstrained seven-factor PMF analysis. To resolve the unmixed sea salt factor time series from the fireworks, the background Cl concentration was calculated for the fireworks data points (K > 220 ng m$^{-3}$) only. A Cl concentration < 30 ng m$^{-3}$ was considered as a background Cl concentration, and the fireworks data points were replaced with the linear interpolation between the background Cl concentrations adjacent to the fireworks peaks. In this way 42 % of the data points were interpolated during the fireworks days. The calculated background Cl during the FD was constrained (with *a* -value 0.01) in the sea salt factor time series. After applying all the above four constraints, the FH PMF analysis identified a fireworks factor profile on the basis of the K / S elemental concentration ratio (~2.76) in black powder (Dutcher et al., 1999), and the concentration peak of 42 µg m$^{-3}$, which is close to the total elemental concentration peak of 48.4 µg m$^{-3}$ on 1 August 23:00 LT in the factor time series. The FD PMF analysis also identified a fireworks factor but the highest peak was 30 µg m$^{-3}$ on 1 August 23:00 LT in the factor time series and the K / S elemental concentration ratio was 2.55 in the factor profile. Therefore, the fireworks factor profile from FH PMF analysis was considered for the final complete dataset PMF analysis. The FD and FH PMF analyses resolved a five-factor solution with secondary sulfate, sea salt, fireworks, background dust and a K-rich factor with fireworks related elements. In the K-rich factor, the K / S ratio was slightly higher (3.56) than the black powder ratio. The unconstrained FH PMF analysis was also performed for two to five factors with different seeds. As shown in Fig. S4, the constrained five-factor $Q/Q_{exp}$ is 1.5, which is ~3 times higher than the unconstrained five-factor $Q/Q_{exp}$ but the same as the unconstrained 3-factor $Q/Q_{exp}$ (1.52). The increase in Q for the constrained five-factor FH PMF (unexplained variation (UEV) for each variable is between 2 % to 16 % with an average ~10 %) is significant when compared to the unconstrained five-factor FH PMF (UEV for each variable is between 1.5 % to 15 % with an average ~5 %), as shown in Fig. S5. The three-factor unconstrained FH PMF analysis validates the five-factor constrained FH PMF analysis, which suggests that both methods are working fine, and gives us more confidence to identify the fireworks factor profile.

Note that the number of time points contained in the input matrices for the data subsets (as opposed to the full dataset) are in some cases smaller than typical recommendations for ambient PMF (Olivier et al., 2019). This is most extreme in the case of the FH dataset, where only 11 time points are used. However, there are two important differences between PMF analyses of these sub-datasets and typical ambient PMF: (1) the sub-datasets are constructed to maximize variability of a factor or set of

factors; and (2) we are concerned only with accurately characterizing the profile(s) of these selected major factors. These two points work together to greatly reduce the number of time points required for the analysis (Fig. S6). A similar approach has been successfully applied by Froehlich et al. (2015), in which short-duration spikes in organic aerosol concentration were combined into a sub-dataset to determine an anchor profile related to local cigarette smoke, and Visser et al. (2015), in which a subset of a trace element dataset with high residuals was analysed separately to retrieve a factor profile related to industrial

emissions. In the final complete dataset PMF analysis, the factor profiles of fireworks, secondary sulfate and sea salt were constrained (*a*-value 0.1) while the time series of secondary sulfate and the calculated background Cl concentration interpolation were constrained during the fireworks period only (*a*-value 0.01). The solution that best represented the input data was an eight-factor solution, consisting of factors interpreted as sea salt, secondary sulfate, non-exhaust traffic-related, industrial, two dust-related and fireworks-related (two factors).

Residual analysis (*Q*-contribution over time series) of the PMF runs showed significant structure in the residuals (*Q* maximum value was 15 during fireworks period as shown in Fig. S8) for solutions having up to seven factors. Increasing the number of factors to eight gave evidence of structure removal, with mostly random errors remaining, by another fireworks-II factor which was explained by K, S, Ba, Ti, Cu, Bi (see details in Supplement, Figs. S8 and S9), while a further increase led to a new mixed factor of non-exhaust traffic-related and background dust which however showed a noisy diurnal pattern. All the variables

showed approximately unimodal scaled residuals between ±3 (Paatero and Hopke, 2003) (Fig. S10).

## 3.3 Uncertainty estimate of PMF results

The statistical and rotational uncertainties were explored by the bootstrap (BS) resampling strategy (Efron, 1979) and the exploration of the *a*-value space of the constrained information as well as random initialization of the unconstrained information. Briefly, the bootstrap algorithm generates new input matrices by randomly resampling variables from the original

input matrix. Each newly generated PMF input matrix had a total number of samples equal to the original matrix (456 samples); although some of the original 456 samples were represented several times, while others were not represented at all. A systematic investigation of the *a*-value space in combination with each individual BS run is computationally impractical and was therefore replaced by random initialization of the *a*-value of the secondary sulfate, sea salt and fireworks-I factor profiles between 0 and 0.5 with an increment of 0.1 for 1000 BS runs. Moreover, to avoid rejection of many solutions due to mixing

of the sea salt factor time series and the secondary sulfate time series with the fireworks factor peaks, both the sea salt and secondary factor time series (for the fireworks period only) were also constrained with an *a*-value 0.01. The small *a*-value (0.01) for the sea salt and the secondary sulfate factor time series (for the fireworks period only) were estimated based on sensitivity analyses on the *a*-value from 0 to 0.1 with an increment of 0.01. The time series of both factors were showing

fireworks peaks during the fireworks period for $a$-values greater than 0.01. Solutions were selected and retained based on the correlation (correlation of coefficient Pearson $r$) of the time series between the factors of the base case and the factors of the BS runs. Solutions with low correlation and some solutions with high correlation have a factor of completely different type, i.e., mixed or split or otherwise altered factor profile/time series based on visual inspection. These kinds of solutions were

rejected. This approach was used only for uncertainty assessment rather than uncertainty exploration to find the environmentally reasonable solution.

We also performed separate random bootstrap analyses for 1000 times on the correlation ($r$) between the time series of a base case factor and the respective external marker, e.g., the secondary sulfate factor vs ACSM sulfate, and the non-exhaust traffic-related factor vs NO$_x$ to assess the acceptable uncertainty of the $r$ value. The resulting correlation coefficients were represented

in probability density functions (PDF) over 1000 bootstrap runs for both bootstrap analysis methods. In total 86 % of the bootstrap runs were classified as environmentally good solutions. The average $a$-value retained by the selected bootstrap runs was 0.233, 0.255 and 0.241 for the fireworks-I, sea salt and secondary sulfate factor profiles, respectively. The spread of the $a$-value for these three factors is presented as mean, median and interquartile in the supplement (Fig. S11). The selected solutions factor profiles are represented as box whisker plot in the sequence of p10 (10th percentile), p25 (25th percentile), p50

or median (50th percentile), p75 (75th percentile) and p90 (90th percentile) in Fig. 2.

During the non-fireworks period, uncertainties in the source apportionment results are assessed by a bootstrap analysis as described above. However, this approach cannot be used to assess uncertainties in the sea salt and secondary sulfate factors during the fireworks period, as during this period these factor time series are constrained with an artificially low $a$-value selected to optimize deconvolution. For these two factors, uncertainties during the fireworks period are determined by our

ability to accurately predict the factor time series. The secondary sulfate and sea salt factors cases are discussed separately below.

Secondary sulfate concentrations during the fireworks period were estimated from the linear fit of the secondary sulfate factor to ACSM sulfate during the entire non-fireworks period. Uncertainties of ±5 % were calculated as the standard deviation of the actual secondary sulfate concentrations to the predicted values and included for the fireworks period only in Fig. 3a and

Fig. S12.

As described above, the sea salt factor time series during the four-day fireworks period was investigated to determine measurements that were affected or not affected by fireworks, where the measurements determined to be affected were replaced with a linear interpolation between the nearest good points. To determine the uncertainties of this approach, we applied this calculation to random segments of the non-fireworks data. Specifically, the four day-long sequence of affected/non-affected

time points determined during the fireworks period was applied to a randomly chosen segment of data, and the standard deviation of measurement data to the estimated values calculated by interpolation was determined. This analysis was repeated for 38 randomly selected locations through the non-fireworks data, and a mean standard deviation of ±42 % was determined. This value is used as the uncertainty of the sea salt factor time series (during the fireworks period only) in Fig. 3a and Fig. S12.

## 4 Results and discussion

### 4.1 Overview of retrieved factors

The solution that best represented the input data was the eight-factor solution. The eight factors from the PMF results are as follows:

1. Two fireworks factors with prominent relative contributions of Bi, Ba, K, S, Ti, Cu and Cl, which are important components of fireworks (Kong et al., 2015; Vecchi et al., 2008);
2. A sea salt factor explaining a large fraction of Cl in the $PM_{10}$ fraction;
3. A secondary sulfate factor mostly dominated by S and highly correlated with ACSM sulfate; (Fig. S1);
4. Two dust factors, one dominated by Ca and showing traffic rush hours peaks, and the other dominated by Si without
a clear diurnal pattern;
5. A non-exhaust traffic-related factor characterized by Fe, Cr, Cu, Mn, Zn and Ba;
6. An industrial factor showing relatively high contributions of Pb and Zn.

### 4.2 Detailed factor description

In this section, the results of the $PM_{10el}$ mass driven by the sum of 14 elements are presented and validated. Fig. 2 represents
the fractional composition of the factor profile ($f_{kj}/\sum_j f_{kj}$) (left y-axis; colored box whisker plots for each factor) and relative contribution ($\sum_i g_{ik} f_{kj}/\sum_k \sum_i g_{ik} f_{kj}$) of each factor to each variable (right y-axis; black box whisker plots). Fig. 3a shows the time series of the factors contributions in ng m$^{-3}$ (bottom panels) and of the relative contributions (top panel) of the retrieved $PM_{10el}$ factors. The variability of these time series across all good solutions was relatively low. Fig. 3b reports the averaged total $PM_{10el}$ mass and relative contributions of the $PM_{10el}$ sources. The reported variabilities/uncertainties (which correspond
to the interquartile range among selected bootstrap runs) are an indication of the high stability of the solution. The diurnal variations of the absolute concentrations of the identified factors and some of their corresponding external tracers are presented in Fig. 4. CBPF analysis was performed at 90[th] percentile (Fig. 5) as well as at different percentile ranges (Fig. S13) to validate some of the identified sources and their characterization.

**Fireworks:** The fireworks factor profiles and time series are shown in Fig. 2 and Fig. 3a respectively. The fireworks factors
are mostly dominated by K, S, Cl, Ti, Cu, Ba and Bi, which are the chemical elemental species of fireworks (Moreno et al., 2007; Wang et al., 2007; Vecchi et al., 2008; Perrino et al., 2011; Tian et al., 2014; Kong et al., 2015; Lin, 2016; Pongpiachan et al., 2018). Ba and Cu compounds are used to produce green and blue fireworks. The presence of Cl in fireworks-I suggests that the chloride salt might be the main chemical form in the fireworks, such as barium chloride. K is one of the major components of fireworks, which contain 74 % of $KNO_3$ in black powder as the oxidizing agent for the burning process
(Drewnick et al., 2006). The ACSM inorganics concentrations during the fireworks episodes indicate that neither particulate nitrate nor ammonium is generated in the fireworks in significant amounts (Fig. S14). Consistent with previous measurements

of submicron fireworks aerosol (Drewnick et al., 2006; Vecchi et al., 2008; Jiang et al., 2015), nitrate was not enhanced during the fireworks period, suggesting conversion of $KNO_3$ to other forms of nitrogen. The $NO_2$ / K mass ratio (1.66) on the main fireworks hour (1 August 23:00 LT) (see Fig. S14) is close to the molecular / atomic ratio of $NO_2$ / K (1.17). This measured ratio is also in agreement with the $NO_2$ / $K^+$ (2.03) ratio observed during Chinese Spring Festival in Shanghai (Yao et al., 2019). However, the $NO_2$ and/or $NO_x$ variation was not significant during the fireworks peaks in the present study (Fig. S13), which is in agreement with the former studies (Vecchi et al., 2008; Retama et al., 2019; Yao et al., 2019). The K / S ratio of 2.72 in the fireworks-I factor profile is in good agreement with the K / S elemental concentration ratio (~2.76) in black powder (Dutcher et al., 1999). Other K compounds in black powder can be in the form of perchlorate or chlorate (Wang et al., 2007). Bi is used in crackling stars (Dragon's eggs) in the form of bismuth trioxide or subcarbonate as a non-toxic substitute for toxic lead compounds (Perrino et al., 2011). The Ba / K ratio of 0.031-0.054 for fireworks-I and fireworks-II is close to the value 0.057 reported in Pongpiachan et al. (2018). Fireworks particles are usually present in large amounts in the fine fraction, staying longer in the atmosphere (Richard et al., 2011; Moreno et al., 2007). A pronounced increase in both fireworks factor time series is observed when the fireworks traditionally begin. The diurnal patterns of the fireworks-related elements exhibit a peak at 23:00 LT during the fireworks period (Furger et al., 2017) in accordance with the fireworks-I diurnal variation (Fig. 4c). Both fireworks factors contain two sharp peaks. The fireworks-I factor concentration started to increase on 31 July 2015 22:00 LT and formed the extreme peak within 1 h at 23:00 LT (~5 µg m$^{-3}$). After that it decayed quickly (10 times lower concentration than maximum fireworks concentration) within 1 h and remained more or less constant until 1 August 2015 21:00 LT. It again started to increase from 22:00 LT and formed a second sharp peak on 1 August 2015 23:00 LT, followed by a gradual decay over the next 6 to 10 h. Fireworks-II presents a slightly different pattern in its time series. It started to increase from 31 July 2015 22:00 LT and depicted its highest peak at 00:00 LT (~3.6 µg m$^{-3}$) with a quite slow decay until 1 August 2015 06:00 LT. The concentration remained slightly higher than fireworks-I over daytime. It then started to increase again from 1 August 2015 17:00 LT and yielded the highest peak on 2 August 04:00 LT with 6.7 µg m$^{-3}$. It remained higher until 2 August 2015 08:00 LT and slowly decayed until the afternoon, followed by further prominent peaks at 23:00 LT on 2 August and 3 August. Chemical reactions of KCl with $H_2SO_4$ will result in a release of gaseous HCl and may explain the absence of particulate Cl in the fireworks-II factor profile. The time series variations in both fireworks suggest that fireworks-I might be related to the main fireworks celebration while fireworks-II might result from burning of leftover crackers after the main fireworks day, as well as the influence of other sources such as bonfires, which are a common activity during Swiss National Day celebrations. Another possibility could be the advection of fireworks clouds from nearby cities where grand firework displays and bonfires are carried out at large scale to celebrate the Swiss National Day. The average relative contributions of fireworks-I and fireworks-II to the analysed mass were 7.4 % and 11 % respectively (Fig. 3b). Fig. S15 represents an estimate of overall fireworks composition and temporal variability, complementing the PMF results. A K concentration > 220 ng m$^{-3}$ was used as the criterion to separate fireworks data points (65 data points) from the whole data set. The detailed description is mentioned in the supplement. Most of the elements are well captured by two fireworks factors, such as S, K, Fe, Cu, Ba etc., since they lie within the fireworks data distribution. In contrast, there are some elements that are not

contributing to fireworks, e.g., Ca and Cr, while some elements are explained by only one firework, e.g., Si, Cl and Pb. This indicates that a single fireworks factor is not enough to represent the fireworks data variability.

**Sea salt:** The sea salt factor was mainly composed of Cl (81 % of the factor mass) as shown in the factor profile (Fig. 2), with no diurnal pattern (Fig. 4e). The average relative contribution of the sea salt factor to the analysed mass was 3.7 % (Fig. 3b).

Sea salt also includes Na and Mg, which were not measured by the Xact but analysed by ICP-OES for 24-h offline filters. Based on the high correlation of Na and Cl in a previous study, Cl alone can be used as a marker for sea salt particles (Vallius, 2005). The existence of sea salt particles was confirmed by a low Mg / Na ratio in the filter data, equalling 0.13 and 0.16 for 28 July and 30 July, respectively, in line with a ratio of 0.132 to 0.185 for marine aerosol (Chesselet et al., 1972). For the remaining 8 filter samples Mg / Na was higher than 0.18, probably due to absence of sea salt (Fig. S16). A comparison with

ACSM data revealed that Cl was mainly present in the fine fraction ($PM_1$) during firework days, while during the rest of the campaign it was in $PM_{10}$ (Fig. S17). The measured Cl / Ca ratio (0.33) from Xact does not lie in the range of de-icing salt composition (5.27) measured in northern Germany (Pernigotti et al., 2016), which validates our interpretation as a sea salt factor. In addition, the highest Cl concentrations were observed only in the last week of July, with westerly winds at higher wind speed (5–8 m s$^{-1}$) (Figs. S17, S18). This result is in agreement with the CBPF plot where the high concentration of the

sea salt factor dominates for westerly winds with high wind speed (Figs. 5, S13), and confirms previous studies (Visser et al., 2015; Twigg et al., 2015).

**Dust:** The PMF analysis resolved two dust-related factors, i.e., a road dust (Ca-rich) and a background dust (Si-rich) factor, with average relative contributions to the analysed mass of 18 % and 16 %, respectively (Fig. 3b). The road dust factor was mainly composed of Ca (68 % of the factor mass) followed by Si (26 %), with relative contributions of 89 % to Ca, 19 % to

20 Si and 12 % to Mn, while the background dust factor highly contributed to Ti, Si, Mn, Fe and Ca, with 65 %, 58 %, 22 %, 16 % and 6 %, respectively. The two dust factors together explain 95 % of Ca, while the remaining factors explain only 5% of Ca. The solution with two dust factors resulted in reduced scaled residuals for Si and Ca compared to a solution with one dust factor (Fig. S19). The scatter plot of the absolute concentrations of Si and Ca also indicates the presence of two different sources (Fig. S20). The high relative contribution of Ca in road dust has also been found in other source apportionment studies

(Ducret-Stich et al., 2013; Bukowiecki et al., 2010). In general, Ca and Si are commonly associated with mineral dust, construction activities, vehicular emissions and iron/steel plants (Lee and Pacyna, 1999; Vega et al., 2001; Bukowiecki et al., 2010; Crilley et al., 2016; Maenhaut, 2017). Iron/steel plants produce furnace slacks, a glass-like by-product which consists of Ca, Si, Mg and Al oxides. The higher fraction of crustal elements such as Ca and Si in road dust might be a consequence of the widespread use of asphalt/concrete to make roads (Fullova et al., 2017; Li et al., 2004). The sampling site is located close

to the freeway and must be influenced by wear and tear of the asphalt/concrete roads because of heavy traffic. Ca has been associated with construction activities in previous studies (Bernardoni et al., 2011; Crilley et al, 2016), which have been found to peak during the day and decreased to almost zero outside of normal working times (08:00 until 17:00 LT). This evidence is

not supported by the road/background dust factor diurnal pattern in this study. Further evidence of non-construction activity is found in the Xact elemental ratio of Ca / S (0.62) which is not in the agreement with the pure gypsum Ca / S ratio (1.25) used for construction work (Hassan et al., 2014), where the two elements are the main constituents. Road dust profiles are often difficult to identify due to resuspension of materials deposited on the road surface such as mineral dust, vehicle wear/tear,

and/or road surface wear/tear. However, the road dust factor profile is distinct from the non-exhaust traffic-related factor profile. It is possible that the road dust factor is related to resuspension of mineral dust and road wear/tear particles. The high contribution of Ca alone (> 80%) has been seen in previous studies (Ducret-Stich et al., 2013; Bukowiecki et al., 2010; Hueglin et al., 2005) where it was named as "road dust". Another study at a rural site in Switzerland also found the Ca rich factor in the coarse fraction where it was linked to soil resuspension (Minguillon et al., 2012). The background dust factor profile

exhibits elements associated with mineral dust, such as Ti, Si and Fe. These and other terrestrial elements are commonly present in soil as oxides (Rudnick and Gao, 2003). The background dust factor also contains a significant fraction of the measured Mn, which is one of the most abundant compounds in the earth's crust, where it occurs in the form of $MnO_2$ (Taylor and McLennan, 1995). Since the sampling site is adjacent to agricultural fields in the north and west directions, the contribution of these elements to this factor can be expected. A similar background factor with high contributions of Si, Ti and Ca was

found by Richard et al. (2011) at an urban site in Switzerland.

The separation of two dust factors is in line with Amato et al. (2009), where ME-2 yielded a road dust factor distinct from a mineral dust factor. The CBPF plot shows that higher concentrations of the road dust factor are associated with the southern wind sector, while the background dust factor is influenced by the southwest and northeast wind sectors (Figs. 5, S12). The diurnal pattern of the road dust factor shows morning rush hour traffic peaks similar to $NO_x$, BC, HDV count and non-exhaust

traffic-related factor (Figs. 4a, 4b) indicating resuspension of road dust is due to the vehicle fleet. A similar relationship between road dust and the non-exhaust traffic-related source was observed in a previous study (Amato et al., 2009). Resuspension of road dust in the early morning traffic is not triggered by wind speed (Fig. S21), but by the traffic fleet, whereas in the afternoon, an increase in wind speed leads to resuspension of dust deposited on the road as well as resuspension of agricultural soil dust near the sampling site. Therefore, meteorology plays a vital role for the contribution of the background

dust factor.

**Non-exhaust traffic-related:** The non-exhaust traffic-related factor profile was mostly dominated by Fe (73 % of the factor mass) and contributed strongly to Cr (96 %), Fe (76 %), Cu (71 %), Mn (50 %), Zn (31 %), Ba (26 %) and Si (12 %). Its average relative contribution to the analysed mass was 20 % (Fig. 3b). Coarse particles from brake/disc wear could appear as flakes and mainly consist of iron oxides (Wahlström et al., 2010). The higher fraction of Fe in the non-exhaust traffic-related

source has been found in several previous studies (Visser et al., 2015; Amato et al., 2014a; Bukowiecki et al., 2010; Dall'Osto et al., 2013; Crilley et al., 2016). However, the ratio of Fe to other elements is variable between studies. Fe, Cr, Cu, Zn, Mn and Ba are the most abundant trace elements in brake pads, brake lining and thus attributed to tire/brake wear (Thorpe and Harrison, 2008; Grigoratos and Martini, 2015; Gianini et al., 2012) and engine wear. Si is one of the brake lining components

used as abrasive to increase friction, and as fillers to reduce manufacturing costs, respectively (Thorpe and Harrison, 2008; Grigoratos and Martini, 2015). This factor is characterized by a strong diurnal peak coinciding with the morning rush hour at 08:00 LT, similar to $NO_x$ and BC (Fig. 4b). The LDV and HDV counts start to increase from 05:00 LT (Fig. 4a), unlike the primary traffic emission $NO_x$ and BC. The similar diurnal pattern of this factor might be due to the high braking load for vehicles during peak traffic hours, resulting in increased emissions of vehicle wear particles.

**Secondary sulfate:** This source is characterized by sulfur (S) and is most likely due to the regional background contribution of secondary sulfate due to conversion of $SO_2$ to $SO_4^{2-}$, consistent with the results of many previous source apportionment studies (Dall'Osto et al., 2013; Richard et al., 2011). It explains the highest fraction of S (91 % of the factor mass) with relative contributions to S (73%) and Pb (30%) (Fig. 2). The average relative contribution of this factor to the analysed mass is 21 % (Fig. 3b). Similar factor profiles were found in previous studies (Visser et al., 2015; Dall'Osto et al., 2013). This factor correlates well with ACSM $SO_4^{2-}$ measurements ($r^2 = 0.91$), suggesting a dominant contribution from the submicron fraction and thus a slow rate of dry deposition. Combined with $SO_2$ oxidation processes occurring on timescales of hours to days, it is thus reasonable that the secondary sulfate factor does not exhibit a clear diurnal variation (Fig. 4f) and is consistent with regional rather than local sources. The time series of secondary sulfate exhibits peaks during the fireworks event (Fig. 3a), in agreement with the ACSM $SO_4^{2-}$ time series (Fig. S22). Inorganic gases ($SO_2$, $NO_x$, etc.) emitted during the fireworks events may be oxidized to secondary organic and inorganic components that may condense to the particle phase (Sarkar et al., 2010) within a very short span of time, as observed in previous studies (Wang et al., 2007; Yang et al., 2014). We monitored two parameters to explain the influence of fireworks in the secondary sulfate time series: (1) mass balance for ACSM $PM_1$ data; (2) secondary sulfate peaks during the main fireworks hours (1 August 23:00 LT to midnight). The equivalent concentration of ammonium ($NH_{4eq}$) balances the sum of $NO_{3eq}$ and $SO_{4eq}$ during non-fireworks periods, while during fireworks peaks, the balance shifts towards the sum of $NO_{3eq}$ and $SO_{4eq}$ (Fig. S23). This indicates that sulfate related to fireworks adds up to an acidic budget of particles. The excess sulfate observed from this analysis is approximately in quantitative agreement (within 20%) with the enhancement of the secondary sulfate during the fireworks period. The peak in the secondary sulfate observed at 23:00 LT 1 August is slightly offset from the main fireworks peak because the secondary sulfate formed maximum peaks 1 h later (00:00 LT 2 August). Taking these two together suggests that we may have some downstream production of sulphuric acid and conversion to ammonium sulfate, which makes the secondary sulfate factor time series slightly delayed relative to main fireworks plume. Because this delay is consistent with chemical processing, we consider inclusion of this temporal feature in the secondary sulfate factor to be reasonable, although we cannot completely rule out some degree of mathematical mixing with the direct fireworks emissions. The time series of ACSM $SO_4^{2-}$ and ACSM $NH_4^+$ show significant correlation (Fig. S22), indicating the formation of ammonium sulfate particles except for the main fireworks peaks.

**Industrial:** This factor is characterised by high relative contributions to Zn (50 %) and Pb (63 %), with a low contribution to the analysed mass (in the order of 3 %, Fig. 3b). The profile is shown in Fig. 2 and the time series in Fig. 3a. The time series

contains a few spikes after 4 August, when the wind was predominantly from the south and south-east sector, suggesting an industrial emission. A similar factor profile was observed in previous source apportionment studies (Crilley et al., 2016; Richard et al., 2011) from local point source emissions without any link to specific industrial activities. Industrial emissions play a minor role in this study area, as it is surrounded by only a few small scale industrial buildings (logistics businesses

approximately 500 m to the north-west, and a wheel manufacturing company to the south-east across the freeway). The CPBF plot confirms the dominance of high concentrations in the south-east direction (Fig. 5). The diurnal cycle shows a clear peak at 10:00 LT (Fig. 4g) which is related to a few peaks.

## 5 Conclusion

A source apportionment study of 14 elements in $PM_{10}$ measured at a traffic-influenced site in Härkingen, Switzerland, during

the summer of 2015 was conducted using the ME-2 implementation of PMF. The PMF model was able to resolve and evaluate the contributions and compositions of eight sources: two fireworks factors, sea salt, secondary sulfate, background dust, road dust, non-exhaust traffic-related, and an industrial source. The use of ME-2 allowed the use of constraints via the *a*-value approach, which improved the factor resolution relative to conventional PMF. We show that the rotational control available in ME-2 provides a means for treating extreme events such as fireworks within a PMF analysis. This was only achievable when

controlling problematic factors, i.e., factors that tend to mix within the constraining technique. Two dust factors with different time profiles and two fireworks factors were identified by the PMF model resulting in better representation of the data variability. A S-rich (secondary sulfate) factor, which can typically be attributed to regional background/transported secondary sulfate, was correlated with fine mode non-refractory sulfate measured by an ACSM. The non-exhaust traffic-related factor followed the diurnal pattern of traffic rush hours similar to $NO_x$ and BC with concentrations up to 4 times higher during

daytime relative to night-time. The outcome of this study emphasizes the significant influence of regional background secondary sulfate and local background dust apart from non-exhaust traffic emissions at the sampling location. The small contribution of the industrial factor confirms the low influence of local daily activities from the surroundings.

The source apportionment model performance could possibly be additionally improved by the inclusion of Na and Sr to better resolve the sea salt and fireworks factors, respectively. It was shown that high time resolution elements data sets enable a fully

resolved SA, with considerable improvements compared to 24-h filter analysis, where the attribution to specific sources is possible only on a larger time scale and is mostly based on seasonal variations.

## Data availability

The datasets are available upon request to the corresponding author.

## Author contribution

PR performed SA analysis and wrote the manuscript. MF, RF and CH performed the measurement. MF and RF analysed the data for Xact and ACSM, respectively. FC and JGS provided expertise on software for SA analysis. MCM provided ICP-MS and ICP-AES analysis data. KP lent Xact® 625 for measurement. UB, ASHP, MF, FC and JGS were involved with the supervision and conceptualisation. All authors commented on the paper and assisted in the interpretation of the results.

## Competing interests

Krag Petterson is employed by Cooper Environmental Services, the manufacturer of the Xact® 625.

## Acknowledgements

This study has been funded by the Swiss National Science Foundation (SNSF grants 200021_162448/1 and BSSGI0_155846), and by the Swiss Federal Office for the Environment (FOEN). We thank René Richter and Roland Scheidegger of PSI for their technical support during the measurement campaign. We are grateful to Chris Koch and Varun Yadav of Cooper Environmental Services for instructions on the instrument and numerous technical discussions. We thank the operators of the NABEL station for providing all kinds of support during the measurements.

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

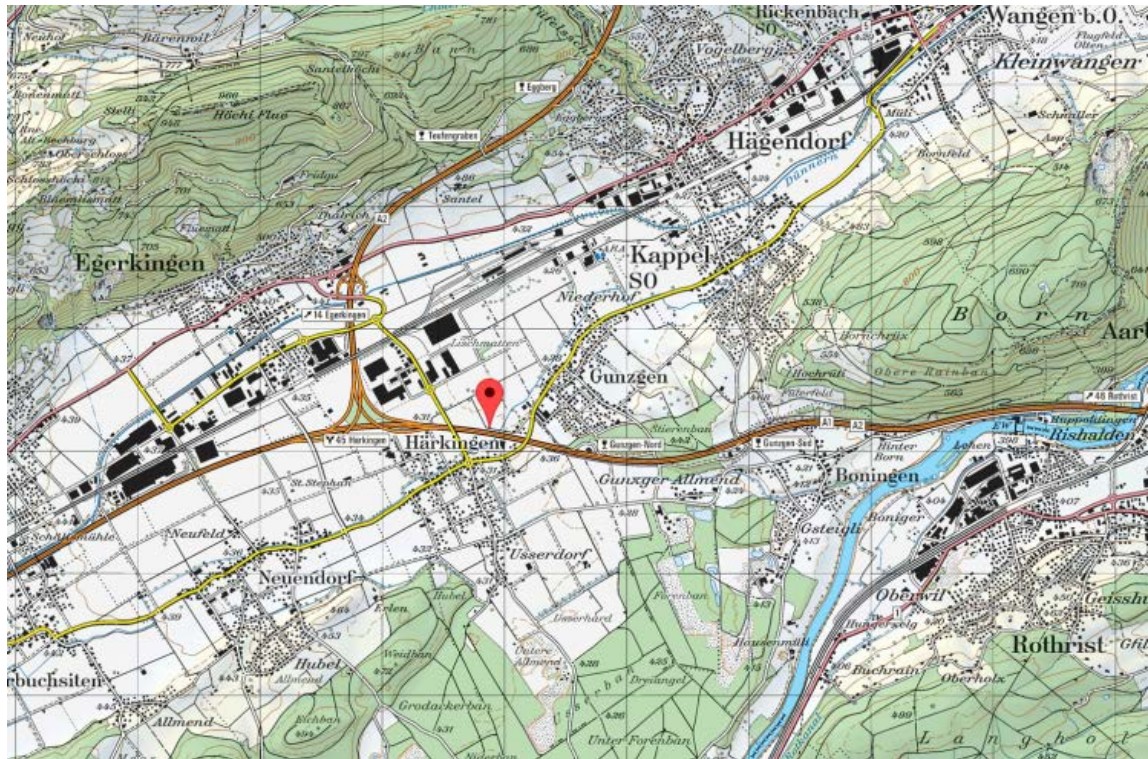

**Figure 1: Map of the sampling location (NABEL site in Haerkingen), The site is marked with red google pin. Map reproduced by permission of swisstopo (JA100119).**

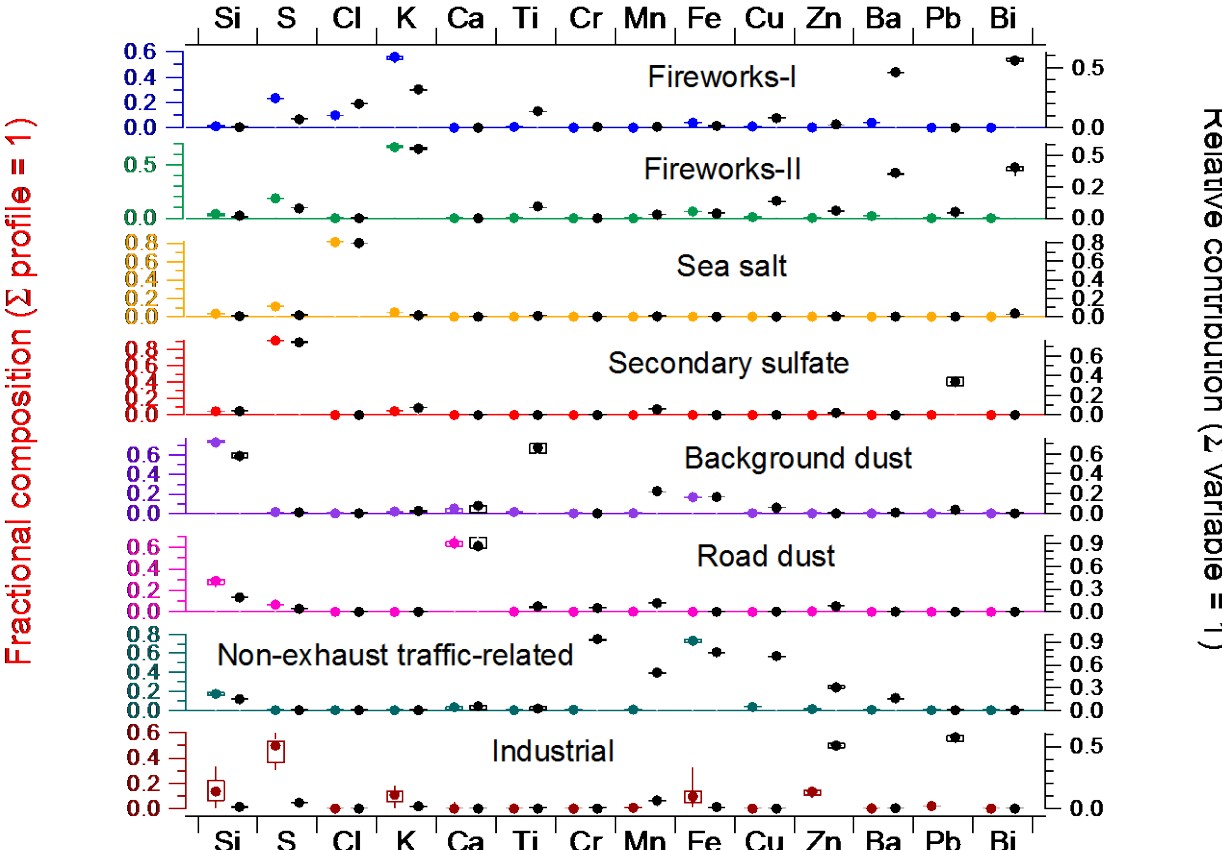

**Figure 2: Source profiles of the finally retained eight-factor PMF results. The data and their corresponding uncertainty are given as box-whisker plot (bottom to top: p10-p25-p50-p75-p90) of good solutions from bootstrap runs. The left y-axis represents the fractional composition of the factor profile in row-wise (presented in coloured box whisker plot) for each factor in ng ng$^{-1}$, the right y-axis represents the relative contribution of each factor to each variable (indicated in black box-whisker plot).**

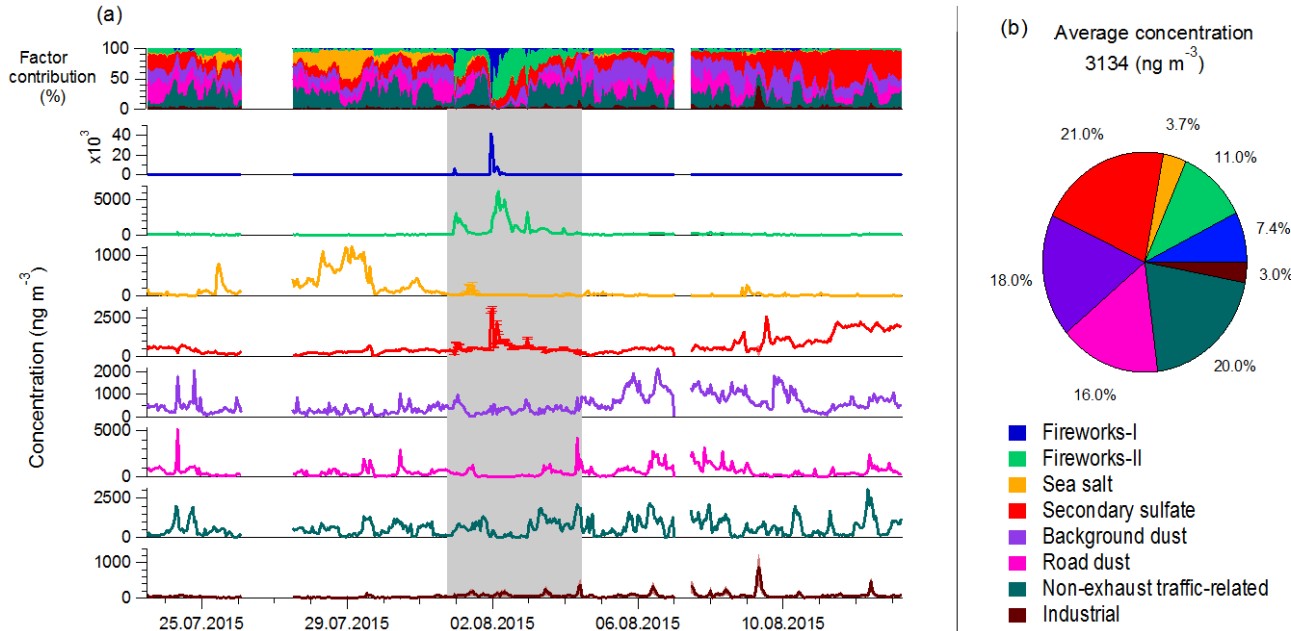

**Figure 3: (a)** Time series of the PM$_{10el}$ sources and relative contributions of the different sources over time; shaded areas indicate the uncertainties (interquartiles) of selected bootstrap runs; grey background color represents the fireworks period; the estimated uncertainties of the secondary sulfate (± 5%) and the sea salt factors (± 42%) during the fireworks period are added as error bars (magnified version is shown in Fig. S12); **(b)** Mean relative contributions of PM$_{10el}$ sources. The average concentration represents the mean value of apportioned sources in PM$_{10el}$ which is the sum of 14 elements.

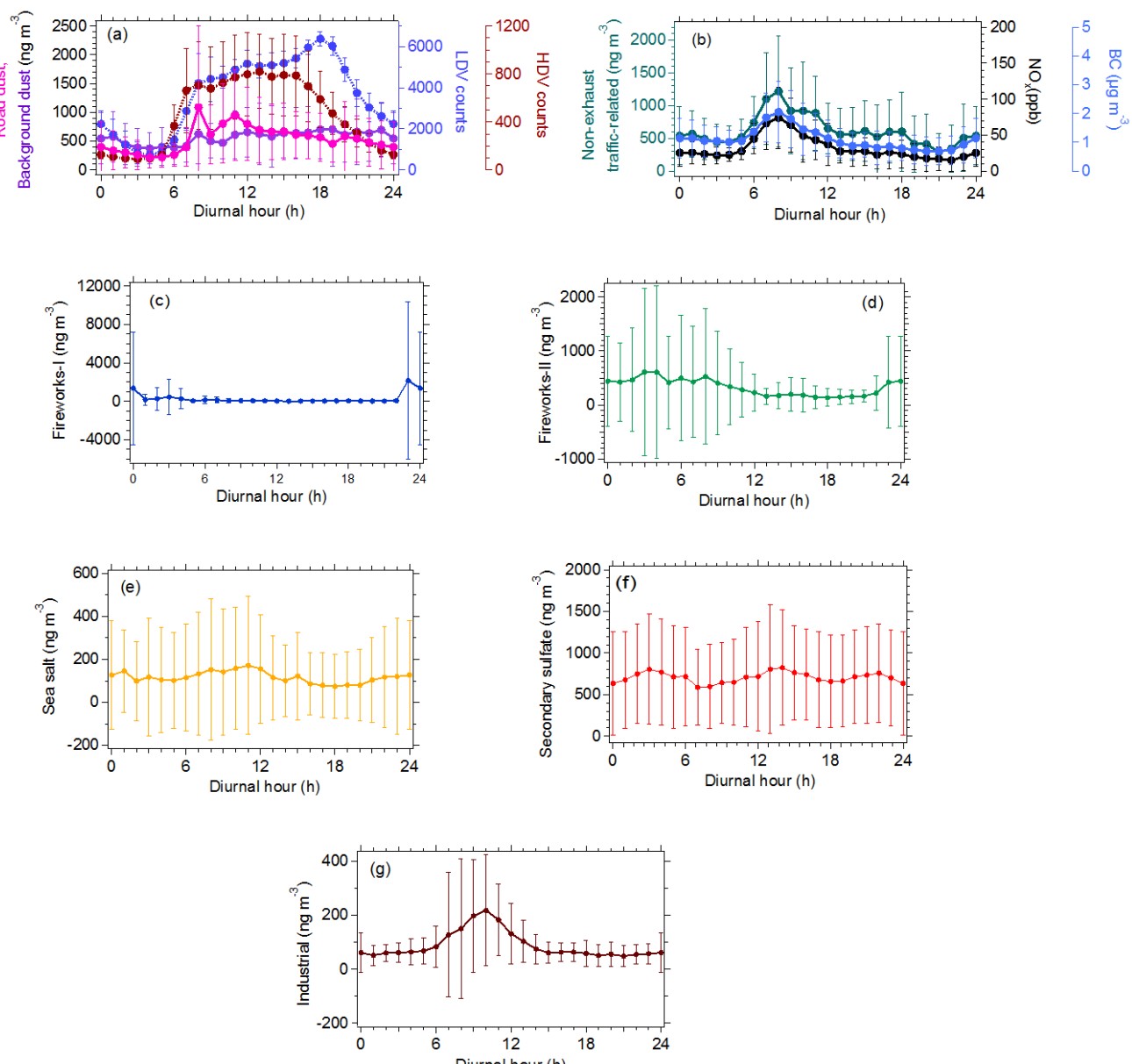

**Figure 4: Mean diurnal patterns of the factors and of some corresponding external tracers with error bars (one standard deviation).**

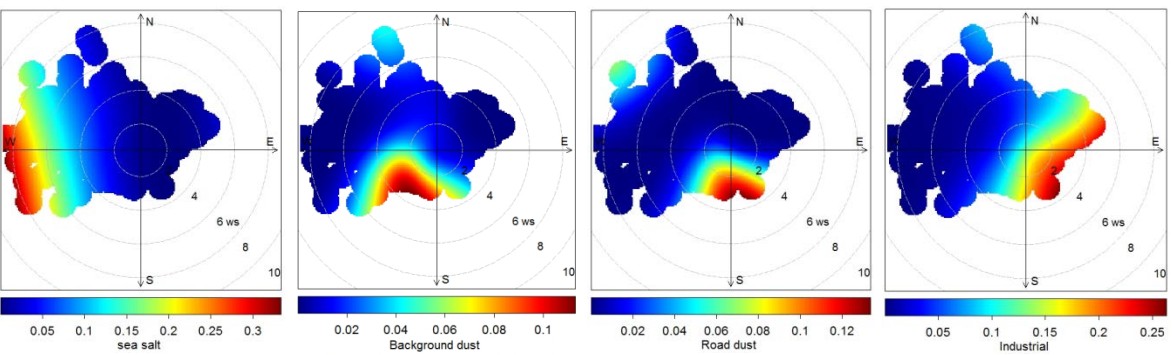

**Figure 5: CBPF analysis (at 90$^{th}$ percentile) of factors in terms of wind speed (m s$^{-1}$) and wind direction. The color code represents the probability of the factor contribution.**