# Peer review of "Source apportionment of highly time resolved elements during a firework episode from a rural freeway site in Switzerland"

_Atmospheric Chemistry and Physics, 2018_

## Referee Comment (RC1) · Anonymous Referee #2 · 12 Mar 2019

GENERAL COMMENTS

The manuscript describes a study to determine the sources impacting on PM10 in a traffic oriented site using a high resolution elemental dataset and processing data with the SoFi tool based on ME-2 receptor model algorithm. The manuscript is well organized, the language is clear and the English style is appropriate.

According to the authors, the SA study was conducted to characterize the source emissions at a traffic-influenced site. This is confusing because receptor models are normally carried out at the receptor site to characterize the source contributions rather than the source emissions. The authors claim the novelty of the work is due to the

limited number of studies for trace element emission sources with high time resolution. However, as the same authors state, many studies were carried out using rotating drum and streakers that, despite being off-line, provide comparable information that can be processed in the same manner as this study.

SA of the entire PM based only on trace elements representing only a limited fraction of the PM mass (in this case 20%) is very uncertain because a considerable amount of information is missing about the main components of PM: elemental carbon, ions and the organic fraction. The risk is to overestimate the contribution of sources for which the elements are markers and neglect sources where elements are less relevant in particular the secondary organic fraction. In addition, the identification of factors into sources is hindered because the chemical profiles contain few diagnostic species or combination of species.

The authors claim MAAP and Q-ACSM were collected in parallel so it is not clear why they didn't use this information to obtain more robust SA results. The impression is that the authors are splitting the study in pieces to make more publications.

The study location is not well described. A map is needed to help the reader understanding the characteristics of the site.

The Conditional Bivariate Polar Function (CBPF) analysis is an important part of this work, there is a dedicated sub-chapter in the methodology section. However, there are no graphs about the results of this technique in the paper. The supplementary material is supposed to be used to provide information that supports the main findings of the work not to place essential results of the work.

The conclusion that the study result emphasizes the large influence of highway traffic on the composition of PM10 is trivial considering the study was conducted at a traffic site. In addition, this is not in line with the declared objective of characterizing the emissions.

[Figure]

In this manuscript there is a contrast between the highly sophisticated data treatment and the limited data input. The model allocates all the PM10 mass on the basis on the predictive capabilities of only 14 elements which do not provide enough information to properly identify all the factors and to allocate the PM10 mass to each of them reliably. This is due to lack of markers to allocate the carbonaceous fraction and part of the secondary inorganic (nitrate). As a consequence there are not well resolved factors (e.g. the two fireworks and the two dust factors and the road dust with the traffic related factor).

Source apportionment with receptor models is a routine technique and alone does not represent a scientific novelty. In addition, the application of these models with only trace elements does not fulfill the minimum requirements to obtain robust PM source apportionment because of the absence of important information to allocate the entire PM10 mass.

SPECIFIC COMMENTS

Page 1 line 29: the literature about the health impacts of PM need to be improved.

Page 3 line 6: the study by Park et al was carried out with XACT 620.

Page 6 line 2: is it reasonable to use 10% analytical uncertainty for all the species?

Page 7 line 18: how do you know salt is from sea and not road salt?

Page 7 line 25: the expression factor composition is not clear. Do you mean factor profile?

Page 9 line 14: the comparison between the on-line and off-line data is not clear. In the text are mentioned 28th and 30th July but the Mg/Na ratio is similar to marine aerosol also on 31st July.

Page 9 line 21 and foll.: There is not clear evidence to prove there are two different dust factors. The diurnal profiles are almost overlapping (Figure 3) and in the scatter

plot of fig S13 there is only one cloud of points. Also the CBPF plots in figure S8 show overlapping wind speed and directions. Soil dust should be farther from the origin than the more local road dust. Split of sources is a typical problem of using only elements in factor analysis source apportionment. In addition, how do you know that Ca rich is road dust and not other sources? Enriched Ca dust is common in construction works, for instance.

Page 9 line 25: only the Ca scaled residuals improve not those of Si.

Page 10 line 17 and foll.: The chemical composition of this factor indicates brake wear mixed to the re-suspended particles as Fe and Si are components of crustal material. Again there is an overlapping in the chemical composition with the so called road dust profile and these two profiles also have very similar diurnal profiles.

Page 10 line 29: the secondary sulfate does not provide any evidence to allocate the ammonium nitrate (another main secondary inorganic component of PM).

Page 11 line 8: The characterization of the industrial source is weak. There is no connection to any known industrial process. The fact that other studies in completely different areas found the same profile is not proving the industrial origin of this factor. On the contrary, it is unlikely that the same industrial process is present in many different sites.

Figure 4: Not clear what is the added value of this figure with respect to figure 1. Normalized concentration (should be explained in the caption). Why use different units than the other graphs. The the percentile box and whisker and the mean -+SD overlaps making the graph difficult to read. Choose one.

Figure S9: the connection between this data and the days with high "sea salt" is not clear in this figure. It should be better evidenced to allow the reader evaluating if there is a difference between the days with high sea salt and those without it.

TECHNICAL CORRECTIONS

[Figure]

Figure S6: not clear why the p25 is a single point while the p75 is a vertical line.

[Figure]

---

## Referee Comment (RC2) · Anonymous Referee #1 · 23 Apr 2019

This paper examines data previously reported by Furger et al (2017) applying PMF source apportionment techniques to a subset of the reported data to provide insight into the chemical composition of different sources. It focuses on firework / burning sources associated with local celebrations (which are a common influence on air quality for short periods in many countries) and to a lesser extent on suspended dust from road and surface sources. The article is generally well written and the approach is methodologically robust. However, the data used is somewhat limited and the authors have not taken advantage of the aerosol mass spectrometer data which was available. I have many suggested edits and some points which require consideration by the authors. Major Comments – Pg 5, line 16 – the authors reduce the data set from 24 to

14 based on manufacturer supplied MDLs to produce 'better' source apportionment results. Firstly, even though the measurements are below the detection limit, they could contain 'valuable' information on variability, especially during firework periods as some of the excluded elements are potential firework components (e.g. Cd). They have been excluded based on % below MDL at a rather arbitrary value e.g. Bi at 93% was included while Cd at 87% was not. It would be worth including this data and downweighting rather that excluding it or at least performing sensitivity test. A further 4 elements are excluded based on data quality but no detail on what constitutes data quality is given. Pg 8 Line 4 and later – The CBPF is an important analysis which is used to justify the identification of sources and the charts should not be shunted to the appendix. Further, the reliance on 90th percentile to identify the sources is flawed when considered on its own, especially when the distribution of measurements is heavily skewed (as it is with Cl). A range of CBPFs should be presented (at least in the supplementary) to support the interpretation. The authors should also consider the use of the min bin statistic to ensure that individual points are not skewing the interpretation. Finally, the conclusion that the two dust sources are from different sources (based on these plots) when their direction is broadly similar is rather weak. Pg 8 Line 9 and later – KNO3 is quoted as comprising 74% of fireworks. In which case NO3 could be a useful tracer for unburnt firework material - it was measured but is not reported. A recognition of this point, even to say that there was no correlation would help. The same is true of the other mass concentrations and m/z measured by the ACSM – were these examined or even considered for inclusion in the PMF or data analysis? Line 29 and later – The use of SoFi to separate highly correlated sources such as the two firework sources is the reason for this paper. The consideration of Cl displacement is interesting and has evidence to support it, why does it therefore come after 2 purely speculative thoughts about possible alternative sources. Promote this point and consider the speculations as alternatives. One of these - the separation of bonfires from fireworks may have been helped by considering the ACSM data. General Comment- The authors use the term 'trace' elements. I would not consider S or Fe trace elements in ambient PM. I suggest

the authors just refer to them as elements

Pg 1 Line 13 - Source Finder software, please quote version number, supplier and country Line 19 - The abstract is partly written in the present tense – 'experiences' should read 'experienced', 'concentrations are similar' should read 'concentrations were similar' Pg 2 Line 29 – clarify that you are talking about off line sampling systems Line 33 – the sampler don't specifically need those analysis techniques, they require techniques with high precision and low detection limits – such as these methods Pg 3 Line 28 – delete XACT manufacturer already provided line 4 Line 30 – with should read at Line 31 onwards – be consistent with how you name equipment – model, make, supplier, country Line 31 – quote filter manufacturer Pg 4 Line 2 – TEOM and FDMS in full then in brackets or just the abbreviation, not a mixture Line 20 – can values be applied to individual data points; this implies that individual points can be used for a-value constraining. Is this true? Certain sections in a time series can but individual points cannot be separated out in this way. Line 28 – this is not the first use of sofi in the manuscript - the reference, version number and supplier should go there Pg 5 Line 1 – 'PMF' rather than 'model' Line 4 – 'identify' rather than 'find' Line 4 – this only helps to identify the high concentration peaks and whether then influence the mean will depend on the distribution of data (see Cl results in fig 4) Line 22 – Bi a major component of fireworks – a reference, some evidence would be good here and even justify the exclusion of other elements Pg 6 Line 1 - need a description on xij Line 16 – listed s1-s4 but only s2 and s3 relate to this point Pg 7 Line 18 – how do you know it is coarse Cl, you have two measures of Cl one is non-refractory Cl- in PM1 and one is Cl in PM10? Pg 8 Line 6 – are these elements really the main components of fireworks by mass, surely you need a reference here to justify this. Line 7 – chlorate and perchlorate (line 11) will also be detected as Cl in XRF Line 16 – the sentence starting 'A pronounced...) should really come at te start of the para Line 29 – this consideration of Cl displacement is interesting and at has evidence to support it, why does it therefore come after 2 purely speculative thoughts. Promote this point and consider the speculations as alternatives. Pg 11 – line 6 – there is no fig S12

---

## Referee Comment (RC3) · Anonymous Referee #3 · 24 Apr 2019

The paper "Source apportionment of highly time resolved trace elements during a firework episode from a rural freeway site in Switzerland" by Rai et al. deals with Positive Matrix Factorization analysis of a dataset of 1-h resolved trace elements. Despite the authors declare the existence of much more information on high time resolved scale (mass concentration by TEOM, equivalent black carbon by MAAP, ACSM data), they carried out the analysis on elements only. This limits strongly the information provided by the study (e.g. they can apportion only the mass related to elements, that they estimated to be about 20% of PM10 mass). Thus, the results cannot be representative of the total contribution of the sources to the measured PM. Furthermore, lots of constraints were implemented to reach the final solution. In some cases, both source

profiles and temporal trends were constrained. So many constraints to the model make questionable the validity of the results, also considering that these constraints are not adequately supported by a methodological description of the way they were obtained, as explained in more detail below. Whether the paper is well structured and well written in most parts (even if some obscure descriptions remain), the scientific aspect is not fully convincing.

Major concerns

Page 5, line 15: excluding mass data (mentioned at p.4 l.2) should be carefully discussed as it strongly reduces the interest of the results, preventing an absolute quantification of the factor contributions to the measured PM. Furthermore, maybe the authors decided to analyse separately ACSM data, but at least equivalent BC information could be of help in source resolution.

Page 6, lines 11-12. "The unconstrained PMF solutions yielded mixed factor solutions. Therefore, it was essential to constrain specific factor profiles and the time series in the PMF analysis to avoid mixing (see details in the Supplement, section S1, Fig. S1)". This is the weakest aspect of all. The way in which the factor profiles/time series are constrained is the key point for all the rest of the analysis. Constraints determination strongly affects the final results (see described differences between the preliminary analysis and the constrained one) and merits detailed description and attention. Opposite, its description was moved to the Supplemental Material, and what is reported there is anot sufficient to determine the robustness of the approach. The following information should have been provided: 1) How many factors were present in the final analyses of the fireworks and non-fireworks periods? 2) Were the sources other than fireworks and sea-salt comparable between the datasets? (In terms of profile and tracer species?) 3) Were 2 firework-related factors identified in the analysis of the firework-days subset? 4) What about the residual of S in the unconstrained 9 factor solution? 5) The need to constrain both profile and temporal trend of the sulphate source is very suspicious (the whole "secondary sulfate" factor was constrained in the final

solution).

Pag 6, line 13: "obvious structure": completely obscure what the authors mean.

Pag 6, line 28: "both the sea salt and secondary sulfate factor time series were also constrained with a-value 0.01". Further constraint implemented. Please note that a small a-value was used. Is it consistent with uncertainty estimates for the non-firework sub-set and for the sulfate factor identified in the preliminary 9-factor solution? Please also note that the constraints come from two different analyses.

Page 7, line 19: obvious. It was checked in the preliminary analysis at 9-factors and then constrained both as far profile and temporal patterns are concerned in the final solution.

Page 8, "Fireworks" paragraph. Considering that the fireworks profile was constrained, it sounds very strange that two fireworks sources were identified in the final analysis. As previously required, the existence of two fireworks sources should be supported by their identification at least in the unconstrained analysis of the fireworks period. If not, the strength of the imposed constrain should be verified to get if it artificially generates the presence of two factors.

Page 9, lines 1-4 and Figure 4. Completely obscure. I interpret "normalized" as "divided by", but in this case how can the negative values be justifiable? Furthermore, what is "composition"? Average contribution of the factor to each element?

Page 11, lines 22-23: "We established that data sets including extreme events such as fireworks can be apportioned by ME-2 without disturbing the model solutions". Untrue. The imposed constraints completely modified the output of the unconstrained analysis.

Minor comments

Page 2, Line 12: A reference to: "Fe in brake lining can reach up to 60 % by weight" is needed

Page 2, line 27: add "among others" after "Wang et al., 2018". Indeed, the list is far from being complete (see e.g.: Li et al. 2017 http://dx.doi.org/10.1016/j.jenvman.2017.02.059; Zhou et al., 2018 https://doi.org/10.5194/acp-18-2049-2018 among the most recent). If the authors intend providing a full list, a much more detailed research has to be done.

Page 3, lines 10-12. "The later, being essential in particular when separating extreme events such as fireworks which are most often excluded from the PMF input matrix (Ducret-Stich et al., 2013; Norris et al., 2014) to avoid distortion in the PMF solution due to unusually high emissions". This sentence is questionable. As reported by Paatero et al., (doi:10.5194/amt-7-781-2014) wrong decision on the outlier status of data can introduce serious modelling errors. Please also note that opposite to what stated by the authors, examples in the literature tried to exploit fireworks tracers to quantify the source contribution, with different approaches (e.g. Scerri et al., 2018 https://doi.org/10.1016/j.chemosphere.2018.07.104, Ji et al., 2018 https://doi.org/10.1016/j.scitotenv.2018.01.304, Vecchi et al., 2008 http://dx.doi.org/10.1016/j.atmosenv.2007.10.047)

Page 7, line 28: "absolute mass". It should be recalled that it refers only to the mass related to elements, as no PM mass was inserted in the analysis.

Page 9, line 25-26: "The two dust factors together explain 95 % of Ca, with no other factor explaining more than 93 %". Obscure. If already explained at 95%, how can other factors explain Ca for more than 93%?

Page 9, lines 29-30: "In general, Ca is commonly associated with 30 mineral dust, construction activities, vehicular emissions and iron/steel plants". References are needed.

---

## Author Comment (AC1) · 5 Jul 2019

**Source apportionment of highly time resolved trace elements during a firework episode from a rural freeway site in Switzerland**

Pragati Rai[1], Markus Furger[1], Jay Slowik[1], Francesco Canonaco[1], Roman Fröhlich[1], Christoph Hüglin[2], María Cruz Minguillón[3], Krag Petterson[4], Urs Baltensperger[1] and André S.H. Prévôt[1]

[1] Laboratory of Atmospheric Chemistry, Paul Scherrer Institute, Villigen PSI, 5232, Switzerland
[2]Laboratory for Air Pollution / Environmental Technology, Empa, 8600 Dübendorf, Switzerland
[3]Institute of Environmental Assessment and Water Research (IDAEA), CSIC, 08034 Barcelona, Spain
[4]Cooper Environmental Services (CES), 9403 SW Nimbus Avenue, Beaverton, OR 97008, USA

*Correspondence to*: André S. H. Prévôt (andre.prevot@psi.ch) and Markus Furger (markus.furger@psi.ch)

**Response to Reviewer #1**

We thank Reviewer#1 for the careful revision and comments which helped improving the overall quality of the manuscript. A point-by-point answer (in regular typeset) to the reviewers' remarks (in italic typeset) follows. Changes to the manuscript are indicated in blue font.

In the following page and line references refer to the manuscript version reviewed by anonymous Reviewer#1.

*GENERAL COMMENTS*

*The manuscript describes a study to determine the sources impacting on PM10 in a traffic oriented site using a high resolution elemental dataset and processing data with the SoFi tool based on ME-2 receptor model algorithm. The manuscript is well organized, the language is clear and the English style is appropriate.*

*According to the authors, the SA study was conducted to characterize the source emissions at a traffic-influenced site. This is confusing because receptor models are normally carried out at the receptor site to characterize the source contributions rather than the source emissions. The authors claim the novelty of the work is due to the limited number of studies for trace element emission sources with high time resolution. However, as the same authors state, many studies were carried out using rotating drum and streakers that, despite being off-line, provide comparable information that can be processed in the same manner as this study.*

We agree with reviewer#1 that receptor models characterize the source contributions. Therefore we have replaced "emissions" by "contributions" to make it clear.

The novelty of this work mainly highlights the inclusion of a strong fireworks event in the PMF input data set. Typically such strong, isolated events are excluded from PMF analysis to avoid distortion of the solution. However, here we established that unmixed fireworks can be apportioned by ME-2, without disturbing the model solutions, in contrast to conventional/unconstrained PMF.

We agree with the reviewer that highly time-resolved trace element measurements can be obtained by rotating drum and streaker techniques, and in principle provide similar information as the Xact. However, these samplers require offline analysis, including accessibility and significant beam time availability at accelerator facilities. In practice, this leads to undesirable trade-offs between time resolution and data coverage even for short duration

field campaigns, whereas highly time-resolved long-term measurements are impractical. In contrast, the Xact 625 is an online system which has been developed and commercialized for rapid, long-term and semi-continuous detection of the ambient aerosol particle's elemental composition. This is a novel and highly useful analytical capability, and characterization of the instrument's measurement and source apportionment capabilities is therefore of significant interest.

We have rewritten page 3 line 1 as follows:

In practice, the offline samplers lead to undesirable trade-offs between time resolution and data coverage even for short duration field campaigns, whereas highly time-resolved long-term measurements are impractical. A recently introduced online high time resolution instrument can collect samples and perform analysis for elements simultaneously in a near real time scenario for long-term measurements without waiting for laboratory analysis.

*SA of the entire PM based only on trace elements representing only a limited fraction of the PM mass (in this case 20%) is very uncertain because a considerable amount of information is missing about the main components of PM: elemental carbon, ions and the organic fraction. The risk is to overestimate the contribution of sources for which the elements are markers and neglect sources where elements are less relevant in particular the secondary organic fraction. In addition, the identification of factors into sources is hindered because the chemical profiles contain few diagnostic species or combination of species.*

It is important to know where the inorganic elements / metals come from. Every source apportionment typically focuses on the components that are measured. For example, all the PMF of organic aerosol just does a source apportionment of the organic aerosol. We do not attempt to provide a full source apportionment of the atmospheric aerosol, but rather focus on the inorganic elements / metals. This has been done to a much lesser extent than for the organic aerosol, but it is actually important, and may become more important in the next years, due to the discussion of the possible health effects of metals.

In addition, we have used BC, $NO_x$ and Q-ACSM data to compare the SA results with the time trends of those components. We did not claim to do a SA of $PM_{10}$ just with the elements measured by the Xact. We fully agree with reviewer#1 that it would introduce a lot of uncertainties if we attempted to scale our element SA to the full $PM_{10}$, however, as mentioned, this is not the case here. We believe that SA of trace elements alone is already interesting. Some elements are directly health-relevant and many are closely related to specific sources. The total SA including elemental carbon, ions, organic fraction and trace elements is quite complicated, requiring blending of data from several instruments. It is also suboptimal for identifying sources of trace elements because it can reduce them to minor factors (in comparison to organics or non-refractory inorganics). In addition, the combined SA would require combining measurements with different size cuts, e.g. Xact with $PM_{10,}$ ACSM with $PM_1$ and BC with $PM_{2.5}$.

*The authors claim MAAP and Q-ACSM were collected in parallel so it is not clear why they didn't use this information to obtain more robust SA results. The impression is that the authors are splitting the study in pieces to make more publications.*

As already mentioned above combining different data sets from different instruments into one ME-2 model is challenging due to differences in error propagation. In our opinion, the combination of $PM_1$ data for Q-ACSM,

PM$_{2.5}$ for MAAP and PM$_{10}$ data for Xact may introduce artefacts in the ME-2 analysis. We have used MAAP and Q-ACSM data sets in the manuscript to compare some of the ME-2 factor time series and diurnal patterns. We are not planning to publish any other paper on the additional data. A major focus was the exploration of the use of the Xact for source apportionment in Europe where the concentrations are considerably lower than in polluted areas in Asia.

We have added in the manuscript as follows on page 3 line 7:

The main focus of this work is the exploration of the use of the Xact for source apportionment in Europe where the concentrations are considerably lower than in polluted areas in Asia.

*The study location is not well described. A map is needed to help the reader understanding the characteristics of the site.*

We have added a map in the manuscript on page 22.

[Figure]

Figure 1: Map of the sampling location (NABEL site in Haerkingen), The site is marked with red google pin. Map reproduced by permission of swisstopo (JA100119).

*The Conditional Bivariate Polar Function (CBPF) analysis is an important part of this work, there is a dedicated sub-chapter in the methodology section. However, there are no graphs about the results of this technique in the paper. The supplementary material is supposed to be used to provide information that supports the main findings of the work not to place essential results of the work.*

This issue was also raised by reviewer #2. We have modified the graph and moved it into the main text of the revised manuscript.

*The conclusion that the study result emphasizes the large influence of highway traffic on the composition of PM10 is trivial considering the study was conducted at a traffic site. In addition, this is not in line with the declared objective of characterizing the emissions.*

We have mentioned in the objective that we conducted SA using PMF to characterize the source contributions of highly time-resolved metals during a three-week campaign at a traffic-influenced and otherwise rural site in Härkingen, Switzerland. Therefore, we decided to mention a brief overview of results in the conclusions including the smallest and largest contributions of sources, which is in line with the objective. In fact, the high contribution

of traffic is not trivial and not easily accessible with 24-hours sampling. In addition, it is important in the light of the role elements may have concerning health effects. In contrast, the contribution of traffic is minor for organics, also at such a site and also for other components it is not clear how much the rather local source would contribute in comparison to other more regionally influenced components.

*In this manuscript there is a contrast between the highly sophisticated data treatment and the limited data input. The model allocates all the PM10 mass on the basis on the predictive capabilities of only 14 elements which do not provide enough information to properly identify all the factors and to allocate the PM10 mass to each of them reliably. This is due to lack of markers to allocate the carbonaceous fraction and part of the secondary inorganic (nitrate). As a consequence there are not well resolved factors (e.g. the two fireworks and the two dust factors and the road dust with the traffic-related factor).*

Again, we are not attempting a source apportionment of the full $PM_{10}$, but rather of the inorganic elements / metals concentrations. For these components, we have been able to retrieve quite well resolved factors, in contrast to the reviewer's opinion. Examples are the strong influence of the fireworks episode and the wind direction dependency. In addition, the high time resolution of the Ca and Si data set make the separation of two dust factors with distinct factor time series possible, while the traffic-related factor shows much stronger diurnal patterns, similar to BC and $NO_x$ as compared to the road dust factor. The BC time series yields a strong correlation with the traffic-related factor (Pearson's $r = 0.86$) while it shows a weak correlation with the road dust factor ($r = 0.34$). The two fireworks factors are resolved due to difference in elemental ratios in the factors as well as based on the residual structure during fireworks days. Fig. 4 in the manuscript shows that we indeed need two fireworks factors to explain the time series of elements related to fireworks.

*Source apportionment with receptor models is a routine technique and alone does not represent a scientific novelty. In addition, the application of these models with only trace elements does not fulfill the minimum requirements to obtain robust PM source apportionment because of the absence of important information to allocate the entire PM10 mass.*

Some source apportionment applications may be regarded as a routine technique. This is mostly the case for standardized chemical mass balance (CMB) or positive matrix factorization (PMF). The use of partial constraints in ME-2, hybrids of CMB and PMF, is not routine. In addition, it has so far mostly been applied to the organic aerosol, and much less to inorganic elements / metals. And, to reiterate, we did not attempt to allocate the entire mass to sources but rather a sub-fraction like it has been done frequently for the organic aerosol, but in this case mostly for trace elements which are partially specifically toxic.

*SPECIFIC COMMENTS*
*Page 1 line 29: the literature about the health impacts of PM need to be improved.*

We have already added several references for health impacts before line 29. We added a line in the manuscript on page 1 line 30 related to health impact of coarse fraction of PM as follows:
In Stockholm, Meister et al. (2012) estimated a 1.7% increase in daily mortality per 10 µg m$^{-3}$ increase in coarse fraction of PM.

Meister, K., Johansson, C., and Forsberg, B.: Estimated short-term effects of coarse particles on daily mortality in Stockholm, Sweden, Environ. Health Perspect., 120, 431–436, https://doi.org/10.1289/ehp.1103995, 2012.

*Page 3 line 6: the study by Park et al was carried out with XACT 620.*

We have added in the text some more references in this paragraph as follows: "The Xact 620, 625 and the newer 625i ambient metals monitors (Cooper Environmental Services, Beaverton, Oregon, USA) based on XRF have been developed in recent years and have been used in several field studies (Fang et al., 2015; Cooper et al., 2010; Furger et al., 2017; Park et al., 2014; Phillips-Smith et al., 2017; Tremper et al., 2018; Chang et al., 2018; Liu et al., 2019; Ji et al., 2018). However, only six studies included SA on Xact data (Park et al., 2014; Fang et al., 2015; Phillips-Smith et al., 2017; Chang et al., 2018; Liu et al., 2019; Ji et al., 2018)."

Cooper, J. A., Petterson, K., Geiger, A., Siemers, A., and Rupprecht, B.: Guide for developing a multi-metals, fenceline monitoring plan for fugitive emissions using X-ray based monitors, Cooper Environmental Services, Portland, Oregon, 1–42, 2010.

Fang, T., Guo, H., Verma, V., Peltier, R. E., and Weber, R. J.: $PM_{2.5}$ water-soluble elements in the southeastern United States: automated analytical method development, spatiotemporal distributions, source apportionment, and implications for health studies, Atmos. Chem. Phys., 15, 11667–11682, https://doi.org/10.5194/acp-15-11667-2015, 2015.

Phillips-Smith, C., Jeong, C.-H., Healy, R. M., Dabek-Zlotorzynska, E., Celo, V., Brook, J. R., and Evans, G.: Sources of particulate matter components in the Athabasca oil sands region: investigation through a comparison of trace element measurement methodologies, Atmos. Chem. Phys., 17, 9435–9449, https://doi.org/10.5194/acp-17-9435–2017, 2017.

Liu, Y., Zheng, M., Yu, M., Cai, X., Du, H., Li, J., Zhou, T., Yan, C., Wang, X., Shi, Z., Harrison, R. M., Zhang, Q., and He, K.: High-time-resolution source apportionment of $PM_{2.5}$ in Beijing with multiple models, Atmos. Chem. Phys., 19, 6595–6609, https://doi.org/10.5194/acp-19-6595-2019, 2019.

Ji, D., Cui, Y., Li, L., He, J., Wang, L., Zhang, H., Wang, W., Zhou, L., Maenhaut, W., Wen, T., and Wang, Y.: Characterization and source identification of fine particulate matter in urban Beijing during the 2015 Spring Festival, Sci. Total Environ., 430–440, 628–629, https://doi.org/10.1016/j.scitotenv.2018.01.304, 2018.

*Page 6 line 2: is it reasonable to use 10% analytical uncertainty for all the species?*

An estimated analytical uncertainty of 10% for all species has been used to derive the uncertainty data set in several previous studies (Kim et al., 2005; Kim and Hopke, 2007; Tian et al., 2016; Ji et al., 2018), which did not impact the interpretability of the PMF results. We also tried metal specific analytical uncertainty from Phillips-Smith et al. (2017) and Jeong et al. (2016) but it did not change the PMF results.

We have modified line 2 on page 6 as follows:

In this study, an estimated analytical uncertainty ($p_j$) of 10 % was used to derive the error matrix data set (Kim et al., 2005; Kim and Hopke, 2007; Tian et al., 2016; Ji et al., 2018), which did not change the PMF solution. The metal-specific analytical uncertainty was also considered from the previous studies (Jeong et al., 2016; Phillips-Smith et al., 2017), where it was calculated on the basis of high/medium-concentration metal standards laboratory

experiments and the additional 5-10% flow rate accuracy, which yielded similar PMF solutions compared to an overall 10 % analytical uncertainty.

Kim, E., Hopke, P. K., and Qin, Y.: Estimation of organic carbon blank values and error structures of the speciation trends network data for source apportionment, J. Air Waste Manage. Assoc., 55, 8, 1190–1199, https://doi.org/10.1080/10473289.2005.10464705, 2005.

Kim, E. and Hopke, P. K.: source identifications of airborne fine particles using positive matrix factorization and U.S. environmental protection agency positive matrix factorization, J. Air Waste Manage. Assoc., 57, 7, 811–819, https://doi.org/10.3155/1047-3289.57.7.811, 2007.

Tian, S. L., Pan, Y. P., and Wang, Y. S.: Size-resolved source apportionment of particulate matter in urban Beijing during haze and non-haze episodes, Atmos. Chem. Phys., 16, 1–19, https://doi.org/10.5194/acp-16-1-2016, 2016.

Jeong, C.-H., Wang, J. M., and Evans, G. J.: Source Apportionment of Urban Particulate Matter using Hourly Resolved Trace Metals, Organics, and Inorganic Aerosol Components, Atmos. Chem. Phys. Discuss., https://doi.org/10.5194/acp-2016-189, 2016.

Ji, D., Cui, Y., Li, L., He, J., Wang, L., Zhang, H., Wang, W., Zhou, L., Maenhaut, W., Wen, T., and Wang, Y.: Characterization and source identification of fine particulate matter in urban Beijing during the 2015 Spring Festival, Sci. Total Environ., 430–440, 628–629, https://doi.org/10.1016/j.scitotenv.2018.01.304, 2018.

Phillips-Smith, C., Jeong, C.-H., Healy, R. M., Dabek-Zlotorzynska, E., Celo, V., Brook, J. R., and Evans, G.: Sources of particulate matter components in the Athabasca oil sands region: investigation through a comparison of trace element measurement methodologies, Atmos. Chem. Phys., 17, 9435–9449, https://doi.org/10.5194/acp-17-9435–2017, 2017.

*Page 7 line 18: how do you know salt is from sea and not road salt?*

This point is discussed later in the detailed presentation of the sea salt factor (section 4.2, "sea salt"). As mentioned in the manuscript the observed Mg / Na ratios for sea salt events (0.13 and 0.16) are in line with the expected ratio for marine aerosol (0.135 to 0.185). The measured Cl / K and Cl / Ca ratios (0.27 and 0.33 respectively) from Xact do not lie in the range of road salt snow samples (150±39 and 103±22, respectively) collected near the roadside (≤50 m) in the USA (Williams et al., 2000). The CBPF plot (Fig. 5) and backward trajectories (Fig. S11) also validate the occurrence of the sea salt event. Moreover, during summer (mean temperature 21 $^0$C), we do not expect de-icing salt on the road.

We have added a line on Page 9 line 17 as follows:

The measured Cl / K and Cl / Ca ratios (0.27 and 0.33 respectively) from Xact do not lie in the range of road salt snow samples (150±39 and 103±22, respectively) collected near the roadside (≤50 m) in USA (Williams et al., 2000), which validates our interpretation as a sea salt factor.

Williams, A. L., Stensland, G. J., Peters, C. R., and Osborne, J.: Atmospheric dispersion study of deicing salt applied to roads: first progress report, Illinois State Water Surv. Contract Report, 2000-05, Champaign, IL, 2000.

*Page 7 line 25: the expression factor composition is not clear. Do you mean factor profile?*

Yes, the factor composition expression represents the factor profile. The mathematical expression for factor composition is mentioned in line 26 of that page. We replaced "factor composition" with "factor profile" to avoid confusion for readers.

*Page 9 line 14: the comparison between the on-line and off-line data is not clear. In the text are mentioned 28th and 30th July but the Mg/Na ratio is similar to marine aerosol also on 31st July.*

We apologize for the misunderstanding. Here we discuss the offline measurements of Na and Mg, which were not measured by the Xact, and their ratio in sea salt aerosols. The Mg / Na ratio on 31$^{st}$ July (0.28) does not lie in the range for marine aerosol but it is close to literature values (0.132 – 0.185). The sea salt factor time series as well as the time series of Cl from the Xact have a quite flat pattern on 31$^{st}$ July except for the fireworks peak at 23:00 LT with mostly south-east-north wind directions at low wind speeds (1–4 m s$^{-1}$) (Fig. S10) which may indicates the background concentration. In contrast, during the sea salt event, wind was mostly dominating from west directions at higher wind speeds (5–8 m s$^{-1}$).

We modified the text on Page 9 line 11 as follows:

Sea salt also includes Na and Mg, which were not measured by the Xact but analysed by ICP-OES for 24-h offline filters.

*Page 9 line 21 and foll.: There is not clear evidence to prove there are two different dust factors. The diurnal profiles are almost overlapping (Figure 3) and in the scatter plot of fig S13 there is only one cloud of points. Also the CBPF plots in figure S8 show overlapping wind speed and directions. Soil dust should be farther from the origin than the more local road dust. Split of sources is a typical problem of using only elements in factor analysis source apportionment. In addition, how do you know that Ca rich is road dust and not other sources? Enriched Ca dust is common in construction works, for instance.*

The scatter plot of Si and Ca in Fig. S13 shows quite scattered data points which might be due to influence of more than one source especially during the end of the measurement period. Therefore, there is no strong correlation between Si and Ca elements (Pearson's $r^2$ 0.56). The possible sources for Si and Ca could be the local and regional transport of crustal material partly re-suspended by traffic (south sector) and partly originating from the agricultural area north of the freeway. The factor time series of both factors are not identical except for a few overlapping points. The road dust diurnal pattern shows a maximum traffic rush hour peak similar to NO$_x$, black carbon (BC), heavy duty vehicle (HDV) count and the traffic-related factor while the background dust factor has no spikes in the diurnals (Fig. 3). Also our interpretation brackets the road dust factor because the road dust resuspension in the road side environment is primarily traffic induced. We have added the modified CBPF plots as well as an explanation about two dust factors in the 2$^{nd}$ referee's general comments section. However we would like to add that the agricultural land immediately to the west and north direction and another freeway (A2) in the northwest direction from the sampling location might be the reason for background dust in the CBPF plot.

We agree that Ca dust is also possible from construction dust. But this hypothesis does not support the Ca rich factor diurnal pattern when we assume that this kind of sources peaked during the day and decreased to almost zero outside of normal working times (08:00 until 17:00 LT). We also found that there is only a weak correlation in the hourly concentrations between Ca and S. If calcium sulfate dihydrate is the main compound in gypsum used

for construction work, Ca and S should show a better correlation.  The weak correlation could be possible due to the contribution from different sources for Ca (road dust) and S (ammonium sulfate). During the Ca peaks, ACSM $PM_1$ sulfate and ammonium did not show high peaks. However, comparing the ACSM $PM_1$ sulfate with $PM_{10}$ S may not be direct because of the existence of coarse sulfate in the form of calcium sulfate. Pure gypsum typically contains 29.4% Ca and 23.5% S, resulting in an elemental ratio of 1.25 which was not found in our measurements except one data point.

We added the text in the manuscript as follows on page 10 line 1:

Ca has been associated with construction activities in previous studies (Bernardoni et al., 2011; Crilley et al, 2016), which have been found to peak during the day and decreased to almost zero outside of normal working times (08:00 until 17:00 LT). This evidence is not supported by the road/background dust factor diurnal pattern in this study. Further evidence of non-construction activity is found in the Xact elemental ratio of Ca / S (0.62) which is not in the agreement with the pure gypsum Ca / S ratio (1.25) used for construction work (Hassan et al., 2014), where they are the main constituents.

Bernardoni, V., Vecchi, R., Valli, G., Piazzalunga, A., Fermo, P.: $PM_{10}$ source apportionment in Milan (Italy) using time-resolved data, Sci. Total Environ., 409, 4788–4795, https://doi.org/10.1016/j.scitotenv.2011.07.048, 2011.

Crilley, L. R., Lucarelli, F., Bloss, W. J., Harrison, R. M., Beddows, D. C., Calzolai, G., Nava, S., Valli, G., Bernardoni, V., and Vecchi, R.: Source apportionment of fine and coarse particles at a roadside and urban background site in London during the 2012 summer ClearfLo campaign, Environ. Pollut., 220, 766–778, https://doi.org/10.1016/j.envpol.2016.06.002, 2016.

Hassan, A. K., Fares, S., and Abd El-Rahma, M.: Natural radioactivity levels and radiation hazards for gypsum materials used in Egypt, J. Environ. Sci. Technol., 7, 56–66, https://doi.org/10.3923/jest.2014.56.66, 2014.

*Page 9 line 25: only the Ca scaled residuals improve not those of Si.*

The Si scaled residual was also improved a bit (from 6 to 4). Increasing the number of factors further did not improve the Si scaled residual structure.

*Page 10 line 17 and foll.: The chemical composition of this factor indicates brake wear mixed to the re-suspended particles as Fe and Si are components of crustal material. Again there is an overlapping in the chemical composition with the so called road dust profile and these two profiles also have very similar diurnal profiles.*

Although Fe and Si are components of crustal material, both can be emitted from anthropogenic sources as well. The diurnal pattern of Fe shows traffic rush hour peaks while Si diurnal cycle is not as prominent as Fe. The sources of Fe are already discussed in the manuscript. We added a sentence for Si in the manuscript. If the road dust factor is attributable mainly to abrasion of the road surface and tires, and the resuspension by traffic vortices, a similar diurnal pattern with the traffic-related factor would be expected.

We have added the following text in the manuscript on page 10 line 24:

Si is one of the brake lining components used as abrasive to increase friction, and as fillers to reduce manufacturing costs, respectively (Thorpe and Harrison, 2008; Grigoratos and Martini, 2015).

Thorpe, A. and Harrison, R. M.: Sources and properties of non-exhaust particulate matter from road traffic: A review, Sci. Total Environ., 400, 270–282, https://doi.org/10.1016/j.scitotenv.2008.06.007, 2008.

Grigoratos, T. and Martini, G.: Brake wear particle emissions: a review, Environ. Sci. Pollut. Res., 22, 2491–2504, https://doi.org/10.1007/s11356-014-3696-8, 2015.

*Page 10 line 29: the secondary sulfate does not provide any evidence to allocate the ammonium nitrate (another main secondary inorganic component of PM).*

Yes, we agree with reviewer#1 that it does not provide evidence to allocate the ammonium nitrate but this is not what we are attempting in this paper. It is outside the scope of this paper to perform an ACSM PMF analysis.

*Page 11 line 8: The characterization of the industrial source is weak. There is no connection to any known industrial process. The fact that other studies in completely different areas found the same profile is not proving the industrial origin of this factor. On the contrary, it is unlikely that the same industrial process is present in many different sites.*

This factor time series contains a few spikes which correlate with wind direction, indicating a local point source. Based on this hypothesis, we interpreted this factor as industrial source. The other previous studies are mentioned to represent similar factor profiles from local point sources without a specific industrial emission.

We have modified the text in the manuscript on page 11 line 11:

A similar factor profile was observed in previous source apportionment studies (Crilley et al., 2016; Richard et al., 2011) from local point source emissions without any link to specific industrial activities. Amato et al. (2010) and Dall'Osto et al. (2013) also reported a PMF factor with high concentrations of Zn and Pb and attributed it to emissions from smelters in Barcelona, while Vossler et al. (2016) reported a similar factor profile from coal combustion processes in Ostrava.

Crilley, L. R., Lucarelli, F., Bloss, W. J., Harrison, R. M., Beddows, D. C., Calzolai, G., Nava, S., Valli, G., Bernardoni, V., and Vecchi, R.: Source apportionment of fine and coarse particles at a roadside and urban background site in London during the 2012 summer ClearfLo campaign, Environ. Pollut., 220, 766–778, https://doi.org/10.1016/j.envpol.2016.06.002, 2016.

Richard, A., Gianini, M. F. D., Mohr, C., Furger, M., Bukowiecki, N., Minguillón, M. C., Lienemann, P., Flechsig, U., Appel, K., DeCarlo, P. F., Heringa, M. F., Chirico, R., Baltensperger, U., and Prévôt, A. S. H.: Source apportionment of size and time resolved trace elements and organic aerosols from an urban courtyard site in Switzerland, Atmos. Chem. Phys., 11, 8945–8963, https://doi.org/10.5194/acp-11-8945-2011, 2011.

Amato, F., Nava, S., Lucarelli, F., Querol, X., Alastuey, A., Baldasano, J. M., and Pandolfi, M.: A comprehensive assessment of PM emissions from paved roads: Real-world emission factors and intense street cleaning trials, Sci. Total Environ., 408, 4309–4318, https://doi.org/10.1016/j.scitotenv.2010.06.008, 2010.

Dall'Osto, M., Querol, X., Amato, F., Karanasiou, A., Lucarelli, F., Nava, S., Calzolai, G., and Chiari, M.: Hourly elemental concentrations in $PM_{2.5}$ aerosols sampled simultaneously at urban background and road site during SAPUSS – diurnal variations and PMF receptor modelling, Atmos. Chem. Phys., 13, 4375–4392, https://doi.org/10.5194/acp-13-4375-2013, 2013.

Vossler, T., Černikovský, L., Novák, J., and Williams, R.: Source apportionment with uncertainty estimates of fine particulate matter in Ostrava, Czech Republic using positive matrix factorization, Atmos. Pollut. Res., 7, 503–512, https://doi.org/10.1016/j.apr.2015.12.004, 2016.

*Figure 4: Not clear what is the added value of this figure with respect to figure 1. Normalized concentration (should be explained in the caption). Why use different units than the other graphs. The percentile box and whisker and the mean -+SD overlaps making the graph difficult to read. Choose one.*

PMF results (Fig. 1) describe potentially complex and time-dependent sources as a single factor (or linear combination of factors). This is a fundamental limitation of the model, and may not accurately describe the behaviour or a complex source (e.g., residuals may be significant). In contrast, Fig. 4 provides an estimate of the overall fireworks composition and temporal variability, complementing the PMF results (which are shown for reference). One can then see both the explanatory power and limitations of the SA model for this source.

We moved Fig. 4 to the supplement and removed ± SD from the figure.

We added in the supplement on page 8 line 6:

Fig. S8 provides an estimate of the overall fireworks composition and temporal variability, complementing the PMF results (which are shown for reference). The figure is constructed in two stages. First, the time series of fireworks contributions to each element is estimated by subtracting the non-fireworks factors (NFF) from the original measurements. Then, the estimated fireworks contribution for each element is normalized by the total fireworks element contribution, and the displayed statistics are calculated. This is represented mathematically below, and the expression has been added to the main text. Note that the variation in fireworks profiles implied by this figure supports the representation of fireworks by 2 fireworks factors.

$$normalized\ concentration_{ij} = \frac{x_{ij} - \left(g_{ik}f_{kj}\right)_{k=NFF}}{\Sigma_j\left(x_{ij} - \left(g_{ik}f_{kj}\right)_{k=NFF}\right)} \tag{S1}$$

where X represents the input data matrix for PMF, while G and F represent the factor time series and factor profiles driven by six non-fireworks factors (NFF). Here *i* and *j* denote time series and variables, respectively.

*Figure S9: the connection between this data and the days with high "sea salt" is not clear in this figure. It should be better evidenced to allow the reader evaluating if there is a difference between the days with high sea salt and those without it.*

We apologise for the misunderstanding. Fig. S9 only shows the Mg / Na ratio for 24-h filter data analysed by ICP-OES along with the literature Mg / Na ratio (0.132 - 0.185) in marine aerosols. In this figure, we presented the existence of sea salt events and non-sea salt events. The sea salt factor time series with high and low concentrations is already shown in Fig.1 and discussed in the manuscript.

We added a line in the caption of Fig. S9 caption on page 9 line 5:

The Mg / Na ratio was 0.13 and 0.16 for 28 July and 30 July, respectively while for the rest of the days it was higher than 0.185.

*TECHNICAL CORRECTIONS*

*Figure S6: not clear why the p25 is a single point while the p75 is a vertical line.*

We modified Fig. S6.

[Figure]

Figure S6: *a*-value statistics of the accepted solutions. *a*-values between 0 to 0.5 were explored during BS analysis. The average *a*-value of the selected solutions was ranging from 0.2 to 0.3 for the constrained factors. The selected *a*-values were homogeneously distributed over that range.

---

## Author Comment (AC2) · 5 Jul 2019

**Source apportionment of highly time resolved trace elements during a firework episode from a rural freeway site in Switzerland**

Pragati Rai[1], Markus Furger[1], Jay Slowik[1], Francesco Canonaco[1], Roman Fröhlich[1], Christoph Hüglin[2], María Cruz Minguillón[3], Krag Petterson[4], Urs Baltensperger[1] and André S.H. Prévôt[1]

[1] Laboratory of Atmospheric Chemistry, Paul Scherrer Institute, Villigen PSI, 5232, Switzerland
[2] Laboratory for Air Pollution / Environmental Technology, Empa, 8600 Dübendorf, Switzerland
[3] Institute of Environmental Assessment and Water Research (IDAEA), CSIC, 08034 Barcelona, Spain
[4] Cooper Environmental Services (CES), 9403 SW Nimbus Avenue, Beaverton, OR 97008, USA

*Correspondence to*: André S. H. Prévôt (andre.prevot@psi.ch) and Markus Furger (markus.furger@psi.ch)

**Response to Reviewer #2**

We kindly thank the reviewer#2 for taking our manuscript into consideration and we value the comments raised to improve the manuscript. A point-by-point answer (in regular typeset) to the reviewers' remarks (in italic typeset) follows. Changes to the manuscript are indicated in blue font.

In the following page and lines references refer to the manuscript version reviewed by anonymous Reviewer#2.

*This paper examines data previously reported by Furger et al (2017) applying PMF source apportionment techniques to a subset of the reported data to provide insight into the chemical composition of different sources. It focuses on firework / burning sources associated with local celebrations (which are a common influence on air quality for short periods in many countries) and to a lesser extent on suspended dust from road and surface sources. The article is generally well written and the approach is methodologically robust. However, the data used is somewhat limited and the authors have not taken advantage of the aerosol mass spectrometer data which was available. I have many suggested edits and some points which require consideration by the authors.*

*Major Comments –*

*Pg 5, line 16 – the authors reduce the data set from 24 to 14 based on manufacturer supplied MDLs to produce 'better' source apportionment results. Firstly, even though the measurements are below the detection limit, they could contain 'valuable' information on variability, especially during firework periods as some of the excluded elements are potential firework components (e.g. Cd). They have been excluded based on %below MDL at a rather arbitrary value e.g. Bi at 93% was included while Cd at 87% was not. It would be worth including this data and downweighting rather that excluding it or at least performing sensitivity test. A further 4 elements are excluded based on data quality but no detail on what constitutes data quality is given.*

We agree with reviewer#2 and therefore we had included Bi which shows fireworks peaks while for the rest of time it is below MDL. We inspected all the elements before preparing the PMF input data based on MDL. V, Co, As, Se, Cd and Pt were close to or below their Xact MDL. The inter-comparison of Xact vs ICP data for Co, Ni, As, Se, Cd and Hg was not justified based on the weak correlation ($r^2$). Pt was below MDL (98%), and it was not measured on the filters, therefore no conclusion about the Pt accuracy can be drawn. We did not find any fireworks

peaks for any of these elements. The detailed discussion on Xact data quality is described in more detail in the previous study (Furger et al., 2017).

We modified the text on Pg 5 line 20:

Elements (% of data points below MDL, $r^2$ value) which had more than 50 % of data points below MDL and low $r^2$ (0.5) between Xact and offline data were not included in the PMF input, such as: V (98 %, 0.57), Co (100 %, 0.05), Ni (32 %, 0.22), As (96 %, 0.5), Se (62 %, 0.3), Cd (87 %, 0.18), Sn (15 %, 0.27), Sb (6%, 0.42), Hg (13 %, 0.12) and Pt (98 %, not measured on the filters). The element Bi (93% data points below MDL) was an exception to include in the PMF input due to an excellent correlation between Xact and offline data ($r^2 = 0.98$) during fireworks peaks. The detailed description of the Xact data quality is given in the previous study (Furger et al., 2017).

Furger, M., Minguillón, M. C., Yadav, V., Slowik, J. G., Hüglin, C., Fröhlich, R., Petterson, K., Baltensperger, U., and Prévôt, A. S. H.: Elemental composition of ambient aerosols measured with high temporal resolution using an online XRF spectrometer, Atmos. Meas.Tech., 10, 2061–2076, https://doi.org/10.5194/amt-10-2061-2017, 2017.

*Pg 8 Line 4 and later – The CBPF is an important analysis which is used to justify the identification of sources and the charts should not be shunted to the appendix. Further, the reliance on 90th percentile to identify the sources is flawed when considered on its own, especially when the distribution of measurements is heavily skewed (as it is with Cl). A range of CBPFs should be presented (at least in the supplementary) to support the interpretation. The authors should also consider the use of the min bin statistic to ensure that individual points are not skewing the interpretation. Finally, the conclusion that the two dust sources are from different sources (based on these plots) when their direction is broadly similar is rather weak.*

We agree that it would be helpful moving the CBPF plots from the Supplement to the main manuscript. We presented a range of CBPF plots in the supplementary for these four factors (as shown in the figure below). We partially agree with reviewer#2 for min bin statistics. The min bin parameter provides the minimum number of points required to provide a result in a wind speed/wind direction bin and is 1 in this analysis. If we set it >1, there is the risk of removing real data. Concerning the two dust factors, comparing the two CBPF ranges, there are indeed differences in terms of source directions. The road dust factor is quite consistent with southerly winds except for a few low concentration data points (at the 50[th] percentile) while the background dust source direction is quite variable for different ranges of data.

[Figure]

[Figure]

Figure S16: CBPF analysis (from left to right: 50[th], 75[th], 90[th], 95[th] percentiles) of factors (sea salt, background dust, road dust, industrial) in terms of wind speed (m s[-1]) and wind direction. The color code represents the probability of the factor contribution.

*Pg 8 Line 9 and later – KNO3 is quoted as comprising 74% of fireworks. In which case NO3 could be a useful tracer for unburnt firework material - it was measured but is not reported. A recognition of this point, even to say that there was no correlation would help. The same is true of the other mass concentrations and m/z measured by the ACSM – were these examined or even considered for inclusion in the PMF or data analysis?*

We added a figure in the supplement to show the ACSM inorganics concentration along with $NO_2$ and $NO_x$ during the fireworks episode. From the figure it is very obvious that neither particulate nitrate nor ammonium is generated in the fireworks in significant amounts. The ACSM nitrate showed a quick drop immediately before the main fireworks (1 August 2015 23:00 LT). The sulphate peaks coincide with a strong drop of the nitrate concentration, which indicates that ammonium nitrate may have reacted with sulfuric acid, releasing nitric acid to the gas phase. The absence of fireworks nitrate has been observed previously (Drewnick et al., 2006) and indicates that most of the black powder nitrate is converted to other forms of nitrogen species. We calculated the measured $NO_2$ / K

mass ratio for enhanced $NO_2$ concentration during the fireworks based on K concentration. The ratio was 1.66 on the main fireworks hour (1 August 23:00 LT) which is close to the atomic ratio of $NO_2$ / K (1.17). This measured ratio is in-line with the $NO_2$ / $K^+$ (2.03) ratio measured during Chinese Spring Festival in Shanghai (Yao et al., 2019).

We have added the following text in the revised manuscript on pg8 line 10:

The ACSM inorganics concentrations during the fireworks episodes indicate that neither particulate nitrate nor ammonium is generated in the fireworks in significant amounts (Fig. S7). Consistent with previous measurements of submicron fireworks aerosol (Drewnick et al., 2006; Vecchi et al., 2008; Jiang et al., 2015), nitrate was not enhanced during the fireworks period, suggesting conversion of $KNO_3$ to other forms of nitrogen. The $NO_2$ / K mass ratio (1.66) on the main fireworks hour (1 August 23:00 LT) is close to the atomic ratio of $NO_2$ / K (1.17). This measured ratio is also in agreement with the $NO_2$ / $K^+$ (2.03) ratio observed during Chinese Spring Festival in Shanghai (Yao et al., 2019). However, the $NO_2$ and/or $NO_x$ variation was not significant during the fireworks peaks in the present study (Fig. S7), which is in agreement with the former studies (Vecchi et al., 2008; Retama et al., 2019; Yao et al., 2019).

Drewnick, F., Hings, S. S., Curtius, J., Eerdekens, G., and Williams, J.: Measurement of fine particulate and gas-phase species during the New Year's fireworks 2005 in Mainz, Germany, Atmos. Environ., 40, 4316–4327, https://doi.org/10.1016/j.atmosenv.2006.03.040, 2006.

Vecchi, R., Bernardoni, V., Cricchio, D., D'Alessandro, A., Fermo, P., Lucarelli, F., Nava, S., Piazzalunga, A., and Valli, G.: The impact of fireworks on airborne particles, Atmos. Environ., 42, 1121–1132, https://doi.org/10.1016/j.atmosenv.2007.10.047, 2008.

Jiang, Q., Sun, Y. L., Wang, Z., and Yin, Y.: Aerosol composition and sources during the Chinese Spring Festival: fireworks, secondary aerosol, and holiday effects, Atmos. Chem. Phys., 15, 6023–6034, https://doi.org/10.5194/acp-15-6023-2015, 2015.

Yao, L., Wang, D., Fu, Q., Qiao, L., Wang, H., Li, L., Sun, W., Li, Q., Wang, L., Yang, X., Zhao, Z., Kan, H., Xian, A., Wang, G., Xiao, H., and Chen, J.: The effects of firework regulation on air quality and public health during the Chinese Spring Festival from 2013 to 2017 in a Chinese megacity, Environ. Int., 126, 96–106, https://doi.org/10.1016/j.envint.2019.01.037, 2019.

Retama, A., Neria-Hernández, A., Jaimes-Palomera, M., Rivera-Hernández, O., Sánchez-Rodríguez, M., López-Medina, A., and Velasco, E.: Fireworks: a major source of inorganic and organic aerosols during Christmas and New Year in Mexico city, Atmos. Environ., 2, 100013, https://doi.org/10.1016/j.aeaoa.2019.100013, 2019.

[Figure]

Figure S7: Time series of the non-refractory aerosol components (measured with the ACSM), NO₂ and NOₓ concentration. The fireworks episodes are underlain in grey color.

*Line 29 and later – The use of SoFi to separate highly correlated sources such as the two firework sources is the reason for this paper. The consideration of Cl displacement is interesting and has evidence to support it, why does it therefore come after 2 purely speculative thoughts about possible alternative sources. Promote this point and consider the speculations as alternatives. One of these - the separation of bonfires from fireworks may have been helped by considering the ACSM data.*

We have rearranged the text from line 26 onwards as follows:

Chemical reactions of KCl with $H_2SO_4$ will result in a release of gaseous HCl and may explain the absence of particulate Cl in the fireworks-II factor profile. The time series variations in both fireworks suggest that fireworks-I might be related to the main fireworks celebration while fireworks-II might result from burning of leftover crackers after the main fireworks day, as well as the influence of other sources such as bonfires, which are a common activity during Swiss National Day celebrations. Another possibility could be the advection of fireworks clouds from nearby cities where grand firework displays and bonfires are carried out at large scale to celebrate the Swiss National Day.

*General Comment-*
*The authors use the term 'trace' elements. I would not consider S or Fe trace elements in ambient PM. I suggest the authors just refer to them as elements.*

We removed "trace" from the manuscript.

*Pg 1 Line 13 - Source Finder software, please quote version number, supplier and country*
Done.

*Line 19 - The abstract is partly written in the present tense – 'experiences' should read 'experienced', 'concentrations are similar' should read 'concentrations were similar'*

We corrected the grammatical error in abstract.

Further, there were minor contributions (on the order of a few percent) of sea salt and industrial sources. The regionally influenced secondary sulfate factor showed negligible resuspension, and concentrations were similar throughout the day. The significant loads of the traffic-related and road dust factors with strong diurnal variations highlight the continuing importance of vehicle-related air pollutants at this site.

*Pg 2 Line 29 – clarify that you are talking about off line sampling systems*

We added the word "offline" in the text.

*Line 33 – the sampler don't specifically need those analysis techniques, they require techniques with high precision and low detection limits – such as these methods*

We modified this line as follows:

These offline analyses require high precision and low detection limit techniques such as synchrotron radiation induced X-ray fluorescence spectrometry (SR-XRF) of aerosol samples collected with a RDI, particle induced X-ray emission (PIXE) with the streaker sampler and graphite furnace atomic absorption spectrometry (GFAAS) with the SEAS.

*Pg 3 Line 28 – delete XACT manufacturer already provided line 4*
Done.

*Line 30 – with should read at Line 31 onwards – be consistent with how you name equipment – model, make, supplier, country*
Done.

*Line 31 – quote filter manufacturer*
Done.

*Pg 4 Line 2 – TEOM and FDMS in full then in brackets or just the abbreviation, not a mixture*
Done.

*Line 20 – can values be applied to individual data points; this implies that individual points can be used for a-value constraining. Is this true? Certain sections in a time series can but individual points cannot be separated out in this way.*

Yes, it is possible to apply an *a*-value on individual data points because PMF renormalizes the factors at the end.

*Line 28 – this is not the first use of sofi in the manuscript - the reference, version number and supplier should go there.*

Done, except reference. The first use of SoFi is in the abstract section. One should not use references in the abstract.

*Pg 5 Line 1 – 'PMF' rather than 'model'*
Done.
*Line 4 – 'identify' rather than 'find'*
Done.
*Line 4 – this only helps to identify the high concentration peaks and whether then influence the mean will depend on the distribution of data (see Cl results in fig 4)*

We partially agree with the reviewer#2. CBPF gives directional information of sources contributing to pollutant concentrations at the sampling site.

*Line 22 – Bi a major component of fireworks –a reference, some evidence would be good here and even justify the exclusion of other elements.*

We deleted Bi a major tracer of fireworks from line 22 and added the reference for Bi on Pg8 line 12. Concerning exclusion of elements, please see the response in the major comments section.
We added a sentence on Pg 8 line 12 as follows:
Bi is used in crackling stars (Dragon's eggs) in the form of bismuth trioxide or subcarbonate as a non-toxic substitute for toxic lead compounds (Perrino et al., 2011).
Perrino, C., Tiwari, S., Catrambone, M., Torre, S. D., Rantica, E., and Canepari, S.: Chemical characterization of atmospheric PM in Delhi, India, during different periods of the year including Diwali festival, Atmos. Pollut. Res., 2, 418–427, https://doi.org/10.5094/apr.2011.048, 2011.

*Pg 6 Line 1 - need a description on xij*
Done.

*Line 16 – listed s1-s4 but only s2 and s3 relate to this point*

We removed S1, S2 and S4 according to text change in the revised manuscript.

*Pg 7 Line 18 – how do you know it is coarse Cl, you have two measures of Cl one is non-refractory Cl- in PM1 and one is Cl in PM10?*

We replaced "coarse" with $PM_{10}$ in line 18.

*Pg 8 Line 6 –are these elements really the main components of fireworks by mass, surely you need a reference here to justify this.*

The fireworks factors are mostly dominated by K, S, and Cl, among the elements analysed here. Moreover, Ti, Cu, Ba and Bi are key tracers of these factors.

We have added references in line 6:

(Moreno et al., 2007; Wang et al., 2007; Vecchi et al., 2008; Perrino et al., 2011; Tian et al., 2014; Kong et al., 2015; Lin, 2016; Pongpiachan et al., 2018).

Moreno, T., Querol, X., Alastuey, A., Minguillón, M. C., Pey, J., Rodriguez, S., Miró, J. V., Felis, C., and Gibbons, W.: Recreational atmospheric pollution episodes: Inhalable metalliferous particles from firework displays, Atmos. Environ., 41, 913–922, https://doi.org/10.1016/j.atmosenv.2006.09.019, 2007.

Perrino, C., Tiwari, S., Catrambone, M., Torre, S. D., Rantica, E., and Canepari, S.: Chemical characterization of atmospheric PM in Delhi, India, during different periods of the year including Diwali festival, Atmos. Pollut. Res., 2, 418–427, https://doi.org/10.5094/apr.2011.048, 2011.

Wang, Y., Zhuang, G., Xu, C., and An, Z.: The air pollution caused by the burning of fireworks during the lantern festival in Beijing, Atmos. Environ., 41, 417–431, https://doi.org/10.1016/j.atmosenv.2006.07.043, 2007.

Vecchi, R., Bernardoni, V., Cricchio, D., D'Alessandro, A., Fermo, P., Lucarelli, F., Nava, S., Piazzalunga, A., and Valli, G.: The impact of fireworks on airborne particles, Atmos. Environ., 42, 1121–1132, https://doi.org/10.1016/j.atmosenv.2007.10.047, 2008.

Kong, S. F., Li, L., Li, X. X., Yin, Y., Chen, K., Liu, D. T., Yuan, L., Zhang, Y. J., Shan, Y. P., and Ji, Y. Q.: The impacts of firework burning at the Chinese Spring Festival on air quality: insights of tracers, source evolution and aging processes, Atmos. Chem. Phys., 15, 2167–2184, https://doi.org/10.5194/acp-15-2167-2015, 2015.

Lin, C.-C.: A review of the impact of fireworks on particulate matter in ambient air, J. Air Waste Manage. Assoc., 66, 12, 1171–1182, https://doi.org/10.1080/10962247.2016.1219280, 2016.

Pongpiachan, S., Iijima, A., and Cao, J.: Hazard quotients, hazard indexes, and cancer risks of toxic metals in $PM_{10}$ during firework displays, Atmosphere, 9, 144, https://doi.org/10.3390/atmos9040144, 2018.

Tian, Y. Z., Wang, J., Peng, X., Shi, G. L., and Feng, Y. C.: Estimation of the direct and indirect impacts of fireworks on the physicochemical characteristics of atmospheric $PM_{10}$ and $PM_{2.5}$, Atmos. Chem. Phys., 14, 9469–9479, https://doi.org/10.5194/acp-14-9469-2014, 2014.

*Line 7 – chlorate and perchlorate (line 11) will also be detected as Cl in XRF*

Indeed, the Xact measures total chlorine including all particulate Cl compounds such that we cannot distinguish them with the Xact measurements.

*Line 16 – the sentence starting 'A pronounced: : :) should really come at te start of the para*
Done.

*Line 29 – this consideration of Cl displacement is interesting and at has evidence to support it, why does it therefore come after 2 purely speculative thoughts. Promote this point and consider the speculations as alternatives.*

Please see the response above in the major comment.

*Pg 11 – line 6 – there is no fig S12*

We corrected it as Fig. S9.

---

## Author Comment (AC3) · 5 Jul 2019

**Source apportionment of highly time resolved trace elements during a firework episode from a rural freeway site in Switzerland**

Pragati Rai[1], Markus Furger[1], Jay Slowik[1], Francesco Canonaco[1], Roman Fröhlich[1], Christoph Hüglin[2], María Cruz Minguillón[3], Krag Petterson[4], Urs Baltensperger[1] and André S.H. Prévôt[1]

[1] Laboratory of Atmospheric Chemistry, Paul Scherrer Institute, Villigen PSI, 5232, Switzerland
[2] Laboratory for Air Pollution / Environmental Technology, Empa, 8600 Dübendorf, Switzerland
[3] Institute of Environmental Assessment and Water Research (IDAEA), CSIC, 08034 Barcelona, Spain
[4] Cooper Environmental Services (CES), 9403 SW Nimbus Avenue, Beaverton, OR 97008, USA

*Correspondence to*: André S. H. Prévôt (andre.prevot@psi.ch) and Markus Furger (markus.furger@psi.ch)

**Response to Reviewer #3**

We kindly thank the reviewer#3 for taking our manuscript into consideration and we value the comments raised to improve the manuscript. A point-by-point answer (in regular typeset) to the reviewers' remarks (in italic typeset) follows. Changes to the manuscript are indicated in blue font.

In the following page and line references refer to the manuscript version reviewed by anonymous Reviewer#3.

*The paper "Source apportionment of highly time resolved trace elements during a firework episode from a rural freeway site in Switzerland" by Rai et al. deals with Positive Matrix Factorization analysis of a dataset of 1-h resolved trace elements. Despite the authors declare the existence of much more information on high time resolved scale (mass concentration by TEOM, equivalent black carbon by MAAP, ACSM data), they carried out the analysis on elements only. This limits strongly the information provided by the study (e.g. they can apportion only the mass related to elements, that they estimated to be about 20% of PM10 mass). Thus, the results cannot be representative*

*of the total contribution of the sources to the measured PM. Furthermore, lots of constraints were implemented to reach the final solution. In some cases, both source profiles and temporal trends were constrained. So many constraints to the model make questionable the validity of the results, also considering that these constraints are not adequately supported by a methodological description of the way they were obtained, as explained in more detail below. Whether the paper is well structured and well written in most parts (even if some obscure descriptions remain), the scientific aspect is not fully convincing.*

*Major concerns:*
*Page 5, line 15: excluding mass data (mentioned at p.4 l.2) should be carefully discussed as it strongly reduces the interest of the results, preventing an absolute quantification of the factor contributions to the measured PM. Furthermore, maybe the authors decided to analyse separately ACSM data, but at least equivalent BC information could be of help in source resolution.*

This issue was also raised by Reviewer #2 (major comments section) and our response is presented in both places for clarity. The exclusion of elements was done based on online (Xact 625) and offline (ICP analysis) elemental quality comparison as discussed in Furger et al. (2017). We modified the text on Pg 5 line 20:

Elements (% of data points below MDL, $r^2$ value) which had more than 50 % of data points below MDL and low $r^2$ (0.5) between Xact and offline data were not included in the PMF input, such as: V (98 %, 0.57), Co (100 %, 0.05), Ni (32 %, 0.22), As (96 %, 0.5), Se (62 %, 0.3), Cd (87 %, 0.18), Sn (15 %, 0.27), Sb (6 %, 0.42), Hg (13 %, 0.12) and Pt (98 %, not measured on the filters). The element Bi (93% data points below MDL) was an exception to include in the PMF input due to an excellent correlation between Xact and offline data ($r^2 = 0.98$) during fireworks peaks. The detailed description of the Xact data quality is given in the previous study (Furger et al., 2017).

Concerning ACSM and BC data, the same issue was also raised by Reviewer#1 and Reviewer#2. We repeat here the same. In our opinion, the combination of $PM_1$ data for Q-ACSM, $PM_{2.5}$ for MAAP and $PM_{10}$ data for Xact may introduce artefacts in the ME-2 analysis. We have used $NO_x$, MAAP and Q-ACSM data sets in the manuscript to compare some of the ME-2 factor time series and diurnal patterns. We are not planning to publish a separate paper on the other data. A major focus was the exploration of the use of the Xact for source apportionment of inorganic elements / metals in Europe where the concentrations are considerably lower than in polluted areas in Asia.

Furger, M., Minguillón, M. C., Yadav, V., Slowik, J. G., Hüglin, C., Fröhlich, R., Petterson, K., Baltensperger, U., and Prévôt, A. S. H.: Elemental composition of ambient aerosols measured with high temporal resolution using an online XRF spectrometer, Atmos. Meas.Tech., 10, 2061–2076, https://doi.org/10.5194/amt-10-2061-2017, 2017.

*Page 6, lines 11-12. "The unconstrained PMF solutions yielded mixed factor solutions. Therefore, it was essential to constrain specific factor profiles and the time series in the PMF analysis to avoid mixing (see details in the Supplement, section S1, Fig. S1)". This is the weakest aspect of all. The way in which the factor profiles/time series are constrained is the key point for all the rest of the analysis. Constraints determination strongly affects the final results (see described differences between the preliminary analysis and the constrained one) and merits detailed description and attention. Opposite, its description was moved to the Supplemental Material, and what is reported there is a not sufficient to determine the robustness of the approach. The following information should have been provided: 1) How many factors were present in the final analyses of the fireworks and non-fireworks periods? 2) Were the sources other than fireworks and sea-salt comparable between the datasets? (In terms of profile and tracer species?) 3) Were 2 firework-related factors identified in the analysis of the firework-days subset? 4) What about the residual of S in the unconstrained 9 factor solution? 5) The need to constrain both profile and temporal trend of the sulphate source is very suspicious (the whole "secondary sulfate" factor was constrained in the final solution).*

We agree with the reviewer#3. We have rewritten the section S1 below and we moved this section into the main text. However, we would like to respond point by point to the reviewer's five questions.

The input data set was divided into two parts: fireworks days (FD; 31 July–4 August) and non-fireworks days (NFD; all days except 31 July–4 August). To obtain a specific fireworks profile, we further selected only fireworks

hours (FH; 31 July 21:00–1 August 07:00 LT) as PMF input data. The PMF analysis was performed on the NFD, FD, FH and the complete datasets separately for three to ten factors, with each of these solutions investigated with ten seeds (each seed represents a different pseudorandom initialization).

1) Seven factors were resolved in the NFD PMF analysis while a five-factor solution was identified in the FH PMF analysis.

2) The unconstrained NFD PMF analysis resolved factors such as sea salt, secondary sulfate, traffic-related, road dust, background dust, industrial and a K-rich factor. The FD and FH constrained PMF analyses resolved a five-factor solution with secondary sulfate, sea salt, fireworks, background dust and a K-rich factor with fireworks related elements. The traffic-related and road dust factors were resolved in FD and FH PMF analyses in seven and eight-factor solution, which were comparable to NFD PMF analysis. The fireworks factor contribution was going down in seven and eight-factor solution than the five-factor solution. The industrial factor in the FD and FH PMF analysis was resolved in eight-factor solution where it was mainly characterised by high contribution to Pb (67 %) and Zn (30 %) whereas in NFD PMF analysis the relative contribution to Pb and Zn were 90 % and 87 %, respectively.

3) The FD and FH constrained PMF analyses identified two fireworks-related factors, which were comparable to the final two fireworks-related factors profiles.

4) The scaled residual of S in the unconstrained nine-factor solution was between -0.46 and 0.85.

5) We tested several approaches for the secondary sulfate constraint: a) constraining only the profile; b) constraining only the time series to preserve secondary sulfate temporal trend during fireworks; c) constraining the entire time series along with factor profile; d) constraining the factor profile together with a segment of the time series (i.e., only during fireworks days). From all the above four tests, unmixed secondary sulfate was possible only in case of c and d, and we prefer the approach in d as it provides maximum freedom to the algorithm. We suggest that the need to constrain both the profile and part of the time series is driven by the very high concentration and variation in composition during the fireworks period. Because the signal-to-noise is very high, imperfections in the model description exert a strong influence on Q, and the model therefore tries to compensate by assigning fireworks mass to other factors. The double constraint tactic in d avoids this problem, while minimizing overall constraints on the solution.

These points, as well as the requested general information regarding the PMF analysis, are added to the manuscript beginning at Page 6 line 9:

[revised manuscript text omitted]

*Pag 6, line 13: "obvious structure": completely obscure what the authors mean.*

We meant significant structure in residuals.  We modified the text on Page 6 line 13 as follows:
Residual analysis (Q-contribution over the time series) of the PMF runs showed significant structure in the residuals ((Q maximum value was 15 during fireworks period as shown in Fig. S3) for solutions having up to seven factors.

*Pag 6, line 28: "both the sea salt and secondary sulfate factor time series were also constrained with a-value 0.01". Further constraint implemented. Please note that a small a-value was used. Is it consistent with uncertainty estimates for the non-firework sub-set and for the sulfate factor identified in the preliminary 9-factor solution? Please also note that the constraints come from two different analyses.*

We understand the concern of the reviewer#3. The selected *a*-values are not intended to reflect the uncertainties y in the factor time series of secondary sulfate and sea salt during the fireworks period; we now assess these using a different method, as discussed below. Rather, the selected values are empirically chosen to provide a clean separation of these factors from the fireworks emissions. We performed a sensitivity analysis on the small *a*-value from 0 to 0.1 over partial (fireworks period only) time series of secondary sulfate and sea salt. The structure during

the fireworks period in the sea salt and the secondary sulfate factor time series was mixing with the fireworks factor time series for $a$-values > 0.01.

We have now added the following text on page 6 line 28:

The small $a$-value (0.01) for the sea salt and the secondary sulfate factor time series (for the fireworks period only) were estimated based on sensitivity analyses on the $a$-value from 0 to 0.1 with an increment of 0.01. The time series of both factors were showing fireworks peaks during the fireworks period for $a$-values greater than 0.01.

In the original manuscript, section 3.3 presented an analysis of the uncertainty of the time series across the entire campaign. However, this analysis did not account for the tight constraints applied to the sea salt and secondary sulfate factors during the fireworks period, and as a result these uncertainties were underestimated. We have now added the following text (page 7 line 12), which describes the uncertainty estimates of these factors during the fireworks period, and updated Fig. 3a accordingly.

During the non-fireworks period, uncertainties in the source apportionment results are assessed by a bootstrap analysis as described above. However, this approach cannot be used to assess uncertainties in the sea salt and secondary sulfate factors during the fireworks period, as during this period these factor time series are constrained with an artificially low $a$-value selected to optimize deconvolution. For these two factors, uncertainties during the fireworks period are determined by our ability to accurately predict the factor time series. The secondary sulfate and sea salt factors cases are discussed separately below.

Secondary sulfate concentrations during the fireworks period were estimated from the linear fit of the secondary sulfate factor to ACSM sulfate during the entire non-fireworks period. Uncertainties of ±5 % were calculated as the standard deviation of the actual secondary sulfate concentrations to the predicted values and included for the fireworks period only in Fig. 3a.

As described above, the sea salt factor time series during the four-day fireworks period was investigated to determine measurements that were affected or not affected by fireworks, where the measurements determined to be affected were replaced with a linear interpolation between the nearest good points. To determine the uncertainties of this approach, we applied this calculation to random segments of the non-fireworks data. Specifically, the four day-long sequence of affected/non-affected time points determined during the fireworks period was applied to a randomly chosen segment of data, and the standard deviation of measurement data to the estimated values calculated by interpolation was determined. This analysis was repeated for 38 randomly selected locations through the non-fireworks data, and a mean standard deviation of ±42 % was determined. This value is used as the uncertainty of the sea salt factor time series (during the fireworks period only) in Fig. 3a.

[Figure]

Figure 3: (a) Time series of the PM$_{10}$ elemental sources and relative contributions of the different sources over time; shaded areas indicate the uncertainties (interquartile) of selected bootstrap runs; grey background color represent the fireworks period; estimated uncertainty of the secondary sulfate (± 5%) and the sea salt factors (± 42%) during fireworks period are added as error bars; (b) Mean relative contributions of PM$_{10}$ elemental sources.

*Page 7, line 19: obvious. It was checked in the preliminary analysis at 9-factors and then constrained both as far profile and temporal patterns are concerned in the final solution.*

This factor (secondary sulfate) was identified in the unconstrained PMF solution at a specific seed where it was mostly dominated by S and had a high correlation with ACSM sulfate. However, the high and variable S concentrations during the fireworks period led to mixing between the fireworks and the secondary sulfate factors during this period. Therefore we constrained part of the secondary sulfate time series (only during fireworks days) to avoid mixing along with factor profile.

*Page 8, "Fireworks" paragraph. Considering that the fireworks profile was constrained, it sounds very strange that two fireworks sources were identified in the final analysis. As previously required, the existence of two fireworks sources should be supported by their identification at least in the unconstrained analysis of the fireworks period. If not, the strength of the imposed constrain should be verified to get if it artificially generates the presence of two factors.*

Based on the suggested changes due to the previous comments, we believe that we have responded to this concern as well. We have already discussed the residuals structure in Fig. S3 from 4 to 10 factors solution. In the final solution, the secondary sulfate, sea salt and fireworks-I factor profiles were constrained along with the factor time series (for the fireworks period only) of sea salt and secondary sulfate. Because of significant scaled residuals in the 7-factor solution, we found that fireworks related elements such as K, Ba, Ti, Cu, Bi had significant residuals during fireworks days (scaled residual over time: ±8) which is consistent with Fig. S8. Both analyses (residual and Fig. S8) indicate that the fireworks composition was highly variable and not well explained by a single factor.

We identified a K-rich factor in both the fireworks PMF and in the non-fireworks PMF analyses which were dominated by these elements along with other elements (Si, S, Fe, Mn, Zn, Pb). Although the second fireworks factor profile yielded a slightly higher K / S ratio (3.56) than the black powder ratio (2.76), we decided to keep it based on the improved residual structure and the factor time series which captured the fireworks variability.

*Page 9, lines 1-4 and Figure 4. Completely obscure. I interpret "normalized" as "divided by", but in this case how can the negative values be justifiable? Furthermore, what is "composition"? Average contribution of the factor to each element?*

The similar issue was also raised by Reviewer #1, and our response is presented in both places for clarity. The PMF results (Fig. 2) describe potentially complex and time-dependent sources as a single factor (or linear combination of factors). This is a fundamental limitation of the model, and may not accurately describe the behaviour or a complex source (e.g., residuals may be significant). In contrast, Fig. 4 provides an estimate of overall fireworks composition and temporal variability, complementing the PMF results (which are shown for reference). One can then see both the explanatory power and limitations of the SA model for this source.
We moved Fig. 4 to the supplement as Fig. S8 and removed ± SD from the figure as suggested by reviewer#1.
We added in the supplement on page 8 line 4:
Fig. S8 provides an estimate of the overall fireworks composition and temporal variability, complementing the PMF results (which are shown for reference). The figure is constructed in two stages. First, the time series of fireworks contributions to each element is estimated by subtracting the non-fireworks factors (NFF) from the original measurements. Then, the estimated fireworks contribution for each element is normalized by the total fireworks element contribution, and the displayed statistics are calculated. This is represented mathematically below, and the expression has been added to the main text. Note that the variation in fireworks profiles implied by this figure supports the representation of fireworks by 2 fireworks factors.

$$normalized\ concentration_{ij} = \frac{x_{ij} - (g_{ik}f_{kj})_{k=NFF}}{\sum_j (x_{ij} - (g_{ik}f_{kj})_{k=NFF})}$$

(S1)
where X represents the input data matrix for PMF, while G and F represent the factor time series and factor profiles driven by six non-fireworks factors (NFF). Here *i* and *j* denote time series and variables, respectively.
NFF and fireworks measurements have some uncertainties, and therefore for elements and/or time periods in which the concentration of fireworks is low, positive and negative values fluctuating around zero are expected. Here the negative values occur mostly for Si (for which fireworks are a minor fraction of the total signal), while the other elements were mostly captured very well. We used the factor composition for the factor profiles (fingerprints). We have already added the mathematical expressions for the factor composition as well as for the factor relative contribution on Page 7, lines 26-27.

*Page 11, lines 22-23: "We established that data sets including extreme events such as fireworks can be apportioned by ME-2 without disturbing the model solutions". Untrue. The imposed constraints completely modified the output of the unconstrained analysis.*

We have discussed the unconstrained nine-factor solution which was basically mixed with the fireworks in the factor time series during fireworks period. To avoid mixing, we used the ME-2 method by restricting the fireworks structure in the S and Cl driven factors time series.

We have modified the text as follows:

We show that the rotational control available in ME-2 provides a means for treating extreme events such as fireworks within a PMF analysis.

*Minor comments*

*Page 2, Line 12: A reference to: "Fe in brake lining can reach up to 60 % by weight" is needed.*

Done. (Chan and Stachowiak, 2004; Schauer et al., 2006)

Chan, D. and Stachowiak, G. W.: Review of automotive brake friction materials, Proc Inst. Mech. Eng. Part D: J Automob. Eng., 218, 953–966, https://doi.org/10.1243/0954407041856773, 2004.

Schauer, J. J., Lough, G. C., Shafer, M. M, Christensen, W. C., Arndt, M. F., DeMinter, J. T., and Park, J. S.: Characterization of emissions of metals emitted from motor vehicles, Research report (Health Effects Institute), 133, 1–76; discussion 77, 2006.

*Page 2, line 27: add "among others" after "Wang et al., 2018". Indeed, the list is far from being complete (see e.g.: Li et al. 2017 http://dx.doi.org/10.1016/j.jenvman.2017.02.059; Zhou et al., 2018 https://doi.org/10.5194/acp-18-2049-2018 among the most recent). If the authors intend providing a full list, a much more detailed research has to be done.*

Done.

*Page 3, lines 10-12. "The later, being essential in particular when separating extreme events such as fireworks which are most often excluded from the PMF input matrix (Ducret-Stich et al., 2013; Norris et al., 2014) to avoid distortion in the PMF solution due to unusually high emissions". This sentence is questionable. As reported by Paatero et al., (doi:10.5194/amt-7-781-2014) wrong decision on the outlier status of data can introduce serious modelling errors. Please also note that opposite to what stated by the authors, examples in the literature tried to exploit fireworks tracers to quantify the source contribution, with different approaches (e.g. Scerri et al., 2018 https://doi.org/10.1016/j.chemosphere.2018.07.104, Ji et al., 2018 https://doi.org/10.1016/j.scitotenv.2018.01.304, Vecchi et al., 2008 http://dx.doi.org/10.1016/j.atmosenv.2007.10.047)*

We modified the text in the manuscript as follows:

The rotational control available in ME-2 provides a means for treating extreme events such as fireworks within a PMF analysis. Such events are often excluded from the PMF input matrix to avoid modelling errors due to the pulling of a solution by outliers (Ducret-Stich et al., 2013; Norris et al., 2014; Paatero et al., 2014).

Regarding the papers mentioned by the reviewer, it was shown that fireworks are not necessarily outliers as considered in the PMF analysis. However, these articles have discussed different methods to apportion fireworks contribution in the PMF analysis e.g. Scerri et al. (2018) has down weighted some of the elements (not specific to fireworks-related elements) as "weak" variables without discussing the influence of fireworks on the other non-fireworks factors. The down weighting approach is not clear in this paper due to lack of criteria defining down

weighting random elements. In contrast Vecchi et al. (2018) reported the up weighting of fireworks tracer by a factor of 2 to highlight its role during fireworks. At the same time, they down weighted some of the variables by the factor of 2 to 4 with trial and error method until the model resolved fireworks sources. This approach may have worked to resolve fireworks contribution but no discussion has been provided for other non-fireworks sources. They discussed the other sources in a separate paper (Bernardoni et al., 2011) with a slightly different PMF input set on the same data by excluding fireworks-related tracers. Ji et al. (2018) quantified fireworks factor on the basis of signal to noise ratio instead of up/down weighing elements specific to fireworks which yielded the mixed factors with fireworks (in terms of factor profile and time series) in their PMF analysis. This mixing issue can be resolved by using ME-2 approach as adopted in our study.

Bernardoni, V., Vecchi, R., Valli, G., Piazzalunga, A., and Fermo, P.: $PM_{10}$ source apportionment in Milan (Italy) using time-resolved data, Sci. Total Environ., 409, 4788–4795, https://doi.org/10.1016/j.scitotenv.2011.07.048, 2011.

Ji, D., Cui, Y., Li, L., He, J., Wang, L., Zhang, H., Wang, W., Zhou, L., Maenhaut, W., Wen, T., and Wang, Y.: Characterization and source identification of fine particulate matter in urban Beijing during the 2015 Spring Festival, Sci. Total Environ., 430–440, 628–629, https://doi.org/10.1016/j.scitotenv.2018.01.304, 2018.

Scerri, M. M., Kandler, k., Weinbruch, S., Yubero, E., Galindo, N., Prati, P., Caponi, L., and Massabò, D.: Estimation of the contributions of the sources driving $PM_{2.5}$ levels in a Central Mediterranean coastal town, Chemosphere, 211 465–481, https://doi.org/10.1016/j.chemosphere.2018.07.104, 2018.

Vecchi, R., Bernardoni, V., Cricchio, D., D'Alessandro, A., Fermo, P., Lucarelli, F., Nava, S., Piazzalunga, A., and Valli, G.: The impact of fireworks on airborne particles, Atmos. Environ., 42, 1121–1132, https://doi.org/10.1016/j.atmosenv.2007.10.047, 2008.

*Page 7, line 28: "absolute mass". It should be recalled that it refers only to the mass related to elements, as no PM mass was inserted in the analysis.*

We have rewritten it as follows:
Fig. 3a shows the time series of the factors contributions in ng m$^{-3}$ (bottom panels) and relative contributions (top panel) of the retrieved $PM_{10}$ factors.

*Page 9, line 25-26: "The two dust factors together explain 95 % of Ca, with no other factor explaining more than 93 %". Obscure. If already explained at 95%, how can other factors explain Ca for more than 93%?*

We apologize for the misunderstanding. For clarity, we have rewritten it as:
The two dust factors together explain 95 % of Ca, while the remaining factors explain only 5% of Ca.

*Page 9, lines 29-30: "In general, Ca is commonly associated with mineral dust, construction activities, vehicular emissions and iron/steel plants". References are needed.*

Done.

(Lee and Pacyna, 1999; Vega et al., 2001; Bukowiecki et al., 2010; Crilley et al., 2016; Maenhaut, 2017)

Lee, D. S. and Pacyna, J. M.: An industrial emissions inventory of calcium for Europe, Atmos. Environ. 33, 1687–1697, https://doi.org/10.1016/S1352-2310(98)00286-6, 1999.

Crilley, L. R., Lucarelli, F., Bloss, W. J., Harrison, R. M., Beddows, D. C., Calzolai, G., Nava, S., Valli, G., Bernardoni, V., and Vecchi, R.: Source apportionment of fine and coarse particles at a roadside and urban background site in London during the 2012 summer ClearfLo campaign, Environ. Pollut., 220, 766–778, https://doi.org/10.1016/j.envpol.2016.06.002, 2016.

Vega, E., Mugica, V., Reyes, E., Sanchez, G., Chow, J. C., and Watson, J. G.: Chemical composition of fugitive dust emitters in Mexico City, Atmos. Environ., 35, 4033–4039, https://doi.org/10.1016/S1352-2310(01)00164-9, 2001.

Bukowiecki, N., Lienemann, P., Hill, M., Furger, M., Richard, A., Amato, F., Prévôt, A. S. H., Baltensperger, U., Buchmann, B., and Gehrig, R.: $PM_{10}$ emission factors for non-exhaust particles generated by road traffic in an urban street canyon and along a freeway in Switzerland, Atmos. Environ., 44, 2330–2340, https://doi.org/10.1016/j.atmosenv.2010.03.039, 2010.

Maenhaut, W.: Source apportionment revisited for long-term measurements of fine aerosol trace elements at two locations in southern Norway, Nucl. Instrum. Methods Phys. Res. Sect. B, 417, 133–138 https://doi.org/10.1016/j.nimb.2017.07.006, 2017.

---

## Referee Report (RR1)

The manuscript has improved considerably and can be published if the remarks listed below are implemented.

In the methodological section it must be clarified the total apportioned variable is the sum of the 14 trace elements. This is not obvious as SA studies, including some of those cited by the authors (Liu et al, 2019; Ji et al., 2018), most commonly apportion the total PM on the basis of elements and other chemical components. Similarly, in section 4.2, the sentence "In this section the final results of the elemental PM10 source apportionment are presented and validated" must be reworded to clarify the total variable is only the sum of the 14 elements used for the analysis. The same applies to Figure 2 where it must be stated that the average concentration is the sum of the elements.

Combining measurements obtained with different size cuts (in the fine fraction) introduces an additional degree of uncertainty but also gives the chance to handle a wider range of information at once. The additional uncertainty due to using different sizes could be modelled with PMF/ME-2. However, I agree that in this case it would be challenging to combine elemental data in the fine + coarse fractions with those of other components in the fine fraction only.

There are other recent source apportionment studies using Xact 620/625 data than the six listed in Page 3 line 11 (you find some suggested papers in the references below).

The discussion about road salt is more convincing now. However, the composition of the deicing products may vary considerably between countries. It would be more appropriate to check with European rather than US data.

The characterization of the industrial source is still rather vague. The authors should support their statement about the similarity with the profiles in the literature better. Two of the cited emission sources, smelter and coal combustion process, are pretty different from each other. In Ostrava, Vossler et al. identified two point sources associated with coal combustion: a power plant (Coal Power) and a "dirty" industrial profile (Industry 1) with high content of typical biomass burning components (K and Rb). None of these is a good representative of the industry sources.

The conclusion that traffic is the main source of PM10 elements in a traffic site is weak as outcome of a source apportionment study. I suggest putting more emphasis on the influence of other sources (fireworks, as suggested by the authors) or background levels on the elemental composition at a traffic site.

References

Sofowote U.M., Healy R.M., Su Y., Debosz J., Noble M., Munoz A., Jeong C.-H., Wang J.M., Hilker N.,. Evans G.J, Hopke P.K., 2018. Understanding the PM2.5 imbalance between a far and near-road location: Results of high temporal frequency source apportionment and parameterization of black carbon, Atmospheric Environment, Volume 173, 277-288.

Jeong, C.-H., Wang, J.M., Hilker, N., Debosz, J., Sofowote, U., Su, Y., Noble, M., Healy, R.M., Munoz, T., Dabek-Zlotorzynska, E., Celo, V., White, L., Audette, C., Herod, D., Evans, G.J., 2019. Temporal and spatial variability of traffic-related PM2.5 sources: Comparison of exhaust and non-exhaust emissions. Atmospheric Environment 198, 55–69. https://doi.org/10.1016/j.atmosenv.2018.10.038. (already cited in page 2 line 25)

Belis, C.A., Pikridas, M., Lucarelli, F., Petralia, E., Cavalli, F., Calzolai, G., Berico, M., Sciare, J., 2019. Source apportionment of fine PM by combining high time resolution organic and inorganic chemical composition datasets. Atmospheric Environment, 116816. https://doi.org/10.1016/j.atmosenv.2019.116816.

---

## Author Response (AR2)

**Comments to the author by Co-Editor:**

*Three Anonymous Referees provided comments for your ACPD manuscript; two of them requested major revisions and the third one requested rejection. Therefore, I nominated the same three referees again to have a look at your revised ACPD submission. However, one of the three (who had requested major revisions) missed the nomination deadline. The other two provided comments; the one who had requested major revisions now requests minor revisions, but the one who recommended rejection still asks for rejection. I therefore invited a new third Referee; he/she requests major revisions.*

*Considering the comments of the three Referees for your revised ACPD submission, I request major revisions for it. I suggest that you take the comments of the three Anonymous Referees seriously into consideration and that you submit a newly revised manuscript.*

Dear Prof. Dr. Maenhaut,

Thank you for giving us this opportunity to resubmit our manuscript. We appreciate the comments of the three anonymous referees to improve this paper, and have prepared our responses accordingly.

**Anonymous Referee #2**

**Response to Reviewer #2**

We kindly thank the referee for considering our manuscript and we value the comments raised to improve the manuscript. A point-to-point response (in regular typeset) to the reviewers' remarks (in italic typeset) follows. Changes to the manuscript are indicated in blue font. In the following, page and lines references refer to the previous version of the manuscript reviewed by anonymous referee #2.

*The manuscript has improved considerably and can be published if the remarks listed below are implemented.*

**Comment #1**

*In the methodological section it must be clarified the total apportioned variable is the sum of the 14 trace elements. This is not obvious as SA studies, including some of those cited by the authors (Liu et al, 2019; Ji et al., 2018), most commonly apportion the total PM on the basis of elements and other chemical components. Similarly, in section 4.2, the sentence "In this section the final results of the elemental PM10 source apportionment are presented and validated" must be reworded to clarify the total variable is only the sum of the 14 elements used for the analysis. The same applies to Figure 2 where it must be stated that the average concentration is the sum of the elements.*

We agree with reviewer #2 and we have modified the text in Abstract as follows:

We conducted source apportionment by positive matrix factorization (PMF) of the elemental mass measurable by the Xact (i.e., major elements heavier than Al), defined here as $PM_{10el}$.

Eight different sources were identified in $PM_{10el}$ (elemental $PM_{10}$) mass driven by the sum of 14 elements.

We also added the following line in section 4.2 (page 11 line 4):
In this section, the results of the $PM_{10el}$ mass driven by the sum of 14 elements are presented and validated.

Accordingly, we modified Figure 3 caption as follows:

[Figure]

**Figure 3: (a) Time series of the $PM_{10el}$ sources and relative contributions of the different sources over time; shaded areas indicate the uncertainties (interquartiles) of selected bootstrap runs; grey background color represents the fireworks period; the estimated uncertainties of the secondary sulfate (± 5%) and the sea salt factors (± 42%) during the fireworks period are added as error bars; (b) Mean relative contributions of $PM_{10el}$ sources. The average concentration represents the mean value of apportioned sources in $PM_{10el}$ which is the sum of 14 elements.**

**Comment #2**
*Combining measurements obtained with different size cuts (in the fine fraction) introduces an additional degree of uncertainty but also gives the chance to handle a wider range of information at once. The additional uncertainty due to using different sizes could be modelled with PMF/ME-2. However, I agree that in this case it would be challenging to combine elemental data in the fine + coarse fractions with those of other components in the fine fraction only.*

We appreciate the reviewer's feedback on this issue.

**Comment #3**
*There are other recent source apportionment studies using Xact 620/625 data than the six listed in Page 3 line 11 (you find some suggested papers in the references below).*

Done.

**Comment #4**

*The discussion about road salt is more convincing now. However, the composition of the deicing products may vary considerably between countries. It would be more appropriate to check with European rather than US data.*

We have adopted the European deicing salt composition. We modified the text as follows:

The measured Cl / Ca ratio (0.33) from Xact does not lie in the range of de-icing salt composition (5.27) measured in northern Germany (Pernigotti et al., 2016).

**Comment #5**

*The characterization of the industrial source is still rather vague. The authors should support their statement about the similarity with the profiles in the literature better. Two of the cited emission sources, smelter and coal combustion process, are pretty different from each other. In Ostrava, Vossler et al. identified two point sources associated with coal combustion: a power plant (Coal Power) and a "dirty" industrial profile (Industry 1) with high content of typical biomass burning components (K and Rb). None of these is a good representative of the industry sources.*

We agree with reviewer #2 for concerning the wish for better literature to support the industrial factor. Since different workshops and ateliers surround the sampling site, small contributions from metal workshops can be expected. However, we have already mentioned some other references (Crilley et al., 2016; Richard et al., 2011) in the manuscript which have likewise retrieved a similar source profile without a direct link to a specific point source.

We have modified the text by removing the above citations (Amato et al., 2010; Dall'Osto et al. 2013; Vossler et al., 2016).

**Comment #6**

*The conclusion that traffic is the main source of PM10 elements in a traffic site is weak as outcome of a source apportionment study. I suggest putting more emphasis on the influence of other sources (fireworks, as suggested by the authors) or background levels on the elemental composition at a traffic site.*

We added the following text in the conclusions (page 15 line 29):

The outcome of this study emphasizes the significant influence of regional background secondary sulfate and local background dust apart from non-exhaust traffic emissions at the sampling location.

**Anonymous Referee #3**

**Response to Reviewer #3**

We kindly thank the referee for considering our manuscript and we value the comments raised to improve the manuscript. A point-to-point response (in regular typeset) to the reviewers' remarks (in italic typeset) follows. Changes to the manuscript are indicated in blue font. In the following, page and line references refer to the previous version of the manuscript reviewed by anonymous referee #3.

*The authors of the paper ACP-2018-1229 performed a huge job to revise the paper and they partially answered the concerns risen during the discussion phase. The reviewer appreciated their efforts, which gave a better explanation of data treatment helping the reader to follow the analysis. Nevertheless, in the reviewer's opinion the procedure presented here is not fully convincing and it is still affected by a degree of subjectivity whose impact cannot be evaluated. Thus, in the reviewer's opinion it does not meet the scientific standards for publication in Atmospheric Chemistry and Physics.*
*More in detail:*

**Comment #1**
*The authors define the fireworks hour FH as July 31st 21:00 to August 1st 7:00. They run PMF on such database to constrain fireworks profile. The period includes 10 hours – i.e. 10 time slots. The authors state that they performed PMF on such data. At the reviewer's knowledge, 14 variables and 10 time slots used to resolve 5 factors by PMF should lead to scarcely reliable results due to lack of statistics (see https://source-apportionment.jrc.ec.europa.eu/Docu/european%20_guide_SA_RMs_revision_2019.pdf). Please consider that the results of the analysis are used to constrain fireworks profile in the final solution. Whether deriving constraints within a dataset is in principle possible (as proposed by Sofowote et al. 2015, http://dx.doi.org/10.1016/j.atmosenv.2015.02.045) in that case a very high number of samples was available to identify the constraints.*

We agree that the appropriateness of an input dataset for PMF analysis is an important consideration. The number of time points required for such analysis is strongly dataset-dependent. To quote the PMF guidelines cited by the reviewer (https://source-apportionment.jrc.ec.europa.eu/Docu/european%20_guide_SA_RMs_revision_2019.pdf, p. 37):

> "*In practice, the minimum number of samples required to detect the latent variables cannot be established a priori as it depends on the amount of information contained in the dataset. If the relative contribution of sources were the same in all samples, analysing new samples would not add any new information to the model. Therefore, there should be enough samples to catch the variability of the sources, including samples where some sources are absent or negligible.*"

This excerpt highlights the fact that the number of points itself is not the determining factor, but whether the dataset contains sufficient variability for the factors to be distinguished. The approximate rules for dataset size referenced by the reviewer are developed for application to typical ambient datasets. Such datasets are fundamentally different from analysis of the FH period, where the input PMF matrix is selected specifically to maximize the variability of a specific factor. As a result, far fewer time points are required to resolve the dominant (fireworks) factor.

This does still leave the question of how many points are too few. We test this issue with a bootstrap (BS) analysis on the FH dataset, which resamples 11 data points randomly by repeating or removing data points from the original 11 data points. If number of time points in the dataset were too small, one would expect a high degree of variability in the output solutions. The stability of the fireworks factor profile was assessed by investigation of 1000 random BS runs on the FH dataset (see more details below). Solutions were accepted based on a K / S ratio in the fireworks factor of $2.76 \pm 0.5$, consistent with black powder composition (Dutcher et al., 1999) and the concentration peak between 40-45 µg m$^{-3}$ on 1 August 23:00 LT in the factor time series. This criterion yielded an acceptance rate of 21.3 %, with an average $Q/Q_{exp}$ of 1.3. The results of the BS analysis are provided in Fig. S6, where we show the fractional composition of the fireworks factor profiles from: (a) FH BS analysis (blue color); (b) 5-factor constrained FH PMF analysis (base case, in red color). The data and their corresponding uncertainty are given as box-whisker plot (bottom to top: p10-p25-p50-p75-p90) of selected 213 solutions out of 1000 BS runs. The displayed error bars for the base case correspond to the a-value of 0.1 used to constrain fireworks profile in the complete dataset PMF analysis. This figure shows that the variation in fireworks profiles retrieved from the bootstrap analysis is relatively stable despite the small number of data points; indeed, it generally denotes a smaller range of values than those allowed by the constrained fireworks profile in the full dataset PMF. Further, the resolved fireworks profile from FH BS solution agrees well with the base case, and the $Q/Q_{exp}$ values are reasonable. Taken together, these results suggest that the FH dataset contains sufficient variability to allow retrieval of a robust fireworks profile.

We fully agree that *if* analysis of the FH period were intended to quantitatively resolve *non-fireworks* factors, the number of time points in the dataset would be almost certainly insufficient. However, extraction of the fireworks factor from the specially constructed FH dataset is a different case that does not fall within the class of datasets considered in formulating general rules for PMF. Indeed, our FH analysis is conceptually similar to a simple subtraction of background aerosol from a plume in order to obtain the profile of the plume emissions source; the implementation of such a subtraction through PMF merely allows the subtraction to be more adaptive. This concept also motivates the independent estimation and constraint of the sea salt profile and time series during FH analysis, which further reduces the number of points required to resolve the FH profile.

Finally, we note that the general strategy of deriving an anchor profile from analysis of selected time points from a full dataset has been performed successfully on several occasions. Most similar to the current study are the examples of Froehlich et al. (2015), in which short-duration spikes in organic aerosol concentration were combined into a sub-dataset to determine an

anchor profile related to local cigarette smoke, and Visser et al. (2015), in which a subset of a trace element dataset with high residuals was analysed separately to retrieve a factor profile related to industrial emissions.

We agree that it is important for the reader to be aware of the fundamental differences between analysis of a typical ambient dataset and a constructed one such as FH. Therefore we have added the following paragraph to section 3.2 (Page 8 line 31):

Note that the number of time points contained in the input matrices for the data subsets (as opposed to the full dataset) are in some cases smaller than typical recommendations for ambient PMF (Olivier et al., 2019). This is most extreme in the case of the FH dataset, where only 11 time points are used. However, there are two important differences between PMF analyses of these sub-datasets and typical ambient PMF: (1) the sub-datasets are constructed to maximize variability of a factor or set of factors; and (2) we are concerned only with accurately characterizing the profile(s) of these selected major factors. These two points work together to greatly reduce the number of time points required for the analysis (Fig. S6). A similar approach has been successfully applied by Froehlich et al. (2015), in which short-duration spikes in organic aerosol concentration were combined into a sub-dataset to determine an anchor profile related to local cigarette smoke, and Visser et al. (2015), in which a subset of a trace element dataset with high residuals was analysed separately to retrieve a factor profile related to industrial emissions.

We added the following text and figure in the supplement (page 2 line 3):
The stability of the fireworks factor profile was assessed by investigation of 1000 random BS runs on the FH dataset (see more details below). Solutions were accepted based on a K / S ratio in the fireworks factor of $2.76 \pm 0.5$, consistent with black powder composition (Dutcher et al., 1999) and the concentration peak between 40 - 45 µg m$^{-3}$ on 1 August 23:00 LT in the factor time series. This criterion yielded an acceptance rate of 21.3 %, with an average $Q/Q_{exp}$ of 1.3. The results of the BS analysis are provided in Fig. S6, where we show the fractional composition of the fireworks factor profiles from: (a) FH BS analysis (blue color); (b) five-factor constrained FH PMF analysis (base case, in red color). The data and their corresponding uncertainty are given as box-whisker plot (bottom to top: p10-p25-p50-p75-p90) of selected 213 solutions out of 1000 BS runs. The displayed error bars for the base case correspond to the a-value of 0.1 used to constrain fireworks profile in the complete dataset PMF analysis. This figure shows that the variation in fireworks profiles retrieved from the bootstrap analysis is relatively stable despite the small number of data points; indeed, it generally denotes a smaller range of values than those allowed by the constrained fireworks profile in the full dataset PMF. Further, the resolved fireworks profile from FH BS solution agrees well with the base case, and the $Q/Q_{exp}$ values are reasonable. Taken together, these results suggest that the FH dataset contains sufficient variability to allow retrieval of a robust fireworks profile.

[Figure]

**Figure S6: Fireworks factor profiles from: (a) FH BS analysis (blue color); (b) 5-factor constrained FH PMF analysis (base case, red color). The data and their corresponding uncertainties are given as box-whisker plot (bottom to top: p10-p25-p50-p75-p90) of selected 213 solutions from 1000 BS runs in blue color. Base case uncertainties are given ± 10 % as error bar by considering that *a*-value 0.1 is applied to constrain fireworks profile in complete dataset PMF analysis. Y-axis represents the fractional composition of the factor profile in row-wise for each factor in ng ng$^{-1}$.**

**Comment #2**

*No discussion on the Q increase due to the constraints is carried out. Opposite, this should be used to support the reliability of the imposed constraints. Excessive increase of Q may indicate an excess of subjectivity in the solution (Paatero and Hopke, 2009 DOI: 10.1002/cem.1197).*

We agree that the effect of the constraints on the residuals for the FH dataset should be addressed. This can be done in a broad sense by investigation of constraint effects on $Q$, as suggested by the reviewer. However, because the purpose of this analysis is not a full description of the FH dataset but rather the fireworks component, it makes sense to also investigate the explained variation (EV) and unexplained variation (UEV) on an element-by-element basis. Finally, an additional consideration is that the application of constraints changes which factors are resolved and in what order as the number of factors in a solution increases. Specifically, while the five-factor constrained FH solution contains secondary sulfate, sea salt, fireworks, dust-related, and mixed K-rich/traffic factors, the five-factor unconstrained FH solution contains three factors related to fireworks, 1 mixed fireworks/traffic factor, and a dust-related factor. This significantly decreases the residuals but would be rejected in any normal PMF analysis due to a non-interpretable explanation of the data (i.e., excessive factor splitting/mixing). We therefore compare three solutions: (1) constrained five-factor FH, used in the paper to derive the fireworks profile; (2) unconstrained five-factor FH, and (3) unconstrained three-factor FH (which represents the most interpretable unconstrained FH solution.

As shown in Fig. S4, the constrained five-factor $Q/Q_{exp}$ is 1.5, which is ~3 times higher than the unconstrained 5-factor $Q/Q_{exp}$ but same as unconstrained five-factor $Q/Q_{exp}$ (1.52). The

increase in $Q$ is significant over the constrained five-factor FH PMF and unconstrained three-factor because its UEV for each variable is between 2 % to 16 % (average ~10 %), while is it between 1.5 % to 15 % (average ~5 %) (see Fig. S5). The three-factor unconstrained FH PMF analysis validates the five-factor constrained FH PMF analysis, which suggests that both methods are working fine, and gives us more confidence to identify the fireworks factor profile.

We modified the text on page 8 line 30 as follows:

The unconstrained FH PMF analysis was also performed for two to five factors with different seeds. As shown in Fig. S4, the constrained five-factor $Q/Q_{exp}$ is 1.5, which is ~3 times higher than the unconstrained five-factor $Q/Q_{exp}$ but same as unconstrained 3-factor $Q/Q_{exp}$ (1.52). The increase in $Q$ is significant over the constrained five-factor FH PMF and unconstrained five-factor because its UEV for each variable is between 2 % to 16 % (average ~10 %), while is it between 1.5 % to 15 % (average ~5 %) (see Fig. S5). The three-factor unconstrained FH PMF analysis validates the five-factor constrained FH PMF analysis, which suggests that both methods are working fine, and gives us more confidence to identify the fireworks factor profile.

We have added the following text and figures in the supplement on page 2 line 3:

The unexplained variation (UEV, Paatero et al., 2004; Canonaco et al., 2013) which is a dimensionless quantity, describes how much of the measured variation (time or variable) is not explained by each PMF factor (Eq. S1, for variable $j$):

$$UEV_{j,k,real} = \frac{\sum_{i=1}^{n}(|e_{i,j}|/s_{i,j})}{\sum_{i=1}^{n}\left(\left(\sum_{k=1}^{p}|g_{i,k}f_{k,j}|+|e_{i,j}|\right)/s_{i,j}\right)} \quad \text{for k = 1, ......, p for } (x_{i,j}/s_{i,j}) > 2 \qquad (S1)$$

Where $f_{k,j}$ are the factor profiles and $g_{i,k}$ represent their time-dependent contributions. The index $i$ represents a specific point in time (up to $n$), $j$ is the variable, and $k$ is a factor (up to $p$). $e_{i,j}$ are the PMF-residuals, $x_{i,j}$ the input data and $s_{i,j}$ the measurement uncertainties. The UEV presented here is the real UEV for high S/N threshold >2, otherwise UEV will be considered as noisy UEV.

[Figure]

**Figure S4: $Q/Q_{exp}$ as a function of the number of factors for FH PMF analyses.**

[Figure]

**Figure S5: Unexplained variation for each variable in FH PMF solutions: (1) constrained 5-factor (black color); (2) unconstrained 5-factor (semi filled blue color); and (3) unconstrained 3-factor (blue color).**

**Comment #3**

*Implemented constraints e.g. to FD and FH analyses arise from different solutions from the same dataset. It is noteworthy that 9-factor unconstrained PMF on NFD was used to constrain the sulphate factor, whereas sea-salt was constrained using the 7-factor solution from the same dataset. In the reviewer's opinion, only one solution from the same dataset should be used as reference for constraints, as the sea-salt solution could have been impacted by the increase of factors needed to derive the sulphate one. Furthermore, the authors performed also background concentration interpolation of the concentrations before and after fireworks to obtain background levels during the episode. The absence of a detailed analysis of the Q-value increases compared to FD and FH unconstrained solutions do not allow to detect the degree of subjectivity in all these assumptions.*

We believe the reviewer's concern lies with the supposed derivation of factor constraints from two solutions to the same dataset having different numbers of factors. However, this is not done in the current analysis. On page 7 line 25, we state that the secondary sulfate factor was derived from a 9-factor unconstrained PMF on the complete dataset, whereas the sea salt profile was derived from a 7-factor unconstrained PMF solution from the NFD (non-fireworks days) dataset. However, we found a typo mistake on page 8 line 11, which should be "complete data set" instead of "NFD dataset". This may have created misunderstanding to the reviewer. We apologise for that. Because these analyses are performed on different datasets, the set of factor(s) that can be cleanly extracted from each analysis also differs.

Based on the suggested changes due to comment #2, we believe that we have also answered the last part of the comment (*Q*-value for FH PMF). Since we did not use any solution from FD (fireworks days) PMF analysis, we do not want to discuss its *Q*-value in the manuscript. However, we have mentioned in the manuscript that we performed PMF on this dataset (test

case) to resolve a fireworks profile, but we were not able to resolve a clean fireworks factor profile.

**Comment #4**

*The authors report scaled residuals for the sulphate source in the range [-0.45, 0.85] for the unconstrained 9-factor NFD analysis: it is asymmetric and far from expected [-3, 3]. This unique factor merit to be commented more in detail.*

Although the range of reported S scaled residuals [-0.45, 0.85] appears asymmetric, these endpoints are merely the minimum/maximum values recorded and do not reflect the shape of the overall distribution. The scaled residual distribution is nearly symmetric. This distribution has been added to the manuscript as Fig. S3 (and included below). We agree that the range of observed values for the scaled residuals are surprisingly close to zero. However, this is consistent with the overall $Q/Q_{exp}$ (0.4) rather than a specific feature of the S variable, and suggests that the Xact uncertainties are systematically overestimated. As discussed in the text, we currently utilize a generic error model, although development of an instrument-specific formulation (that might give more accurate results) would certainly be of interest in the future

We have modified the text in the manuscript on page 7 line 28 as follows:

The scaled residual (over the time series) of S in this solution was within the range of ± 3 with very small values, as shown in Fig. S3. The strong influence of fireworks in this solution yielded small values of the scaled residual for S which are consistent also with lower-than-expected $Q/Q_{exp}$ (0.4) values, suggesting an overestimation of uncertainties by the generic error model used herein (see Eqs. 6 and 7).

[Figure]

**Figure S3: Histograms of the scaled residuals ($e_{ij}/s_{ij}$) for S in the nine-factor unconstrained PMF solution on the complete dataset.**

**Comment #5**

*Performing ten pseudorandom run is quite limited to explore possible multiple minima. Especially for the chosen solutions, the reviewer would expect more runs (30-50). Moreover,*

*it is now evident from the text that only in one case out of ten the authors were able to identify the unique factor sulphate mentioned above. Nevertheless, no comments on being the solution with the lowest Q or not. If not, what is the value compared with the lowest Q obtained in the ten pseudorandom runs?*

We agree, and have extended this analysis to 100 random seeds. We identified the secondary sulfate factor profile for 24 seeds out of 100 seeds. Note that the fraction of seeds yielding a given solution is unrelated to the mathematical quality or environmental interpretability of the solution. The distributions of the $Q/Q_{exp}$ for 24 seeds (where the secondary sulfate factor was resolved) and the remaining 76 seeds are shown in Fig. S2 below. The distributions are not significantly different.

We have modified the text in manuscript on page 7 line 26 as follows:

The identified secondary sulfate factor time series correlated very well with ACSM sulfate ($r^2$ = 0.91) (Fig. S1) at 24 seeds out of 100 seeds while $r^2$ was ≤ 0.88 for the remaining 76 seeds. The $Q/Q_{exp}$ values for all 100 seeds were in the range of 0.41- 0.45 (Fig. S2).

[Figure]

**Figure S2: Box-whisker plots of $Q/Q_{exp}$ for the seed analysis of the 9-factor unconstrained PMF on the full dataset. Shown separately are the 24 solutions in which a secondary sulfate (SS) factor was identified ("SS identified") and the 76 solutions where it was not ("SS not identified").**

**Comment #6**

*Mass apportionment: the reviewer is still convinced that an attempt to insert PM10 TEOM data to provide more information on PM10 source apportionment should be performed and commented.*

The reviewer's suggestion could be a valid approach if $PM_{10}$ were dominantly composed of sources identifiable from PMF analysis of the Xact data. However, this is clearly not the case. For example, secondary organic aerosol, a major $PM_1$ constituents, comprises a significant fraction (~60%) of $PM_{10}$ (Richard et al., 2011), and has no direct relationship to any of the

resolved sources. This is also the case for $NO_3^-$ and primary organic aerosol from biological and cooking sources, while major constituents such as $NH_4^+$, black carbon, and other primary organics derive from sources that are only partly represented by species measured by the Xact (e.g., $NH_4^+$ is related to both S, which the Xact can measure, and $NO_3^-$, which it cannot). Unlike in chemical mass balance, where the ratios of the apportioned quantity to tracers are predetermined and any residual is deposited in its own factor, the PMF model will attempt to maximize apportionment of $PM_{10}$ across all factors. In the present case, where it is expected that $PM_{10}$ contains a significant contribution from sources to which the Xact is entirely blind, such an analysis will yield incorrect results. Further, it is not possible to assess the nature of the biases, e.g. whether the erroneously included $PM_{10}$ mass were to be apportioned uniformly across all factors, predominantly to a single factor, or in a time-dependent way. As a result, we do not consider the addition of $PM_{10}$ mass to the Xact PMF matrix to be a valid analysis strategy.

One could argue that the Xact and ACSM data should be combined in a PMF analysis. While we agree this would be a potentially useful strategy, it is also highly complex as demonstrated by previous combined PMF analyses (e.g. Slowik et al., 2010; Crippa et al., 2013). Given the already complex nature of the Xact-only PMF, we contend that this would need to be addressed in a standalone manuscript and is beyond the scope of the current study.

**Anonymous Referee #4**

**Response to Reviewer #4**

We kindly thank the referee for considering our manuscript and we value the comments raised to improve the manuscript. A point-to-point response (in regular typeset) to the reviewers' remarks (in italic typeset) follows. Changes to the manuscript are indicated in blue font. In the following, page and line references refer to the previous version of the manuscript reviewed by anonymous referee #4.

**Comment #1**
*Regarding the manuscript entitled "Source apportionment of highly time resolved trace elements during a firework episode from a rural freeway site in Switzerland". The study is very interesting as the authors are using an approach that it is commonly used for the organic fraction of PM to a "portion" of the inorganic fraction of PM. The methodology that is used is thoroughly explained, the study is brilliantly written and the results are interesting. As it happens in every case that a new approach is tried, there are some fundamental assumptions that the authors need to make in order to use it. In my opinion there are some important questions to be answered before the study is published. I understand that the authors aim to perform source apportionment analysis not on the full PM10 mass, but on the identified/used elements only. My first minor comment regarding this, is that it should be made very clear from the beginning of the manuscript and in the Abstract. In line 14 of the Abstract, the term "elemental PM10" is used to probably describe that approach. In my opinion this term is unclear and wrong. In line 17 the term "PM10 elemental*

*mass" is used, which is much more accurate but still unclear. It must be stated clearly that the work is representative of PM10 mass that corresponds to the used variables (around 20% of the total).*

We agree with reviewer #4 that our terminology may be misleading. We have modified it on page 1 line 12 as follows:

We conducted source apportionment by positive matrix factorization (PMF) of the elemental mass measurable by the Xact (i.e., major elements heavier than Al), defined here as $PM_{10el}$. Eight different sources were identified in $PM_{10el}$ (elemental $PM_{10}$) mass driven by the sum of 14 elements.

**Comment #2**
*My main concern and the question that rises, is even when the target is clear, is it possible to perform source apportionment analysis without major PM components? In principle you can perform SA in any type of dataset and the results will be representative of that mass. As the authors state correctly, when you do SA in organics, the results are representative for the organic mass only. The problem is that when the organics are used, all the possible (or at least most of them) organics compounds are taken into account. Here only a small fraction of the inorganics is used. My concern in this case, is not (only) the mass that it is missing when the ions and EC are not taken into account, but the tracers that are not present. For example, you can run PMF for the organics without m/z 43, 44, 55, 57 and 60 (or some of them), you still have enough mass to apportion, but you are missing key tracers that will help resolve the factors. It is the same as trying to identify traffic without EC. The authors should thoroughly discuss those limitations in the manuscript and explain why by the way the methodology is implemented the limitations are resolved. For example, in the conclusions the authors state that "We show that the rotational control available in ME-2 provides a means for treating extreme events such as fireworks within a PMF analysis". Even though that is true, the authors need to explain that the fact they have to resort in so complicated methodology to get a good solution, is not only due to the nature of the environmental situation at hand, but because of the lack of data. I strongly believe that if the ions and EC/OC were available the situation would be much simpler. It is extremely important to give the right message to the readers, that they need to use full and complete datasets in order to perform good source apportionment analysis. The authors may have the experience and expertise to work around it, but this fact must not be used as an example.*

We agree with the reviewer on the need to clearly define the quantity being apportioned, which we have done in response to the previous comment. In regards to the reviewer's point about source resolution, we note that the key factor is the inclusion of measured variables that fully represent source variability (e.g., source-specific tracers) rather than the amount of total mass captured. On the other hand, including variables that are only partially representative of the key sources in a dataset can impede source apportionment.

For example, the reviewer suggests to include EC to improve separation of the traffic factor. However, we note that EC is emitted directly from the tailpipe, whereas the traffic factor

identified herein is attributed primarily to break wear and, crucially, is significantly influenced by resuspension. Thus inclusion of EC would be expected to either (1) degrade separation of the current traffic factor or (2) form part of an additional factor not retrievable in the current dataset. Option (2) would be an attractive outcome, but would be best implemented in the context of an integrated source apportionment of data from multiple instruments, which is highly complex and beyond the scope of the current manuscript.

Overall, we find that the currently retrieved factors are well matched with the measured variables, with the following exceptions:

1. Sea salt would benefit from inclusion of Na.
2. Fireworks would benefit from the measurement and inclusion of Sr.

We have added the following line in conclusions (page 15 line 21):

The source apportionment model performance could possibly be additionally improved by the inclusion of Na and Sr to better resolve the sea salt and fireworks factors, respectively.

**Comment #3**
*My other fundamental concern is the use of FH PMF resolved profile for constraining the firework profile. My problem is the number of "samples" that is used to run PMF is this case (for FH). How many samples are used? Are there enough to even run PMF let alone have good statistics? It has been proved that the suggested methodology works brilliantly for a usual organic dataset where thousand of measurements are available, but I am not certain that it works just as well with 500 samples (when you split the dataset and therefore reduce the variability) and I am certain that it does not work with less than 50. One can argue that this is a PMF/CMB hybrid and in CMB you can run the model successfully with one sample in principal. The problem is that in CMB external information is used to constrain the factors (regarding the source profiles), while in this case the information is extracted by the dataset itself, which in a way recycles the uncertainty.*

This issue was addressed in response to reviewer #3 (comment #1) and we refer the reader to our response there.

**Comment #4**
*The naming of the "Road dust" and "Traffic related" factors is wrong. The authors describe the Traffic related factor as "Coarse particles from brake/disc wear..." which is the exact definition of road dust! On the other hand, they describe "Road dust" as "The sampling site is located close to the freeway and must be influenced by wear and tear of the asphalt/concrete roads because of heavy traffic". I strongly suggest that the "Traffic related" factor is renamed to Road dust and the factor named "road dust" should be named "road abrasion". The are two types of traffic related sources: road dust and exhaust emissions. Since the factor that the authors name "road dust" cannot be exhaust emissions (there is no EC and authors describe it differently), it must be road dust. The naming makes much more sense that way since in this*

*case Road dust contains actual tracers of road dust such as Zn and Cu, and "road abrasion" can justify the presence of only Ca in the factor.*

We appreciate the concern raised by reviewer #4. However, we do not agree with reviewer for renaming the traffic-related factor as road dust. The factor profile of traffic-related (Fe, Cr, Cu, Zn, Mn, Zn, Ba) is attributed to non-exhaust traffic emissions such as: brake/disc wear, brake lining, tire wear, brake pads etc. This factor profile contains negligible contributions of crustal elements such as Ca and Ti, which indicates that the traffic-related factor is not influenced by dust-related sources. In the absence of size-segregated elemental PM data, we cannot draw conclusions about coarse or fine particles for these elements. On the other hand, brake wear/pads related elements have been found to be predominately in the coarse fraction (Bukowiecki et al., 2009; Amato et al., 2014; Harrison et al., 2012) which brackets the reference statement "Coarse particles from brake/disc wear could appear as flakes and mainly consist of iron oxides (Wahlström et al., 2010)".

For clarity, we therefore rename the "traffic-related" factor as "non-exhaust traffic-related". The factor named road dust represents the resuspension of particles emitted from road wear/tear, vehicle wear/tear and/or mineral dust during traffic rush hours. Yet, the factor profile of the road dust is different from the traffic-related factor profile, indicating resuspension of mineral dust and/or road wear/tear particles. The separation of road wear/tear and mineral dust profiles is often difficult due to the deposition of road wear/tear particles on the road surface, which are then resuspended together with deposited mineral dust. The high contribution of Ca alone (> 80%) has been seen in previous studies (Ducret-Stich et al., 2013; Bukowiecki et al., 2010; Hueglin et al., 2005) where it was named as "road dust". Another study at a rural site in Switzerland also found the Ca rich factor in the coarse fraction where it was linked to soil resuspension (Minguillon et al., 2012). The composition of road dust has been found to be dominated by elements and compounds typically associated with crustal material. The composition, therefore, often reflects the local geology and subsequently varies greatly with location. All these studies (except Minguillon et al., 2012) reported the road dust composition at various locations in Switzerland.

We have replaced "traffic-related" with "non-exhaust traffic-related" in the manuscript and the supplement. We have added the text in manuscript on page 13 line 29 as follows:

Road dust profiles are often difficult to identify due to resuspension of materials deposited on the road surface such as mineral dust, vehicle wear/tear, and/or road surface wear/tear. However, the road dust factor profile is distinct from the non-exhaust traffic-related factor profile. It is possible that the road dust factor is related to resuspension of mineral dust and road wear/tear particles. The high contribution of Ca alone (> 80%) has been seen in previous studies (Ducret-Stich et al., 2013; Bukowiecki et al., 2010; Hueglin et al., 2005) where it was named as "road dust". Another study at a rural site in Switzerland also found the Ca rich factor in the coarse fraction where it was linked to soil resuspension (Minguillon et al., 2012).

**Comment #5**

*"The time series of secondary sulfate exhibits peaks during the fireworks event". Does this not mean that the factors are somewhat mixed? Secondary sulfate should not be related with fireworks. Please comment about that in the manuscript.*

We agree with reviewer #4 that the secondary sulfate factor yielded peaks during fireworks event. Inorganic gases ($SO_2$, $NO_x$, etc.) emitted during the fireworks events may be oxidized to secondary organic and inorganic components that may condense to the particle phase (Sarkar et al., 2010) within a very short span of time, as observed in previous studies (Wang et al., 2007; Yang et al., 2014). We monitored two parameters to explain the influence of fireworks in secondary sulfate time series: (1) mass balance for ACSM $PM_1$ data; (2) secondary sulfate peaks during the main fireworks hours (1 August 23:00 LT to midnight). The equivalent concentration of ammonium ($NH_{4eq}$) balances the sum of $NO_{3eq}$ and $SO_{4eq}$ during non-fireworks periods, while during fireworks peaks, the balance shifts towards the sum of $NO_{3eq}$ and $SO_{4eq}$. This indicates that sulfate related to fireworks adds up to an acidic budget of particles. The excess sulfate observed from this analysis is approximately in quantitative agreement (within 20%) with the enhancement of the secondary sulfate during fireworks period. The peak in secondary sulfate at 23:00 LT is slightly offset from the main fireworks peak because secondary sulfate formed maximum peak 1 h later (00:00 LT 2 August).

[revised manuscript text omitted]

The unexplained variation (UEV, Paatero et al., 2004; Canonaco et al., 2013) which is a dimensionless quantity, describes how much of the measured variation (time or variable) is not explained by each PMF factor (Eq. S1, for variable $j$):

$$\quad UEV_{j,k,real} = \frac{\sum_{i=1}^{n}(|e_{i,j}|/s_{i,j})}{\sum_{i=1}^{n}\left(\left(\sum_{k=1}^{p}|g_{i,k}f_{k,j}|+|e_{i,j}|\right)/s_{i,j}\right)} \quad \text{for } k = 1, \dots\dots, p \text{ for } (x_{i,j}/s_{i,j}) > 2 \qquad (S1)$$

Where $f_{k,j}$ are the factor profiles and $g_{i,k}$ represent their time-dependent contributions. The index $i$ represents a specific point in time (up to $n$), $j$ is the variable, and $k$ is a factor (up to $p$). $e_{i,j}$ are the PMF-residuals, $x_{i,j}$ the input data and $s_{i,j}$ the measurement uncertainties. The UEV presented here is the real UEV for high S/N threshold >2, otherwise UEV will be considered as noisy UEV.

[Figure]

**Figure S5: Unexplained variation for each variable in FH PMF solutions: (1) constrained five-factor (black color); (2) unconstrained 5-factor (semi filled blue color), and (3) unconstrained three-factor (blue color).**

The stability of the fireworks factor profile was assessed by investigation of 1000 random BS runs on the FH dataset (see more details below). Solutions were accepted based on a K / S ratio in the fireworks factor of 2.76 ± 0.5, consistent with black powder composition (Dutcher et al., 1999) and the concentration peak between 40-45 $\mu g\ m^{-3}$ on 1 August 23:00 LT in the factor time series. This criterion yielded an acceptance rate of 21.3 %, with an average $Q/Q_{exp}$ of 1.3. The results of the BS analysis are provided in Fig. S6, where we show the fractional composition of the fireworks factor profiles from: (a) FH BS analysis (blue color); (b) 5-factor constrained FH PMF analysis (base case, in red color). The data and their corresponding uncertainty are given as box-whisker plot (bottom to top: p10-p25-p50-p75-p90) of selected 213 solutions out of 1000 BS runs. The displayed error bars for the base case correspond to the a-value of 0.1 used to constrain fireworks profile in the complete dataset PMF analysis. This figure shows that the variation in fireworks profiles retrieved from the bootstrap analysis is relatively stable despite the small number of data points; indeed, it generally denotes a smaller range of values than those allowed by the constrained fireworks profile in the full dataset PMF. Further, the resolved fireworks profile from FH BS solution agrees well with the base case, and the $Q/Q_{exp}$ values are reasonable. Taken together, these results suggest that the FH dataset contains sufficient variability to allow retrieval of a robust fireworks profile.

[Figure]

**Figure S6: Fireworks factor profiles from: (a) FH BS analysis (blue color); (b) five-factor constrained FH PMF analysis (base case, red color). The data and their corresponding uncertainties are given as box-whisker plot (bottom to top: p10-p25-p50-p75-p90) of selected 213 solutions from 1000 BS runs in blue color. Base case uncertainties are given ± 10 % as error bar by considering that *a*-value 0.1 is applied to constrain fireworks profile in complete dataset PMF analysis. Y-axis represents the fractional composition of the factor profile in row-wise for each factor in ng ng$^{-1}$.**

Based on $Q_{avg}$, defined as $\dfrac{\Sigma_{i=1}^{m}\Sigma_{j=1}^{n}\left(\frac{e_{ij}}{s_{ij}}\right)^{2}}{n*m}$ (*n*: sample time series, *m*: number of variables), the model explains the data variability very well when allowing for eight factors (Fig S7). Furthermore, we access the change in time-dependent $Q_{avg,i}$, $\dfrac{\Sigma_{i=1}^{m}\Sigma_{j=1}^{n}\left(\frac{e_{ij}}{s_{ij}}\right)^{2}}{n}$, when increasing the number of factors i.e., $\Delta Q_{avg,i}$ ; contribution to $Q$ for the (*p*)-factor solution minus that of the (*p*+1)-factor solution (Fig S8). A significant decrease in $\Delta Q_{avg,i}$ indicates that the structure in the residuals disappeared

5      with the additional factor. The removed structure is evident up to eight factors. Increasing the number of factors to nine yields a new mixed factor of the traffic-related and background dust factors (Fig. S1). Overall, a best ME-2 solution was observed up to a number of factors equal to eight.

For the 8-factor solution, we assess how well the different variables are explained by PMF using the quantity $\Delta Q_{avg,j}$ $\dfrac{\Sigma_{i=1}^{m}\Sigma_{j=1}^{n}\left(\frac{e_{ij}}{s_{ij}}\right)^{2}}{m}$ (Fig S9). $Q_{avg,j}$ shows that with 8 factors all variables are explained within their measurement uncertainty

10     except Si and Pb. This might be linked to an underestimation of the measurement uncertainty itself.

[Figure]

**Figure S7: $Q_{avg}$ as a function of the number of factors.**

[Figure]

**Figure S8: Change in the time-dependent contribution of $Q_{avg,i}$ ($\Delta Q_{avg,i}$) as a function of the number of factors.**

[Figure]

**Figure S94:** *Q$_{avg}$* **as a function of variables for the eight-factor solution.**

[Figure]

Histogram

**Figure S105: Histograms of variables as a function of residuals weighted by the uncertainty (residual/uncertainty) for the eight-factor solution.**

[Figure]

**Figure S116:** *a*-value statistics of the accepted solutions. *a*-values between 0 to 0.5 were explored during BS analysis. The average *a*-value of the selected solutions was ranging from 0.2 to 0.3 for the constrained factors. The selected *a*-values were homogeneously distributed over that range.

[Figure]

Figure S126: CBPF analysis (from left to right: 50th, 75th, 90th, 95th percentiles) of factors (sea salt, background dust, road dust, industrial) in terms of wind speed (m s$^{-1}$) and wind direction. The color code represents the probability of the factor contribution.

[Figure]

Figure S13<s>7</s>: Time series of the non-refractory aerosol components (measured with the ACSM), NO₂ and NOₓ concentration. The fireworks episodes are underlain in grey color.

Fig. S14<s>8</s> provides an estimate of the overall fireworks composition and temporal variability, complementing the PMF results (which are shown for reference). The figure is constructed in two stages. First, the time series of fireworks contributions to each element is estimated by subtracting the non-fireworks factors (NFF) from the original measurements. Then, the estimated fireworks contribution for each element is normalized by the total fireworks element contribution, and the displayed statistics are calculated. This is represented mathematically below, and the expression has been added to the main text. Note that the variation in fireworks profiles implied by this figure supports the representation of fireworks by 2 fireworks factors.

$$normalized\ concentration_{ij} = \frac{x_{ij} - \left(g_{ik}f_{kj}\right)_{k=NFF}}{\Sigma_j\left(x_{ij} - \left(g_{ik}f_{kj}\right)_{k=NFF}\right)} \tag{S2<s>1</s>}$$

where X represents the input data matrix for PMF, while G and F represent the factor time series and factor profiles driven by six non-fireworks factors (NFF). Here $i$ and $j$ denote time series and variables, respectively.

[Figure]

**Figure S148: Representation of fireworks data points (normalized concentration) in terms of median and 10-25-75-90th percentiles (bottom to top). Red and green dots denote the factor profiles of fireworks-I and fireworks-II, respectively.**

[Figure]

Figure S15̶0̶: Mg / Na ratio (red bars) for 24-h filter data analysed by ICP-OES. The gray lines represent the Mg / Na ratio range (0.132 -0.185) in marine aerosols. The Mg / Na ratio was 0.13 and 0.16 for 28 July and 30 July, respectively while for the rest of the days it was higher than 0.185.

[Figure]

Figure S16̶1̶: Bottom panel: Time series of the Cl concentration from the Xact, sea salt factor (left y-axis) from the PMF solution, and ACSM chloride concentration (right y-axis). Top panel: Wind speed (WS in m s$^{-1}$) and wind direction (WDir in degree) during the measurement period. The grey area in the bottom panel represents fireworks days.

[Figure]

Figure S17: Backward trajectory (produced from the FLEXTRA Trajectory Model; *https://folk.nilu.no/~andreas/flextra.html*) analysis at different heights during a sea salt event.

[Figure]

Figure S18: Scatter plot of Si vs. Ca scaled residuals: (a) PMF solution with one dust factor; (b) PMF solution with two dust factors.

[Figure]

**Figure S14: Scatter plot of Si vs. Ca.**

[Figure]

5  **Figure S2: Mean diurnal variations of the two dust factors (left-y axis) along with wind speed and wind direction (right y-axes) with error bars (one standard deviation).**

[Figure]

**Figure S219: Time series of the secondary sulfate factor (left y-axis) and mass concentration of SO₄, NH₄ from the ACSM (right y-axis).**

[Figure]

5 **Figure S22: Time series of equivalent concentrations of the ACSM (PM₁) NH₄ₑq, and NO₃ₑq and +2*SO₄ₑq. The NO₃ₑq is stacked on 2*SO₄ₑq.**

---

## Author Response (AR3)

**Comments to the Author:**

*The authors have reasonably replied to the critical comments of the three Anonymous Referees and they have modified their manuscript accordingly. However, the comments given below should be addressed and many alterations are needed for the Main text and Supplement before the manuscript can be published in ACP.*

Dear Prof. Dr. Maenhaut,

Thank you very much for giving us this opportunity to resubmit our final version of the manuscript. We addressed all comments (in italic typeset) and prepared a point-to-point response (in regular typeset). Changes to the manuscript are indicated in blue font. In the following, page and lines references refer to the previous version of the manuscript reviewed by Co-Editor.

*For the Main text:*

*Page 1, line 10: Replace "were performed" by "was performed".*

Done.

*Page 1, line 15: Replace "brackets" by "parentheses" or alternatively by "round brackets".*

Done.

*Page 2, line 20: Replace "Yu, 2013" by "Yu et al., 2013".*

Done.

*Page 2, line 29: "a very limited" is exaggerated; I suggest replacing it by "a limited".*

Done.

*Page 4, line 15: Replace "balck carbon" by "black carbon".*

Done.

*Page 4, line 22: Replace "1μm" by "1 μm".*

Done.

*Page 5, line 11: Place the "a" of "a priori" also in italic.*

Done.

*Page 6, line 16, page 10, line 12, and page 13, line 7: Replace "e.g." by "e.g.,".*

Done.

Page 7, lines 7-9: The authors write that "solutions with three to 10 factors" were examined, and then they state that "The unconstrained PMF solution resulted in mixed factors ... even for

higher numbers of factors (Fig. S1)" but Fig. S1 shows a solution with only 9 factors. This is quite confusing. Rephrasing is needed here.

Done. We modified the text in the manuscript on page 7 line7-9 as follows:

The unconstrained PMF solution resulted in mixed factors, such as sea salt mixed with fireworks in all factors solutions. We show an example of a mixed nine-factor solution in Fig. S1.

*Page 7, line 10: Replace "is very" by "ratio is very".*

Done.

*Page 7, line 16: Abbreviations and acronyms (here "LT") should be defined (written full-out when first used; therefore replace "LT" here by "local time (LT)".*

Done. We modified it as follows:

LT (local time = Coordinated Universal Time + 2 h).

*Page 7, line 28: Replace "Fig. S2" by "Fig. S3".*

Done.

*Page 7, line 32: The sentence "Although these r2 are quite similar, the solution characteristics are notably different" is unclear to me; "similar" to what and "different" from what? Rephrasing is needed.*

We thank to Co-Editor for noticing it. We have rewritten this paragraph in the manuscript on page 7 line 27-32 as follows:

Although these $r^2$ are quite similar at all 100 seeds, the solution characteristics are notably different at 24 seeds ($r^2 = 0.91$) and 76 seeds ($r^2 \leq 0.88$). For the nine-factor solution shown in Fig. S1, the secondary sulfate factor did not respond significantly to the fireworks event, while the other factors time series, such as the sea salt and a mixed traffic plus K-rich factors, were enhanced during the fireworks peaks. In contrast, for the solutions at 76 seeds, visible contamination (i.e., concentration spikes) during the fireworks plumes were observed in the secondary sulfate factor, suggesting mathematical mixing. S is one of the major components of fireworks emissions, the composition of which is highly variable. Because of their high sensitivity (and thus high signal-to-uncertainty ratio), imperfections in the model description of the fireworks composition yield high-signal residuals which strongly influence $Q$. The model responds by apportioning fireworks residuals to the other factors during the fireworks days. A similar issue also occurred for the sea salt factor due to the significant amount of Cl in the fireworks factor profile. Therefore, such events are often excluded from traditional PMF analyses (i.e., time periods removed from the input matrix), to avoid modelling errors due to the pulling of a solution by outliers. Here we take a different approach, exploiting the rotational control available in ME-2 to isolate environmentally reasonable, unmixed solutions. The $Q/Q_{exp}$ values for all 100 seeds were in the range of 0.41- 0.45 (Fig. S2). The scaled residual

(over the time series) of S in this solution (Fig. S1) was within the range of ± 3 with very small values, as shown in Fig. S3. The strong influence of fireworks in this solution yielded small values of the scaled residual for S which are consistent also with lower-than-expected $Q/Q_{exp}$ (0.4) values, suggesting an overestimation of uncertainties by the generic error model used herein (see Eqs. 6 and 7).

*Page 7, line 30: I presume that "r2 = 0.91" refers to the correlation of the sulfate factor with ACSM sulfate in the nine-factor solution shown in Fig. S1. If so, it is strange to call the nine-factor solution "the r2 = 0.91 solution"; it should in this case be replaced by "the nine-factor solution shown in Fig. S1".*

Done.

*Page 8, line 2, and page 10, line 8: Replace "i.e." by "i.e.,".*

Done.

*Page 8, line 5: Replace "composition yields" by "composition yield".*

Done.

*Page 8, line 27: Replace "fireworks factor" by "a fireworks factor".*

Done.

*Page 8, line 34: Replace "but same as" by "but the same as the".*

Done.

*Page 8, line 34, and page 9, lines 1-2: The sentence "The increase in Q is significant over the constrained five-factor FH PMF and unconstrained five-factor because its UEV for each variable is between 2 % to 16 % (average ~10 %), while is it between 1.5 % to 15 % (average ~5 %) (Fig. S5)" is unclear to me. "increase in Q" for which PMF solution?; in "and unconstrained five-factor" which PMF solution is meant here? and for which PMF solutions do the percentages apply?*

We have modified the text in the manuscript on page 8 line 34 and page 9 line 1-2 as follows:

The increase in $Q$ is significant over the constrained five-factor FH PMF (unexplained variation (UEV) for each variable is between 2 % to 16 % with an average ~10 %), as compared to the unconstrained five-factor FH PMF (UEV for each variable is between 1.5 % to 15 % with an average ~5 %), as shown in Fig. S5.

*Page 9, line 23: Replace "variables were" by "variables showed".*

Done.

*Page 10, lines 27-29, and page 32, caption of Figure 3a: The shaded areas that represent the uncertainties for the fireworks period are only visible after magnifying the Figure by 800%. Modification of the Figure and alteration in the text are needed.*

We agree with Co-editor for visibility of Fig. 3a. Therefore, we presented magnified version of uncertainties during the fireworks period only for the sea salt and the secondary sulfate factor in supplement (Fig. S12).

We have modified the text in the manuscript on page 10 line 29 as follows:

Uncertainties of ±5 % were calculated as the standard deviation of the actual secondary sulfate concentrations to the predicted values and included for the fireworks period only in Fig. 3a and Fig S12.

We have also modified the text in the manuscript on page 11 line 4 as follows:

This value is used as the uncertainty of the sea salt factor time series (during the fireworks period only) in Fig. 3a and Fig. S12.

We have also modified the Fig. 3a caption on page 32 line 5 as follows:

[Figure]

**Figure 3: (a) Time series of the PM$_{10el}$ sources and relative contributions of the different sources over time; shaded areas indicate the uncertainties (interquartiles) of selected bootstrap runs; grey background color represents the fireworks period; the estimated uncertainties of the secondary sulfate (± 5%) and the sea salt factors (± 42%) during the fireworks period are added as error bars (magnified version is shown in Fig. S12); (b) Mean relative contributions of PM$_{10el}$ sources. The average concentration represents the mean value of apportioned sources in PM$_{10el}$ which is the sum of 14 elements.**

We have added the magnified version of uncertainties for the sea salt and the secondary sulfate factor during the fireworks period only in the supplement on page 10 line 5 as follows:

[Figure]

**Figure S12: The estimated uncertainties of the secondary sulfate (± 5%) and the sea salt factors (± 42%) during the fireworks period are added as error bars.**

*Page 11, lines 22-23, and caption of Fig. 3b: It is not mentioned in the caption of Fig. 3b that the fireworks factors are excluded in the average concentration of 3134.43 ng m-3. If they are not, "(excluding the fireworks factors)" should be removed in line 23. Furthermore, there are too many significant figures (significant digits) given for that average concentration; 3 significant figures suffice.*

We apologise for this mistake. Fig. 3b shows the averaged total $PM_{10el}$ mass and relative contributions of the $PM_{10el}$ mass (including fireworks factors). Therefore, we removed "(excluding the fireworks factors)" from the manuscript on page 11 line 22-23. We have also corrected the significant figures (significant digits) given for that average concentration as shown above in Fig. 3a.

*Page 11, line 29: Replace "Fig.3a" by "Fig. 3a".*

Done.

*Page 12, line 9: Replace "atomic ratio" by "molecular/atomic ratio".*

Done.

*Page 12, line 17: Replace "in (Pongpiachan et al., 2018)" by "in Pongpiachan et al. (2018)".*

Done.

*Page 12, line 20: Abbreviations and acronyms (here "CEST") should be defined (written full-out when first used; since "CEST" is not used further in the manuscript, it should be replaced here by "Central European Standard Time".*

We have now removed CEST from here based on previous changes on page 7 line 16.

*Page 14, line 10: Replace "where they are" by "where the two elements are".*

Done.

*Page 14, lines 12-13: Replace "where the non-exhaust non-exhaust traffic-related" by "where the non-exhaust traffic-related".*

Done.

*Page 15, line 26: Replace "in secondary" by "in the secondary".*

Done.

*Page 15, line 31: Replace "during fireworks" by "during the fireworks".*

Done.

*Page 15, line 32: It is unclear to me what the authors mean by "because secondary sulfate formed maximum peak 1 h later"; should this text perhaps be replaced by "because the secondary sulfate formed maximum peaks 1 h later"?*

We apologise for the misunderstanding. We have modified the manuscript at line 32 on page 15 as follows:

The peak in the secondary sulfate observed at 23:00 LT 1 August is slightly offset from the main fireworks peak because the secondary sulfate formed maximum peaks 1 h later (00:00 LT 2 August).

*Page 16, line 1: Replace "makes secondary" by "makes the secondary".*

Done.

*Page 16, line 16: Replace "Switzerland;" by "Switzerland,".*

Done.

*Page 16, line 24: Replace "of data" by "of the data".*

Done.

*Page 17, line 8: Replace "performed measurement. MF and RF analysed data" by "performed the measurement. MF and RF analysed the data".*

Done.

*Page 18, line 15: Replace "Environ." by "Environ.: X".*

Done.

*Page 20, line 18: Replace "Geophys. Res.," by "Geophys. Res.: Atmos., 124,".*

Done.

*Page 21, line 24: Replace "Meas.Tech." by "Meas. Tech.".*

Done.

*Page 25, line 9: Replace "of Reseach" by "of Research".*

Done.

Page 26, line 24: Replace "Environ." by "Environ.: X".

Done.

*Page 28, line 21: "Wang et al., 2018" should start on a new line.*

Done.

*Page 28, lines 27-28: There is not referred to this reference within the text.*

We removed this reference from the manuscript.

*Page 31, line 2: Replace "of PMF" by "of the finally retained 8-factor PMF".*

Done.

*For the Supplement:*

*Page 4, line 4: Replace "Paatero et al." by "Paatero".*

Done.

*Page 4, line 5: Replace "S1, for variable j)" by "S1) for variable j".*

Done.

*Page 5, line 3: Replace "11 data" by "the 11 data".*

Done.

Page 5, line 4: Replace "number of" by "the number of".

Done.

*Page 5, line 12: Replace "constrain fireworks" by "constrain the fireworks".*

Done.

*Page 12, line 3: Replace "underlain in" by "indicated in".*

Done.

[revised manuscript text omitted]

The unexplained variation (UEV, Paatero et al., 2004; Canonaco et al., 2013) which is a dimensionless quantity, describes how much of the measured variation (time or variable) is not explained by each PMF factor (Eq. S1), for variable *j*):

$$UEV_{j,k,real} = \frac{\sum_{i=1}^{n}(|e_{i,j}|/s_{i,j})}{\sum_{i=1}^{n}\left((\sum_{k=1}^{p}|g_{i,k}f_{k,j}|+|e_{i,j}|)/s_{i,j}\right)} \text{ for } k = 1, \dots, p \text{ for } (x_{i,j}/s_{i,j}) > 2 \quad \text{(S1)}$$

Where $f_{k,j}$ are the factor profiles and $g_{i,k}$ represent their time-dependent contributions. The index *i* represents a specific point in time (up to *n*), *j* is the variable, and *k* is a factor (up to *p*). $e_{i,j}$ are the PMF-residuals, $x_{i,j}$ the input data and $s_{i,j}$ the measurement uncertainties. The UEV presented here is the real UEV for high S/N threshold >2, otherwise UEV will be considered as noisy UEV.

[Figure]

Figure S5: Unexplained variation for each variable in FH PMF solutions: (1) constrained five-factor (black color); (2) unconstrained 5-factor (semi filled blue color), and (3) unconstrained three-factor (blue color).

This does still leave the question of how many points are too few. We test this issue with a bootstrap (BS) analysis on the FH dataset, which resamples the 11 data points randomly by repeating or removing data points from the original 11 data points. If the number of time points in the dataset were too small, one would expect a high degree of variability in the output solutions.

5  The stability of the fireworks factor profile was assessed by investigation of 1000 random BS runs on the FH dataset (see more details below). Solutions were accepted based on a K / S ratio in the fireworks factor of 2.76 ± 0.5, consistent with black powder composition (Dutcher et al., 1999) and the concentration peak between 40 - 45 µg m$^{-3}$ on 1 August 23:00 LT in the factor time series. This criterion yielded an acceptance rate of 21.3 %, with an average $Q/Q_{exp}$ of 1.3. The results of the BS analysis are provided in Fig. S6, where we show the fractional composition of the fireworks factor profiles from: (a) FH BS

10  analysis (blue color); (b) five-factor constrained FH PMF analysis (base case, in red color). The data and their corresponding uncertainty are given as box-whisker plot (bottom to top: p10-p25-p50-p75-p90) of selected 213 solutions out of 1000 BS runs. The displayed error bars for the base case correspond to the $a$-value of 0.1 used to constrain the fireworks profile in the complete dataset PMF analysis. This figure shows that the variation in fireworks profiles retrieved from the bootstrap analysis is relatively stable despite the small number of data points; indeed, it generally denotes a smaller range of values than those

15  allowed by the constrained fireworks profile in the full dataset PMF. Further, the resolved fireworks profile from FH BS solution agrees well with the base case, and the $Q/Q_{exp}$ values are reasonable. Taken together, these results suggest that the FH dataset     contains     sufficient     variability     to     allow     retrieval     of     a     robust     fireworks     profile.

[Figure]

**Figure S6: Fireworks factor profiles from: (a) FH BS analysis (blue color); (b) five-factor constrained FH PMF analysis (base case, red color). The data and their corresponding uncertainties are given as box-whisker plot (bottom to top: p10-p25-p50-p75-p90) of selected 213 solutions from 1000 BS runs in blue color. Base case uncertainties are given ± 10 % as error bar by considering that $a$-**

**value 0.1 is applied to constrain fireworks profile in complete dataset PMF analysis. Y-axis represents the fractional composition of the factor profile in row-wise for each factor in ng ng$^{-1}$.**

Based on $Q_{avg}$, defined as $\dfrac{\sum_{i=1}^{m}\sum_{j=1}^{n}\left(\frac{e_{ij}}{s_{ij}}\right)^2}{n*m}$ ($n$: sample time series, $m$: number of variables), the model explains the data variability

5   very well when allowing for eight factors (Fig S7). Furthermore, we access the change in time-dependent $Q_{avg,i}$, $\dfrac{\sum_{i=1}^{m}\sum_{j=1}^{n}\left(\frac{e_{ij}}{s_{ij}}\right)^2}{n}$,

when increasing the number of factors i.e., $\Delta Q_{avg,i}$ ; contribution to $Q$ for the ($p$)-factor solution minus that of the ($p$+1)-factor solution (Fig S8). A significant decrease in $\Delta Q_{avg,i}$ indicates that the structure in the residuals disappeared with the additional factor. The removed structure is evident up to eight factors. Increasing the number of factors to nine yields a new mixed factor of the traffic-related and background dust factors (Fig. S1). Overall, a best ME-2 solution was observed up to a number of

10   factors equal to eight.

For the 8-factor solution, we assess how well the different variables are explained by PMF using the quantity $\Delta Q_{avg,j}$

$\dfrac{\sum_{i=1}^{m}\sum_{j=1}^{n}\left(\frac{e_{ij}}{s_{ij}}\right)^2}{m}$ (Fig S9). $Q_{avg,j}$ shows that with 8 factors all variables are explained within their measurement uncertainty except

Si and Pb. This might be linked to an underestimation of the measurement uncertainty itself.

[Figure]

15   **Figure S7: $Q_{avg}$ as a function of the number of factors.**

[Figure]

**Figure S8: Change in the time-dependent contribution of $Q_{avg,i}$ ($\Delta Q_{avg,i}$) as a function of the number of factors.**

[Figure]

Figure S9: $Q_{avg}$ as a function of variables for the eight-factor solution.

[Figure]

**Figure S10: Histograms of variables as a function of residuals weighted by the uncertainty (residual/uncertainty) for the eight-factor solution.**

[Figure]

**Figure S11: a-value statistics of the accepted solutions. a-values between 0 to 0.5 were explored during BS analysis. The average a-value of the selected solutions was ranging from 0.2 to 0.3 for the constrained factors. The selected a-values were homogeneously distributed over that range.**

[Figure]

**Figure S12: The estimated uncertainties of the secondary sulfate (± 5%) and the sea salt factors (± 42%) during the fireworks period are added as error bars.**

[Figure]

**Figure S132: CBPF analysis (from left to right: 50th, 75th, 90th, 95th percentiles) of factors (sea salt, background dust, road dust, industrial) in terms of wind speed (m s-1) and wind direction. The color code represents the probability of the factor contribution.**

[Figure]

**Figure S14: Time series of the non-refractory aerosol components (measured with the ACSM), NO₂ and NO$_x$ concentration. The fireworks episodes are  indicated in grey color.**

Fig. S15 provides an estimate of the overall fireworks composition and temporal variability, complementing the PMF results (which are shown for reference). The figure is constructed in two stages. First, the time series of fireworks contributions to each element is estimated by subtracting the non-fireworks factors (NFF) from the original measurements. Then, the estimated fireworks contribution for each element is normalized by the total fireworks element contribution, and the displayed statistics are calculated. This is represented mathematically below, and the expression has been added to the main text. Note that the variation in fireworks profiles implied by this figure supports the representation of fireworks by 2 fireworks factors.

$$normalized\ concentration_{ij} = \frac{x_{ij} - \left(g_{ik}f_{kj}\right)_{k=NFF}}{\Sigma_j\left(x_{ij} - \left(g_{ik}f_{kj}\right)_{k=NFF}\right)} \tag{S2}$$

where X represents the input data matrix for PMF, while G and F represent the factor time series and factor profiles driven by six non-fireworks factors (NFF). Here *i* and *j* denote time series and variables, respectively.

[Figure]

**Figure S154: Representation of fireworks data points (normalized concentration) in terms of median and 10-25-75-90th percentiles (bottom to top). Red and green dots denote the factor profiles of fireworks-I and fireworks-II, respectively.**

[Figure]

Figure S16̶5̶: Mg / Na ratio (red bars) for 24-h filter data analysed by ICP-OES. The gray lines represent the Mg / Na ratio range (0.132 -0.185) in marine aerosols. The Mg / Na ratio was 0.13 and 0.16 for 28 July and 30 July, respectively while for the rest of the days it was higher than 0.185.

[Figure]

Figure S17̶6̶: Bottom panel: Time series of the Cl concentration from the Xact, sea salt factor (left y-axis) from the PMF solution, and ACSM chloride concentration (right y-axis). Top panel: Wind speed (WS in m s[-1]) and wind direction (WDir in degree) during the measurement period. The grey area in the bottom panel represents fireworks days.

[Figure]

Figure S18̶7̶: **Backward trajectory (produced from the FLEXTRA Trajectory Model; *https://folk.nilu.no/~andreas/flextra.html*) analysis at different heights during a sea salt event.**

[Figure]

Figure S19̶8̶: **Scatter plot of Si vs. Ca scaled residuals: (a) PMF solution with one dust factor; (b) PMF solution with two dust factors.**

[Figure]

**Figure S20: Scatter plot of Si vs. Ca.**

[Figure]

**Figure S21: Mean diurnal variations of the two dust factors (left-y axis) along with wind speed and wind direction (right y-axes) with error bars (one standard deviation).**

[Figure]

**Figure S22: Time series of the secondary sulfate factor (left y-axis) and mass concentration of $SO_4$, $NH_4$ from the ACSM (right y-axis).**

[Figure]

**Figure S22:** Time series of equivalent concentrations of the ACSM (PM$_1$) NH$_{4eq}$, and NO$_{3eq}$ and +2*SO$_{4eq}$. The NO$_{3eq}$ is stacked on 2*SO$_{4eq}$.

---

## Author Response (AR4)

**Comments to the Author:**

*Some alterations are still needed for the Main text and the Supplement before the manuscript can be published in ACP:*

Dear Prof. Dr. Maenhaut,

Thank you very much for giving us this opportunity to resubmit our final version of the manuscript. We addressed all comments (in italic typeset) and prepared a point-to-point response (in regular typeset). Changes to the manuscript are indicated in blue font. In the following, page and lines references refer to the previous version of the manuscript reviewed by Co-Editor.

*For the Main text:*

*Page 7, lines 24-26: The sentence "For the nine-factor ... fireworks peaks" is unclear to me; what is meant by "the secondary sulfate factor did not respond significantly to the fireworks event"?; the secondary sulfate factor in Fig. S1 clearly shows a peak during the fireworks event; furthermore, I do not see that the "mixed traffic plus K-rich factor was enhanced during the fireworks peaks".*

We have rewritten this line as follows:

For the nine-factor solution shown in Fig. S1, the secondary sulfate factor was slightly enhanced, which is in agreement with the ACSM sulfate during the fireworks peaks (see details in Sect. 4.2 "secondary sulfate"), while the sea salt factor was strongly enhanced during the fireworks peaks.

*Page 8, lines 28-30: The sentence "The increase in Q is significant over the constrained five-factor FH PMF (unexplained variation (UEV) for each variable is between 2 % to 16 % with an average ~10 %), as compared to the unconstrained five-factor FH PMF (UEV for each variable is between 1.5 % to 15 % with an average ~5 %), as shown in Fig. S5" is unclear to me. Should it perhaps be changed into "The increase in Q for the constrained five-factor FH PMF (unexplained variation (UEV) for each variable is between 2 % to 16 % with an average ~10 %) is significant when compared to the unconstrained five-factor FH PMF (UEV for each variable is between 1.5 % to 15 % with an average ~5 %), as shown in Fig. S5"?*

We agree with Co-Editor and accept the suggestion to rewrite it as follows:

The increase in Q for the constrained five-factor FH PMF (unexplained variation (UEV) for each variable is between 2 % to 16 % with an average ~10 %) is significant when compared to the unconstrained five-factor FH PMF (UEV for each variable is between 1.5 % to 15 % with an average ~5 %), as shown in Fig. S5.

*Page 20, line 8: Replace "Atmos., 124," by "Atmos., 124, 11,595-11,613,"*

Done.

*For the Supplement:*

*Page 10: The caption of Fig. S12 should be modified.*

We have modified the caption below. We also corrected the error bars (estimated uncertainties) in Fig. S12 and Fig. 3 for sea salt (± 42%) and the secondary sulfate (± 5%) factors during the fireworks period. We apologise for this mistake.

[Figure]

**Figure S12: The factor time series of the sea salt and the secondary sulfate factor during the fireworks period (31.07.2015-04.08.2015). The estimated uncertainties of the sea salt (± 42%) and the secondary sulfate (± 5%) factors during the fireworks period are added as error bars (standard deviation).**

[Figure]

**Figure 3: (a) Time series of the PM10el sources and relative contributions of the different sources over time; shaded areas indicate the uncertainties (interquartiles) of selected bootstrap runs; grey background color represents the**

fireworks period; the estimated uncertainties of the secondary sulfate (± 5%) and the sea salt factors (± 42%) during the fireworks period are added as error bars (magnified version is shown in Fig. S12); (b) Mean relative contributions of PM10el sources. The average concentration represents the mean value of apportioned sources in PM10el which is the sum of 14 elements.

Additional changes in the main text from the author:

We also have modified and rearranged the text in the manuscript on page 4 line 14 as follows:

[revised manuscript text omitted]

The unexplained variation (UEV, Paatero, 2004; Canonaco et al., 2013) which is a dimensionless quantity, describes how

5    much of the measured variation (time or variable) is not explained by each PMF factor (Eq. S1) for variable $j$:

$$UEV_{j,k,real} = \frac{\sum_{i=1}^{n}(|e_{i,j}|/s_{i,j})}{\sum_{i=1}^{n}\left((\sum_{k=1}^{p}|g_{i,k}f_{k,j}|+|e_{i,j}|)/s_{i,j}\right)} \quad \text{for } k = 1, \ldots \ldots, p \text{ for } (x_{i,j}/s_{i,j}) > 2 \tag{S1}$$

Where $f_{k,j}$ are the factor profiles and $g_{i,k}$ represent their time-dependent contributions. The index $i$ represents a specific point in time (up to $n$), $j$ is the variable, and $k$ is a factor (up to $p$). $e_{i,j}$ are the PMF-residuals, $x_{i,j}$ the input data and $s_{i,j}$ the measurement uncertainties. The UEV presented here is the real UEV for high S/N threshold >2, otherwise UEV will be

10    considered as noisy UEV.

[Figure]

**Figure S5: Unexplained variation for each variable in FH PMF solutions: (1) constrained five-factor (black color); (2) unconstrained 5-factor (semi filled blue color), and (3) unconstrained three-factor (blue color).**

This does still leave the question of how many points are too few. We test this issue with a bootstrap (BS) analysis on the FH dataset, which resamples the 11 data points randomly by repeating or removing data points from the original 11 data points. If the number of time points in the dataset were too small, one would expect a high degree of variability in the output solutions.

5    The stability of the fireworks factor profile was assessed by investigation of 1000 random BS runs on the FH dataset (see more details below). Solutions were accepted based on a K / S ratio in the fireworks factor of 2.76 ± 0.5, consistent with black powder composition (Dutcher et al., 1999) and the concentration peak between 40 - 45 µg m$^{-3}$ on 1 August 23:00 LT in the factor time series. This criterion yielded an acceptance rate of 21.3 %, with an average $Q/Q_{exp}$ of 1.3. The results of the BS analysis are provided in Fig. S6, where we show the fractional composition of the fireworks factor profiles from: (a) FH BS

10   analysis (blue color); (b) five-factor constrained FH PMF analysis (base case, in red color). The data and their corresponding uncertainty are given as box-whisker plot (bottom to top: p10-p25-p50-p75-p90) of selected 213 solutions out of 1000 BS runs. The displayed error bars for the base case correspond to the $a$-value of 0.1 used to constrain the fireworks profile in the complete dataset PMF analysis. This figure shows that the variation in fireworks profiles retrieved from the bootstrap analysis is relatively stable despite the small number of data points; indeed, it generally denotes a smaller range of values than those

15   allowed by the constrained fireworks profile in the full dataset PMF. Further, the resolved fireworks profile from FH BS solution agrees well with the base case, and the $Q/Q_{exp}$ values are reasonable. Taken together, these results suggest that the FH dataset     contains     sufficient     variability     to     allow     retrieval     of     a     robust     fireworks     profile.

[Figure]

Figure S6: Fireworks factor profiles from: (a) FH BS analysis (blue color); (b) five-factor constrained FH PMF analysis (base case,
20   red color). The data and their corresponding uncertainties are given as box-whisker plot (bottom to top: p10-p25-p50-p75-p90) of
selected 213 solutions from 1000 BS runs in blue color. Base case uncertainties are given ± 10 % as error bar by considering that $a$-

value 0.1 is applied to constrain fireworks profile in complete dataset PMF analysis. Y-axis represents the fractional composition of the factor profile in row-wise for each factor in ng ng$^{-1}$.

Based on $Q_{avg}$, defined as $\dfrac{\sum_{i=1}^{m}\sum_{j=1}^{n}\left(\frac{e_{ij}}{s_{ij}}\right)^2}{n*m}$ ($n$: sample time series, $m$: number of variables), the model explains the data variability

5 very well when allowing for eight factors (Fig. S7). Furthermore, we access the change in time-dependent $Q_{avg,i}$, $\dfrac{\sum_{i=1}^{m}\sum_{j=1}^{n}\left(\frac{e_{ij}}{s_{ij}}\right)^2}{n}$,

when increasing the number of factors i.e., $\Delta Q_{avg,i}$ ; contribution to $Q$ for the ($p$)-factor solution minus that of the ($p$+1)-factor solution (Fig. S8). A significant decrease in $\Delta Q_{avg,i}$ indicates that the structure in the residuals disappeared with the additional factor. The removed structure is evident up to eight factors. Increasing the number of factors to nine yields a new mixed factor of the traffic-related and background dust factors (Fig. S1). Overall, a best ME-2 solution was observed up to a number of

10 factors equal to eight.

For the 8-factor solution, we assess how well the different variables are explained by PMF using the quantity $\Delta Q_{avg,j}$

$\dfrac{\sum_{i=1}^{m}\sum_{j=1}^{n}\left(\frac{e_{ij}}{s_{ij}}\right)^2}{m}$ (Fig. S9). $Q_{avg,j}$ shows that with 8 factors all variables are explained within their measurement uncertainty except

Si and Pb. This might be linked to an underestimation of the measurement uncertainty itself.

[Figure]

15 **Figure S7: $Q_{avg}$ as a function of the number of factors.**

[Figure]

**Figure S8: Change in the time-dependent contribution of $Q_{avg,i}$ ($\Delta Q_{avg,i}$) as a function of the number of factors.**

[Figure]

**Figure S9:** $Q_{avg}$ **as a function of variables for the eight-factor solution.**

[Figure]

**Figure S10: Histograms of variables as a function of residuals weighted by the uncertainty (residual/uncertainty) for the eight-factor solution.**

[Figure]

**Figure S11: a-value statistics of the accepted solutions. a-values between 0 to 0.5 were explored during BS analysis. The average a-value of the selected solutions was ranging from 0.2 to 0.3 for the constrained factors. The selected a-values were homogeneously distributed over that range.**

[Figure]

[Figure]

**Figure S12:** The factor time series of the sea salt and the secondary sulfate during the fireworks period (31.07.2015-04.08.2015). The estimated uncertainties of the sea salt (± 42%) and the secondary sulfate (± 5%) factors during the fireworks period are added as error bars (standard deviation).

[Figure]

**Figure S13: CBPF analysis (from left to right: 50th, 75th, 90th, 95th percentiles) of factors (sea salt, background dust, road dust, industrial) in terms of wind speed (m s-1) and wind direction. The color code represents the probability of the factor contribution.**

[Figure]

**Figure S14: Time series of the non-refractory aerosol components (measured with the ACSM), NO₂ and NOₓ concentration. The fireworks episodes are indicated in grey color.**

Fig. S15 provides an estimate of the overall fireworks composition and temporal variability, complementing the PMF results (which are shown for reference). The figure is constructed in two stages. First, the time series of fireworks contributions to each element is estimated by subtracting the non-fireworks factors (NFF) from the original measurements. Then, the estimated fireworks contribution for each element is normalized by the total fireworks element contribution, and the displayed statistics are calculated. This is represented mathematically below, and the expression has been added to the main text. Note that the variation in fireworks profiles implied by this figure supports the representation of fireworks by 2 fireworks factors.

$$normalized\ concentration_{ij} = \frac{x_{ij} - (g_{ik}f_{kj})_{k=NFF}}{\sum_j(x_{ij} - (g_{ik}f_{kj})_{k=NFF})} \tag{S2}$$

where X represents the input data matrix for PMF, while G and F represent the factor time series and factor profiles driven by six non-fireworks factors (NFF). Here *i* and *j* denote time series and variables, respectively.

[Figure]

**Figure S15: Representation of fireworks data points (normalized concentration) in terms of median and 10-25-75-90th percentiles (bottom to top). Red and green dots denote the factor profiles of fireworks-I and fireworks-II, respectively.**

[Figure]

**Figure S16: Mg / Na ratio (red bars) for 24-h filter data analysed by ICP-OES. The gray lines represent the Mg / Na ratio range (0.132 -0.185) in marine aerosols. The Mg / Na ratio was 0.13 and 0.16 for 28 July and 30 July, respectively while for the rest of the days it was higher than 0.185.**

[Figure]

**Figure S17: Bottom panel: Time series of the Cl concentration from the Xact, sea salt factor (left y-axis) from the PMF solution, and ACSM chloride concentration (right y-axis). Top panel: Wind speed (WS in m s$^{-1}$) and wind direction (WDir in degree) during the measurement period. The grey area in the bottom panel represents fireworks days.**

[Figure]

**Figure S18: Backward trajectory (produced from the FLEXTRA Trajectory Model;** *https://folk.nilu.no/~andreas/flextra.html*) **analysis at different heights during a sea salt event.**

[Figure]

**Figure S19: Scatter plot of Si vs. Ca scaled residuals: (a) PMF solution with one dust factor; (b) PMF solution with two dust factors.**

[Figure]

**Figure S20: Scatter plot of Si vs. Ca.**

[Figure]

**Figure S21: Mean diurnal variations of the two dust factors (left-y axis) along with wind speed and wind direction (right y-axes) with error bars (one standard deviation).**

[Figure]

**Figure S22: Time series of the secondary sulfate factor (left y-axis) and mass concentration of SO₄, NH₄ from the ACSM (right y-axis).**

[Figure]

**Figure S23: Time series of equivalent concentrations of the ACSM (PM$_1$) NH$_{4eq}$, and NO$_{3eq}$ and +2\*SO$_{4eq}$. The NO$_{3eq}$ is stacked on 2\*SO$_{4eq}$.**